# Machine learning of Antarctic firn density by combining radiometer and scatterometer remote sensing data

Weiran Li [1], Sanne B. M. Veldhuijsen [3], and Stef Lhermitte [2, 1]

[1]Department of Geoscience and Remote Sensing, Delft University of Technology, Delft, The Netherlands
[2]Department of Earth & Environmental Sciences, KU Leuven, Leuven, Belgium
[3]Institute for Marine and Atmospheric research Utrecht (IMAU), Utrecht, The Netherlands

**Correspondence:** Weiran Li (w.li-7@tudelft.nl)

**Abstract.** Firn density plays a crucial role in assessing the surface mass balance of the Antarctic ice sheet. However, our understanding of the spatial and temporal variations in firn density is limited due to i) spatial and temporal limitations of in situ measurements, ii) potential modelling uncertainties, and iii) lack of firn density products driven by satellite remote sensing data. To address this gap, this paper explores the potential of satellite microwave radiometer (SMISS) and scatterometer (AS-CAT) observations for assessing spatial and temporal dynamics of dry firn density over the Antarctic ice sheet. Our analysis demonstrates a clear relation between density anomalies at a depth of 40 cm and fluctuations in satellite observations. However, a linear relationship with individual satellite observations is insufficient to explain the spatial and temporal variation of snow density. Hence, we investigate the potential of a non-linear Random Forest (RF) machine learning approach trained on radiometer and scatterometer data to derive the spatial and temporal variations in dry firn density. In the estimation process, ten years of SSMIS observations (brightness temperature) and ASCAT observations (backscatter intensity) are used as input features to a random forest (RF) regressor. The regressor is first trained on time series of modelled density and satellite observations at randomly sampled pixels, and then applied to estimate densities in dry firn areas across Antarctica. The RF results reveal a strong agreement between the spatial patterns estimated by the RF regressor and the modelled densities. The estimated densities exhibit an error of $\pm 10\ kg\ m^{-3}$ in the interior of the ice sheet and $\pm 35\ kg\ m^{-3}$ towards the ocean. However, the temporal patterns show some discrepancies, as the RF regressor tends to overestimate summer densities, except for high-elevation regions in East Antarctica and specific areas in West Antarctica. These errors may be attributed to underestimations of short-term or seasonal variations in the modelled density and the limitation of RF in extrapolating values outside the training data. Overall, our study presents a potential method for estimating unknown Antarctic firn densities using known densities and satellite parameters.

## 1 Introduction

The accelerated loss of mass from the Antarctic Ice Sheet, a trend anticipated to persist in the coming decades and centuries, underscores Antarctica's pivotal role as a major source of uncertainty in projecting future sea level rise (Pattyn and Morlighem, 2020). Recognising the critical contribution to sea level rise uncertainty highlights the urgency of comprehending Antarctica's surface mass balance (SMB). A typical method to estimate SMB of the Antarctic ice sheet is to convert satellite altimetry height

measurements into SMB (Zwally et al., 2005; Kuipers Munneke et al., 2015; Schröder et al., 2019) with the help of firn (an intermediate state between snow and glacial ice; van den Broeke, 2008; Amory et al., 2024) density. In Antarctica, firn density is highly variable in space and time due to the varying surface climate conditions (Craven and Allison, 1998; Li and Zwally, 2004; van den Broeke, 2008; Fujita et al., 2016). Therefore, it is necessary to continuously monitor firn density in Antarctica.

A variety of methods has been developed to assess firn density. In situ measurements from firn cores, snow pits and local near-infrared pictures are precious for accurately understanding firn densities; however, these measurements are sparse in both space and time due to cost-efficiency considerations, making them insufficient for comprehensive monitoring requirements (Macelloni et al., 2007; Picard et al., 2012; Champollion et al., 2013). In the absence of in situ data, firn densification models (FDMs), such as the semi-empirical IMAU-FDM (Ligtenberg et al., 2011; Veldhuijsen et al., 2023) are commonly utilised to estimate firn density and subsequent elevation changes (Schröder et al., 2019). Nonetheless, FDMs suffer from significant uncertainties (Verjans et al., 2020). For instance, the relationship between wind velocity and density, as derived by Sugiyama et al. (2012) and van den Broeke et al. (1999) exhibits notable discrepancies, introducing uncertainties when parametrising the effects of wind. Therefore, to obtain spatially and temporally continuous assessments of changes in firn densities, satellite remote sensing serves as an important complementary method (Picard et al., 2007; Brucker et al., 2014; Meredith et al., 2019). While numerous studies have investigated these assessments, they have identified intricate relationships between remote sensing observations and firn density, making it challenging to generalise remote sensing models. Consequently, a satellite-based firn density product remains elusive.

Among satellite remote sensing techniques, radiometers are a primary tool used for studying firn properties, offering various frequencies and polarisations that facilitate assessments of different firn properties at different depths (Picard et al., 2007, 2012; Champollion et al., 2013; Brucker et al., 2014; Amory et al., 2024). Radiometers measure the thermal radiation emitted by the ground surface and subsurface within the range of microwave penetration (Picard et al., 2007) and typically have a spatial resolution of $\sim 25$ km. The observed parameter is referred to as brightness temperature ($T_B$), which has been typically used to derive Antarctic surface melting extent by detecting the sharp increase in emissivity and hence $T_B$ (Picard et al., 2007; Tedesco, 2009; Nicolas et al., 2017; de Roda Husman et al., 2022). However, studies show that $T_B$ can also be used to assess firn densities. For example, Champollion et al. (2013) used the temporal variation of polarisation ratio of $T_B$ at 19 GHz and 37 GHz to evaluate the density changes of firn induced by hoar-crystal formation and disappearance at Dome C (75.06° S, 123.21° E, indicated in Fig. 2a). Alternatively, Tran et al. (2008) classified seven firn facies over Antarctica using a combination of $T_B$, a specific ratio defined by $T_B$ at 23.8 GHz and 36.5 GHz, and information from Ku- and S-band altimeters acquired in 2004. They attributed the different facies to varying surface roughness or firn grain size driven by differences in climate parameters such as wind patterns, firn accumulation, and temperature, which are known to influence firn density (Lehning et al., 2002; Champollion et al., 2013).

Alternatively, active microwave observations, specifically radar scatterometer and synthetic aperture radar (SAR) with spatial resolutions of $\sim 25$ km and up to $\sim 5$ m, respectively, have been used to assess firn properties. The backscatter intensity ($\sigma^0$) is a common parameter measured by both scatterometer and SAR. Numerous studies have been performed to link the spatial or temporal variation of $\sigma^0$ to variations of certain firn properties. Fraser et al. (2016) analysed the drivers of spatial

variation of C-band scatterometer $\sigma^0$ acquired between 2007 and 2012 in dry firn zones of Antarctica. Their study concluded that (i) the seasonal variation of $\sigma^0$ is primarily driven by precipitation and firn temperature cycles, and (ii) $\sigma^0$ exhibits a high correlation with long-term precipitation, which also affects long-term densities. On the other hand, Rizzoli et al. (2017) exploited interferometric acquisitions of X-band SAR $\sigma^0$ from TanDEM-X, using the combination of $\sigma^0$ and a volume correlation factor to classify Greenland into four firn facies with an unsupervised machine learning method. The firn facies classified by this study can be attributed to different melt extents.

The aforementioned studies indicate the capability of various passive and active satellite observations, either individually or in combination, to evaluate spatial and temporal patterns of firn density. However, the precise mechanisms underlying the impact of firn density on satellite observations cannot always be fully understood (Champollion et al., 2013; Fraser et al., 2016; Rizzoli et al., 2017). In addition, previous studies using satellite observations to assess firn properties are either restricted to a specific location where in situ measurements are available (Champollion et al., 2013) or to a specific time period (Tran et al., 2008). Generalisation of these aforementioned approaches to other areas or time periods therefore requires further assessment. Hence, it is crucial to identify suitable combinations of satellite observations and data fusion methods that enable the assessment of firn density across extensive regions and multiple seasons.

Consequently, the objective of this study is to propose and assess a methodology to derive firn density and its spatial and temporal variations over the Antarctic ice sheet based on daily satellite observations. To achieve this, we conduct a three-fold experiment involving the comparison of time series data from Special Sensor Microwave Imager/Sounder (SSMIS) and Advanced Scatterometer (ASCAT) satellites with the output of a semi-empirical firn densification model (IMAU-FDM). In the first experiment, we juxtapose the satellite time series with the output of IMAU-FDM to evaluate the potential of individual satellite parameters in linearly explaining density variations. The second experiment involves clustering analysis on the combined SSMIS and ASCAT satellite data to identify spatial and temporal patterns of satellite observations and compare them with IMAU-FDM density patterns. Finally, we assess the potential of a non-linear Random Forest (RF) machine learning approach (Breiman, 1996, 2001) trained on SSMIS and ASCAT data to derive spatial and temporal variations in dry firn density. More specifically, assuming firn densities in certain regions are known, this experiment aims to estimate firn densities of the unknown regions in space and time using a combination of satellite observations. Due to the currently limited availability of in situ density measurements, however, our study uses part of the modelled IMAU-FDM densities as "known" densities to train the RF regressor. Finally, we evaluate our RF predictions with external reference data, i.e. available in situ firn density measurements (Surface Mass Balance and Snow on Sea Ice Working Group; SUMup) and ERA5 climate parameters.

## 2 Data

In this study, we evaluate the potential of satellite microwave radiometer (SMISS) and scatterometer (ASCAT) observations in assessing the spatial and temporal dynamics of dry firn density across the Antarctic ice sheet. We focus on the grounded Antarctic ice sheet only, where wet firn and melting that potentially affect the satellite microwave observations are less pronounced (Lenaerts et al., 2016; Kingslake et al., 2017; Spergel et al., 2021; Li et al., 2021; de Roda Husman et al., 2022). To

account for this, we mask out all satellite observations over the ice shelves using the grounding line defined by Depoorter et al. (2013).

## 2.1 Radiometer data

Time series of brightness temperature ($T_B$) from the Special Sensor Microwave Imager/Sounder (SSMIS) sensors are used in this study as they are widely used to assess variations in firn properties (Tedesco and Kim, 2006; Tran et al., 2008; Brucker et al., 2010). The available measurement channels include vertically and horizontally polarised 19 GHz, 37 GHz and 91.655 GHz, and vertically polarised 22 GHz (Kunkee et al., 2008). However, for the purposes of this study, our focus is solely on the 19 GHz and 37 GHz channels, since the atmospheric influence is negligible at these frequencies (Picard et al., 2009; Brucker et al., 2011; Champollion et al., 2013). Theoretically, the penetration depths are 1–7 m (at 19 GHz) and 0.1–2 m (at 37 GHz) in dry snow zones of Antarctica (Surdyk, 2002; Brucker et al., 2010). With the presence of liquid water, the imaginary part of snow permittivity increases, therefore $T_B$ increases (Tedesco, 2007). However, the actual penetration depths can still vary per region (Picard et al., 2009). These characteristics ensure the possibility for SSMIS at 19 GHz and 37 GHz to monitor the changes of firn properties at a variety of depths. The daily polar-gridded $T_B$ data are acquired from the National Snow and Ice Data Center (NSIDC) with a spatial resolution of 25 km for both the 19 GHz and 37 GHz channels (Meier et al., 2021). All data are acquired by the F17 sensor as it provides continuous daily data acquisition in the period between Jan. 1, 2011 and Dec. 31, 2020.

## 2.2 Scatterometer data

Backscatter intensity ($\sigma^0$) from synthetic aperture radar (SAR) was also previously used to assess density variations due to the melting–refreezing process of certain firn types (Rizzoli et al., 2017) and to examine variations in firn facies (Fahnestock et al., 1993). In this study, we employ time series of backscatter intensity from the Advanced Scatterometer (ASCAT) satellite sensor as an alternative to SAR $\sigma^0$, primarily due to its high temporal resolution (daily) and its coverage over the entire Antarctica. ASCAT is an operational C-band (5.255 GHz) fan-beam scatterometer (Figa-Saldaña et al., 2002; Fraser et al., 2016) that has been in operation on Metop satellites since 2006. It operates in V polarisation and covers multiple incidence angles. For dry firn, penetration depth of C-band ASCAT is approximately 20 m (Rignot, 2002). Following Larue et al. (2021), we also performed a simulation using the Snow Microwave Radiative Transfer (SMRT) model (Picard et al., 2018), where firn properties at different depths of the firn layer are altered, and the impact on both backscattering and brightness temperature are presented (Appendix A). However, the top 1 m is most exposed to atmospheric drivers, which also affect the variability of C-band microwave (Fraser et al., 2016). The ASCAT products used in this study are obtained from Brigham Young University (BYU) Microwave Earth Remote Sensing (MERS) laboratory (2010) (Long et al., 1993; Early and Long, 2001; Lindsley and Long, 2010). The data are processed using the scatterometer image reconstruction (SIR) algorithm, which enhances the spatial resolution of images from 25 km to 4.45 km. The backscattering product adopted in our study is referred to as the $A$ product

in Long and Drinkwater (2000):

$$\sigma^0(\theta) = A + B(\theta - 40°) \tag{1}$$

where $A$ (in dB) is the originally measured $\sigma^0$ normalised to $40°$, and $B$ (in $dB/°$) is a parameter describing the dependence of the original $\sigma^0$ on $\theta$. The processing of Long and Drinkwater (2000) accounts for the incidence angle dependence of the originally measured $\sigma^0$, as the measurements are made over multiple incidence angles (between $20°$ and $55°$). In this study, we only use the isotropic, normalised $A$ parameter (hereafter $\sigma_A^0$) as it has been shown to better correlate with various climate parameters as well as the long-term firn density (Fraser et al., 2016). In addition, the presence of liquid water can reduce the volume scattering and increase the microwave absorption (Stiles and Ulaby, 1980); this should be taken care of and will be elaborated in Sect. 3. To ensure consistent analysis between $T_B$ and $\sigma_A^0$, the BYU $\sigma_A^0$ products are interpolated to the same polar grids as the SSMIS $T_B$ products using bi-linear interpolation. The data acquisition time is the same as that of the radiometer data.

## 2.3 Densities from Firn Densification Model

To understand the spatio-temporal variation in satellite data, we compare the SMISS and ASCAT satellite data to the output of a semi-empirical firn densification model. Therefore, we use output from the latest version of the IMAU Firn Densification Model (IMAU-FDM v1.2A; Veldhuijsen et al., 2023). IMAU-FDM simulates the transient evolution of the Antarctic firn column, and is forced at the upper boundary by outputs of the Regional Atmospheric Climate Model (RACMO2.3p2) at a 27 km horizontal resolution (van Wessem et al., 2018) and with a temporal resolution of 10 days. The model employs up to 300 layers in total of 3 to 15 cm thickness, which represent the firn properties in a Lagrangian way. The output is resampled to a regular grid with layers of 4 cm. The density of the freshly fallen snow is a function of instantaneous wind speed and temperature in IMAU-FDM. Over time, the simulated firn layers become denser due to dry-snow densification and meltwater refreezing.

To estimate at which depth the firn density has impact on satellite microwave, we perform a correlation estimation between satellite observation time series and IMAU-FDM density at different depths, as elaborated in Section 3.1. The unrealistically large values in IMAU-FDM densities (more than $917\ kg\ m^{-3}$) are treated as invalid. To facilitate comparison with the satellite products, the firn density data from IMAU-FDM are re-projected using bi-linear interpolation to the same polar grids as the satellite data, where valid data are restricted to pixels within the Antarctica coastline provided by Depoorter et al. (2013).

## 2.4 Reference in situ density measurements

Furthermore, we employ in situ density measurements obtained from the SUMup dataset (Koenig and Montgomery, 2018; Montgomery et al., 2018) as a reference for spatial evaluation of the satellite data and the RF regressor. SUMup provides information on start-point, end-point and mid-point of measurements. We use the mid-point here to define the depth of the reference data. For each date of measurement at each location, if multiple measurements are available, only the density measurements at the shallowest mid-point depths are used. Such depths are also restricted to $< 1$ m. The measurements within the

depth restriction were taken between Jan. 22, 1984 and Jan. 23, 2017, and consist of 67 valid points. The SUMup dataset does not contain time series, but only single measurements on specific irregular dates throughout the time period between 1984 and 2017. Therefore, we use the SUMup dataset only for spatial evaluation of the potential uncertainties from both the IMAU-FDM densities and the densities estimated by the RF regressor.

## 2.5 ERA5 climate parameters

As mentioned in Sect. 1, IMAU-FDM can introduce discrepancies due to simplified parametrisation (Verjans et al., 2020), which can be propagated in the estimation process with the RF regressor. Therefore, to interpret the difference between the measured (SUMup or Leduc-Leballeur et al. (2017) data), modelled (IMAU-FDM) and estimated (RF) densities, it is important to understand the effects of climate conditions. Therefore, we use ERA5 wind speed estimated at midday (Muñoz-Sabater, 2019; Muñoz-Sabater et al., 2021) as an approximation of the daily wind conditions. By incorporating this information, we aim to better understand the discrepancies between the observed and IMAU-FDM densities, as well as the source of discrepancies between the IMAU-FDM densities and the densities estimated from satellite observations with the RF regressor. The ERA5 wind speed data have a horizontal resolution of 9 km. Similarly to the IMAU-FDM data, we interpolate these climate variables to the same polar grids as the SSMIS data using bi-linear interpolation to ensure consistency in the analysis.

## 3 Method

We assess the potential of SSMIS and ASCAT satellite observations to assess dry firn density in a three-fold experiment. First, we compare the satellite time series with the output of IMAU-FDM to evaluate the potential of individual satellite parameters to linearly explain density variations (Sect. 3.1). Second, we perform a clustering analysis on the combined SSMIS and ASCAT observations to identify spatio-temporal patterns of satellite observations. These patterns are then compared with the density patterns obtained from IMAU-FDM, and dry-snow zones are determined (Sect. 3.2). Finally, we quantify the potential of a non-linear Random Forest (RF) machine learning approach trained on SSMIS and ASCAT observations to derive the spatial and temporal variations in dry firn density (Sect. 3.3). For clarity, the content of Sect. 3.2 and Sect. 3.3 are summarised and visualised as a flowchart in Fig. 1.

### 3.1 Calculation of correlation between satellite parameters and firn density

To gain a general understanding of the spatial patterns of the satellite parameters and densities from IMAU-FDM, we calculate and visualise the map of $T_B$ and $\sigma_A^0$ and the IMAU-FDM firn density at a selected depth averaged between Jan. 1 2011 and Dec. 31 2020 (shown in Appendix B). Then, to observe the temporal correlation between the satellite parameters and the IMAU-FDM densities, for each pixel, the correlation coefficient between different satellite parameters and the firn density over time is calculated and visualised. To ensure consistent temporal resolution for the analysis, the satellite parameters are downsampled from daily resolution to 10-day resolution to match the temporal resolution of the IMAU-FDM densities. Since the scattering properties of microwave are affected by firn properties along the penetration depth (Ulaby et al., 1996; Bingham

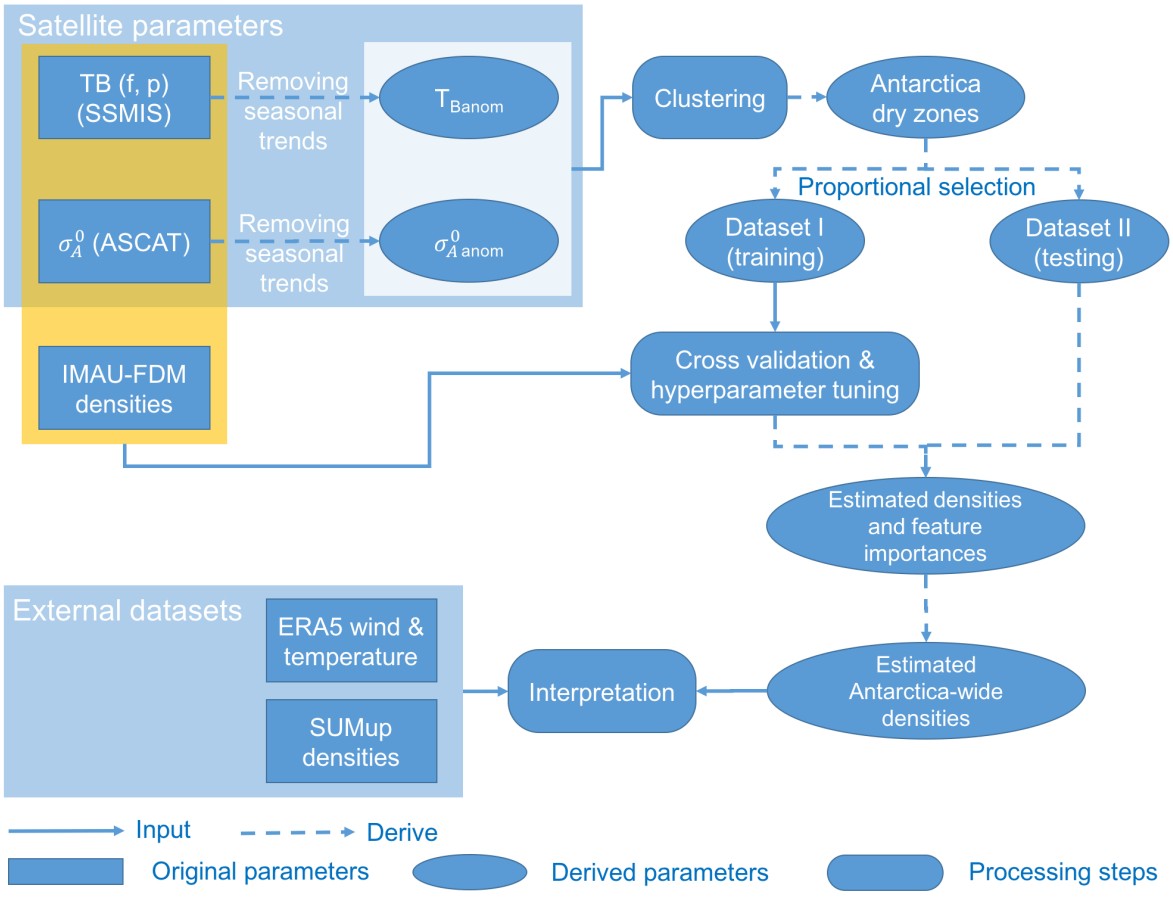

**Figure 1.** Overview flowchart of the data and method used in this study. The clustering process uses $T_{Banom}$ and $\sigma^0_{Aanom}$ as input to derive dry snow zones over the Antarctic ice sheet. Then, pixels clustered as dry snow are included to estimate firn density with the RF regressor. Parameters used as features of the RF regressor are further elaborated in Sect. 3.3. Among the derived parameters, Antarctica dry zones, Dataset I and Dataset II are selected proportionally based on the number of pixels per cluster.

and Drinkwater, 2000; Arndt and Haas, 2019; Cartwright et al., 2022), this analysis utilises densities from a range of depths, including 12 cm, 40 cm, 1 m, 2 m, 5 m and 10 m. The density of each depth is defined not as the specific density at the single depth, but the average density from the surface to this depth. The reason for this comparison is that, although the theoretical penetration depth can be larger than 20 m for C-band in Antarctic dry firn (Rott et al., 1993), the surface conditions such as temperature, wind and precipitation have more impact on shallow depth of the firn layer, as well as on the satellite parameters (Tran et al., 2008; Picard et al., 2012; Champollion et al., 2013; Fraser et al., 2016). By calculating the correlation coefficients between IMAU-FDM densities and satellite parameters, we need to understand at which depth the densities cannot be affected by the surface conditions. We also need to estimate a depth threshold from which 37 GHz cannot penetrate the firn layers hence cannot provide information on spatial and temporal variation of firn from this experiment, as the penetration ability reduces with an increasing frequency (Rott et al., 1993; Surdyk, 2002). Finally, the density at the depth where the best overall correlation between satellite observations and density time series is adopted for the RF experiment.

### 3.2 Characterisation of firn types using time series of microwave observations

In our study, the clustering of satellite observations is primarily carried out as a preparatory step aiming at ensuring that all the representative regions, i.e. the regions with distinctive satellite data patterns, are correctly accounted for in the RF model training procedure in Sect. 3.3. Moreover, we aim to rule out pixels where melt events can be observed, as the melt-induced liquid water and ice-lens formation complicates the satellite measurements (Stiles and Ulaby, 1980; Brucker et al., 2010; Trusel et al., 2012), rendering density estimations invalid in such cases. This step facilitates a comprehensive understanding of the spatio-temporal variations of firn properties based on the available satellite observations. We expect that clustering the time series of satellite observations will effectively differentiate pixels experiencing melting from those unaffected. By identifying and excluding melt-affected pixels, we can ensure the validity of density estimations using the RF regressor described in Sect. 3.3. Additionally, to enhance the ability of the RF regressor to capture the characteristics of various dry snow types, we choose training samples based on the identified dry snow types. This approach enables the representation of diverse snow types in the training dataset, improving the accuracy of the RF regressor in estimating density across different snow types.

To cluster and distinguish the different snow types, we propose to use the anomalies in $T_B$ and $\sigma_A^0$ described as follows. Since $T_B$ is strongly dependent on seasonal variations of firn temperature, the average seasonal signal is removed in the clustering process to obtain time series anomalies that reflect the variations of temporary events such as melt–refreeze (Nicolas et al., 2017) and density or grain size variations (Picard et al., 2012; Champollion et al., 2013). We also derive the $\sigma_A^0$ anomalies due to the impact from temperature seasonal cycles (Fraser et al., 2016). The time series anomalies are calculated by taking the ten-year average of $T_B$ or $\sigma_A^0$ for each day in a year, defined as $\overline{T}_B$ and $\overline{\sigma}_A^0$, and subtracting this averaged time series from the absolute observations for each year, leading to $T_{Banom} = T_B - \overline{T}_B$ and $\sigma_{Aanom}^0 = \sigma_A^0 - \overline{\sigma}_A^0$. The time series anomalies of $T_{Banom}$ and $\sigma_{Aanom}^0$ are then normalised and stacked for clustering.

The adopted clustering solution is a simple hierarchical algorithm (Ward, 1963) which uses the normalised and stacked $T_{Banom}$ and $\sigma_{Aanom}^0$ time series as input. For pre-processing, we remove outliers in the $T_{Banom}$ and $\sigma_{Aanom}^0$ time series per pixel by defining an interval of three standard deviations above and below average. Then, the temporal gaps are filled

with a linear interpolation. The application of the clustering algorithm is illustrated with an example (Fig. 2). The clustering process starts from all clusters each containing one pixel, and the clusters are then hierarchically grouped together based on the similarity of features, which refers to the euclidean distance between the normalised and stacked $T_{Banom}$ and $\sigma^0_{Aanom}$ time series of different pixels in our study (however only $\sigma^0_{Aanom}$ from January 14, 2016 is used in Fig. 2 for illustration). The grouping process is typically represented by a dendrogram, as in Fig. 2b. Finally, the number of clusters is determined empirically; different numbers of clusters result in different outcomes, as in Fig. 2c–e. For our study where the normalised and stacked $T_{Banom}$ and $\sigma^0_{Aanom}$ time series between 2011 and 2020 are used, we select 7 clusters as the optimal number of clusters. To provide a brief overview of the clustering result, we visualise the time series of the mean, 20th percentile, and 80th percentile of different satellite parameters, together with an IMAU-FDM density for each cluster in Appendix C. This allows a comparison of the changes in satellite parameters with density variations across the clusters and an assessment of the reliability of our study to distinguish melt zones from dry ones.

## 3.3   Deriving firn densities using satellite parameters and random forest regressor

Given the complex and often non-linear relationships between satellite observations and firn density (Fraser et al., 2016), a non-linear regression model based on machine learning is explored to relate the satellite time series to firn density. The method relies on a certain amount of known density measurements as the training dataset, and on the continuous satellite parameters as the trained features. We opt for a random forest regressor as machine learning model (RF regressor hereafter) due to the simplicity and usability (Vafakhah et al., 2022; Viallon-Galinier et al., 2023).

Ideally, in situ measurements should be used as the training dataset. However, in situ measurements are often single measurements that lack temporally continuous observations. As our goal is to relate the satellite time series to assess spatio-temporal variations in firn density, we adopt an alternative approach that uses the output of IMAU-FDM as training data instead of relying on in situ data. Although this approach has the disadvantage of training the RF regressor on a noisy IMAU-FDM dataset, which may exhibit spatial and temporal differences compared to actual in situ densities (e.g., biases between the model and in situ observations), we leverage the strengths of RF regression for pattern recognition in noisy datasets. The use of multiple decision trees and random feature selection can reduce the variance of the model and reduce overfitting, resulting in better generalisation performance on noisy data (Hastie et al., 2008). Therefore, we expect that the RF regressor generalises on the density estimations of IMAU-FDM, which is known to capture the spatial variation of in situ density measurements well and the temporal variations reasonably well (Veldhuijsen et al., 2023).

The training, testing, and implementation of the RF regressor involve three main steps:

–   Training and Hyperparameter Tuning: a subset of IMAU-FDM densities (Subset I) is used as the training dataset in a 5-fold cross-validation procedure. Multiple models are evaluated, representing different combinations of hyperparameters defined for the RF regressor (see Table 1). The goal is to identify the configuration that achieves the best cross-validation score, indicating the optimal set of hyperparameters for the RF regressor.

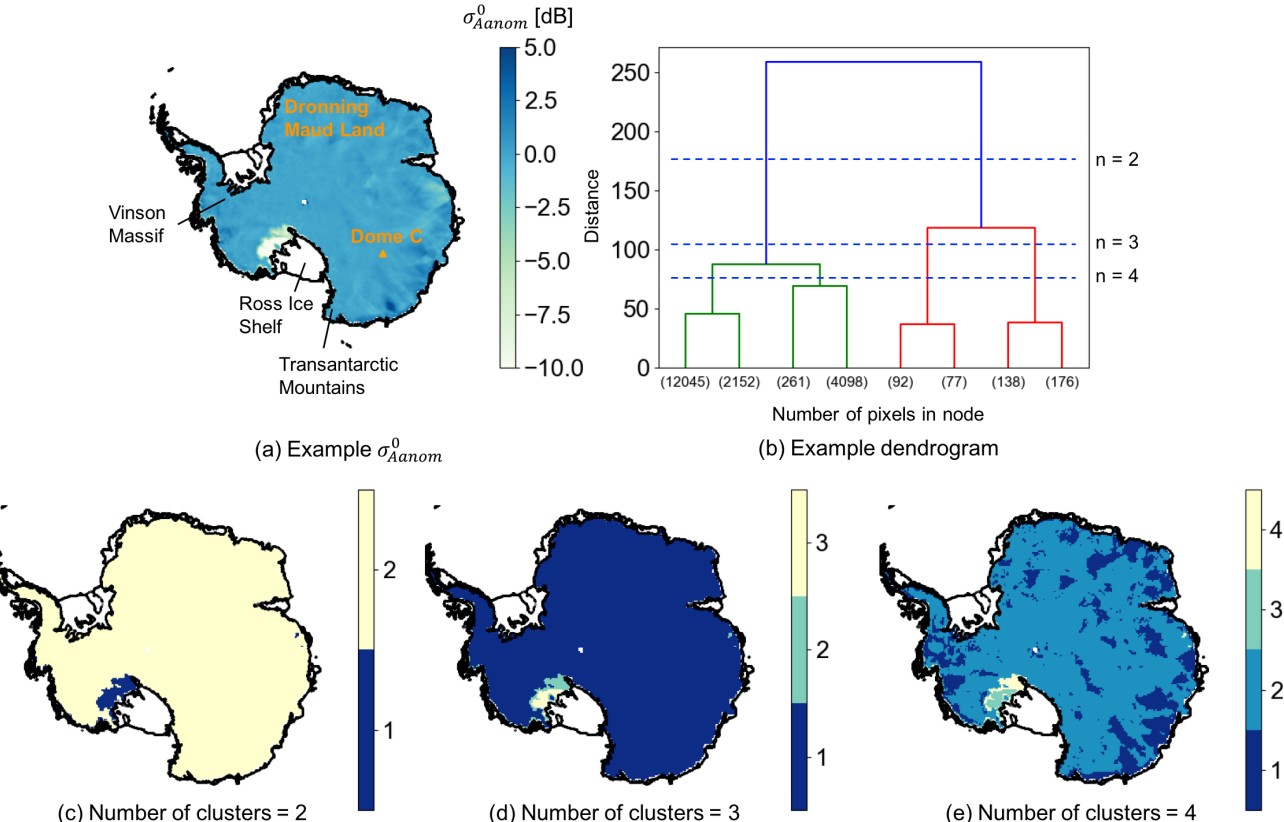

**Figure 2.** An example of the principle of hierarchical clustering. (a) Map of $\sigma^0_{Aanom}$ acquired on January 14, 2016 following the melt event detected by Nicolas et al. (2017), (b) dendrogram obtained from (a), with low-hierarchy nodes simplified and $n$ referring to number of clusters, and (c)–(e) clustering results using different numbers of clusters. Several locations mentioned in this study are labelled in (a). The coastline is from Depoorter et al. (2013).

– Testing and Model Evaluation: a different subset of temporally and spatially coregistered SSMIS and ASCAT measurements for the given pixels (Subset II) is used as input to the RF regressor, which has been trained on Subset I. The purpose of this step is to evaluate the performance of the model and assess the accuracy of the RF density estimations. Additionally, it helps to determine the importance of satellite parameters in the predictions of the regressor.

– Antarctica-wide Implementation: The satellite time series covering the entire study area are fed into the RF regressor, which has been trained on Subset I. This step aims to estimate densities across the entire Antarctic dry-firn region. The output densities are then evaluated by comparing them to both the IMAU-FDM densities and the SUMup densities.

Both Subset I and Subset II consists of pixels randomly selected from the non-melting pixels clustered in Section 3.2. Subset I contains 10 % of the non-melting pixels, and Subset II contains 100 pixels in total. The pixels from both subsets should not

**Table 1.** Hyperparameter range and optimal values used to specify the random forest (RF) model.

| Hyperparameter | Range | Optimal value |
|---|---|---|
| Number of trees | 50, 100, 200 | 100 |
| Maximum depth of the tree | 12, 15, 18 | 12 |
| The minimum number of samples at a leaf node | 1, 3, 5, 7 | 5 |
| The minimum number of samples to split an internal node | 2, 3, 4, 5 | 4 |
| The number of features to consider when searching for the best split | 1, 3, 5 | 1 |

overlap. The time series of each feature in each pixel cover the period between January 1 2011, and December 31 2020 with a 10-day resolution. To ensure consistent temporal resolution between the input features and the target IMAU-FDM densities, the daily satellite parameters are also downsampled to the 10-day temporal resolution of the IMAU-FDM firn density by selecting the corresponding acquisition date, resulting in 366 samples in total for each feature in each pixel. Finally, Subset I consists of 1,748 pixels multiplied by 366 samples (639,768 samples in total), Subset II consists of 100 pixels multiplied by 366 samples (36,600 samples in total), and the Antarctica-wide dataset consists of 17,478 pixels multiplied by 366 samples (6,396,948 samples in total).

The RF regressor is implemented with the target variable, which is the IMAU-FDM density at the depth selected from the correlation analysis, and input features $\boldsymbol{X}$ initially defined as follows:

$$\boldsymbol{X} = (T_B(19V),\ T_B(19H),\ T_B(37V),\ T_B(37H),\ \sigma_A^0 \tag{2}$$

Within $\boldsymbol{X}$, we include $T_B$ and $\sigma_A^0$ to account for variations in temperature, precipitation and other potential climate parameters that show a potential strong seasonality (e.g., Fraser et al., 2016).

In the testing and evaluation step, we assess the performance of the optimal RF regressor. This is achieved by comparing the RF and IMAU-FDM densities of Subset II using a scatterplot and standard evaluation metrics, i.e. the root mean square error (RMSE) and the correlation coefficient between the RF densities and the IMAU-FDM densities. The importance of satellite parameters in the RF regressor is computed by calculating the Gini importance and the permutation importance. Gini importance in RF regression is a measure of feature importance based on the Gini gain, i.e. impurity reduction (Strobl et al., 2007). For each feature used to split the data, the decrease in the Gini node impurity is recorded at each split, and the Gini importance is calculated as the average of all decreases in the Gini impurity in the forest where this feature forms the split (Archer and Kimes, 2008).

In the Antarctic-wide implementation, the optimal RF regressor is implemented to predict the spatial and temporal variations in firn density. These predictions are then compared with IMAU-FDM and the SUMup densities. The spatial agreement is assessed by comparing the temporal averages of the RF predictions, IMAU-FDM and SUMup by using the mean difference and the RMSEs. The temporal agreement is assessed by the RMSE and the correlation coefficient between the per-pixel time series of RF predictions and IMAU-FDM density. We also compare the spatial patterns of the RF-predicted densities with the ERA5 wind velocity as it is a potential driver for spatial variation in firn density, especially for the uncertainties of IMAU-

**Table 2.** Average temporal correlation coefficient between satellite parameters and IMAU-FDM density from different depths.

| Depth | $T_B(19V)$ | $T_B(19H)$ | $T_B(37V)$ | $T_B(37H)$ | $\sigma_A^0$ |
|---|---|---|---|---|---|
| 12 cm | 0.19 | 0.18 | 0.20 | 0.20 | -0.05 |
| 40 cm | 0.24 | 0.23 | 0.20 | 0.19 | -0.06 |
| 1 m | 0.23 | 0.20 | 0.12 | 0.12 | -0.06 |
| 2 m | 0.18 | 0.12 | 0.03 | 0.02 | -0.06 |
| 5 m | 0.08 | 0.02 | -0.07 | -0.08 | -0.04 |
| 10 m | 0.05 | 0.01 | -0.07 | -0.07 | -0.03 |

FDM. Finally, we illustrate this temporal agreement by showing time series over four pixels that show representative differences between RF and IMAU-FDM densities (locations visualised in Fig. 4).

In addition, since satellite parameters may exhibit a certain level of correlation with densities in the long term (Fraser et al., 2016), we also conduct a linear regression (LR) process, which fits a linear function between $X$ and the target density. The RMSE and correlation coefficient between the LR-obtained density and IMAU-FDM density are also used to assess the advantages and drawbacks of RF.

## 4    Results

**4.1    Correlation between satellite parameters and firn density**

The temporal correlation between satellite parameters and the average density from the upper $x$ m depth ($x$ refers to 12 cm, 40 cm, 1 m, 2 m, 5 m and 10 m, respectively) is calculated per pixel, and the spatial average of the correlation coefficient is summarised in Table 2. The results show that on average, the maximum absolute correlation coefficient can be obtained at 40 cm depth. The correlation between density and $T_B$ at 19 GHz frequency drastically decreases at 5 m, and between density

and $T_B$ at 37 GHz frequency largely decreases at 2 m, similar to the penetration ability from Surdyk (2002). The correlation between densities and $\sigma_A^0$ is constantly negative, and the absolute correlation coefficient is constantly low; however, it also demonstrates a slight decrease as the depth increases from 2 m to 10 m, showing a certain degree of sensitivity. Despite the low correlation, however, our study still includes $\sigma_A^0$ due to the long-term correlation derived by Fraser et al. (2016).

The lack of spatial and temporal consistency between satellite and density is illustrated in Fig. 3, which shows the pixel-wise

temporal correlation of each satellite parameter with the 40 cm density in IMAU-FDM. All $T_B$ channels generally show a positive correlation with $\rho_{40cm}$ in East Antarctica, but a negative correlation in parts of West Antarctica and many coastal regions. The negative correlation in coastal regions can be attributed to melt, as shown in the masked out regions in Fig. 4 of Picard et al. (2012). The correlation between $\rho_{40cm}$ and $\sigma^0$ is generally low, except for the region next to the Ross Ice Shelf (location shown in Fig. 2a, where the correlation coefficient can be up to 0.75.

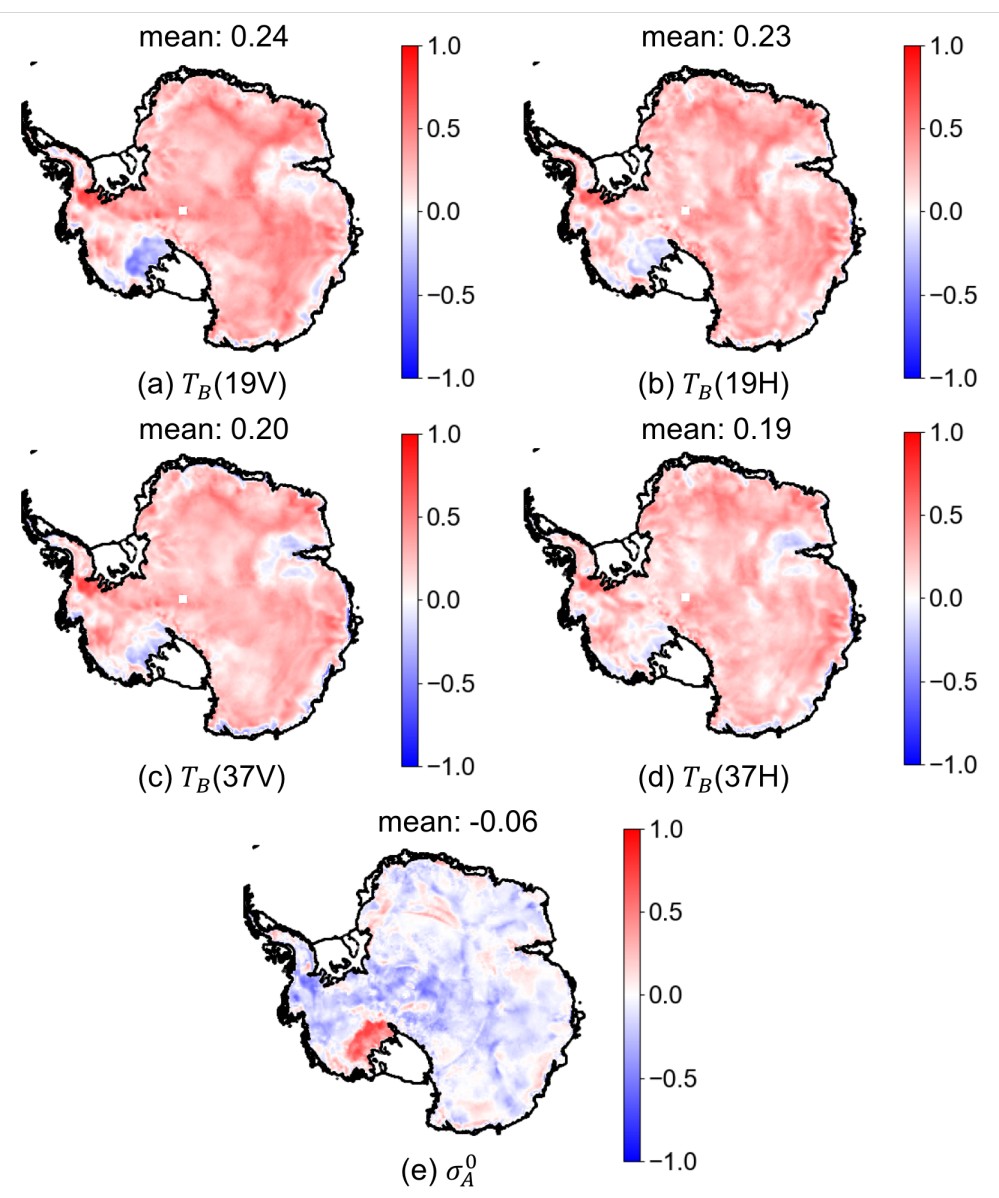

**Figure 3.** Map of temporal correlation calculated per pixel between 40 cm IMAU-FDM density and (a) brightness temperature ($T_B$) from 19 GHz vertical polarisation, (b) $T_B$ from 19 GHz horizontal polarisation, (c) $T_B$ from 37 GHz vertical polarisation, (d) $T_B$ from 37 GHz horizontal polarisation, and (e) backscatter intensity ($\sigma_A^0$). The coastline is from Depoorter et al. (2013).

Overall, this correlation analysis indicates that the relationship between satellite parameters and firn density is complex, and simple linear relationships may not adequately describe the IMAU-FDM density based on different satellite parameters. Therefore, non-linear approaches such as the RF regressor should be employed to assess the potential of relating the IMAU-FDM firn density to various satellite parameters (Vafakhah et al., 2022; Anilkumar et al., 2023).

## 4.2 Firn-type clusters

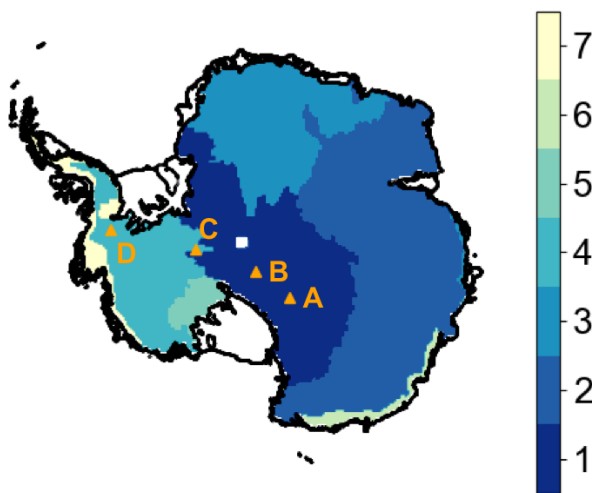

**Figure 4.** Clustering results from the combination of normalised $T_B$ and $\sigma_A^0$ after removing the seasonal trend. Triangles show the locations where temporal assessment per pixel is performed. The coastline is from Depoorter et al. (2013).

Figure 4 shows the map of clusters derived from time series of the combined satellite parameters, where each cluster represents a natural grouping of pixels with similar satellite time series behaviour. The map shows that four large clusters (referred to as Firn 1–4) cover the dry firn interior of Antarctica. Firn 1–3 in East-Antarctica and Firn 4 in West-Antarctica. Firn 5 is a cluster in West Antarctica close to Ross Sea which corresponds to the region that showed a strong melt event in Jan. 2016 (Nicolas et al., 2017) while Firn 6 and Firn 7 show small regions near the coastline in East- and West-Antarctica respectively 320   that also show clear melting signals (details shown in Appendix C).

## 4.3 Assessment of RF densities at sample pixels

Figure 5a presents the results of the RF regressor for estimating firn densities based on satellite parameters. It demonstrates that the non-linear multivariate approach of the RF regressor captures the spatial variations in IMAU-FDM density, exhibiting a linear relationship between IMAU-FDM and RF densities with a slope of 0.86. The RMSE is $19.23\ kg\ m^{-3}$ and the 325   correlation coefficient between the estimated and training densities is 0.67. Moreover, the RF regressor performs most ideally

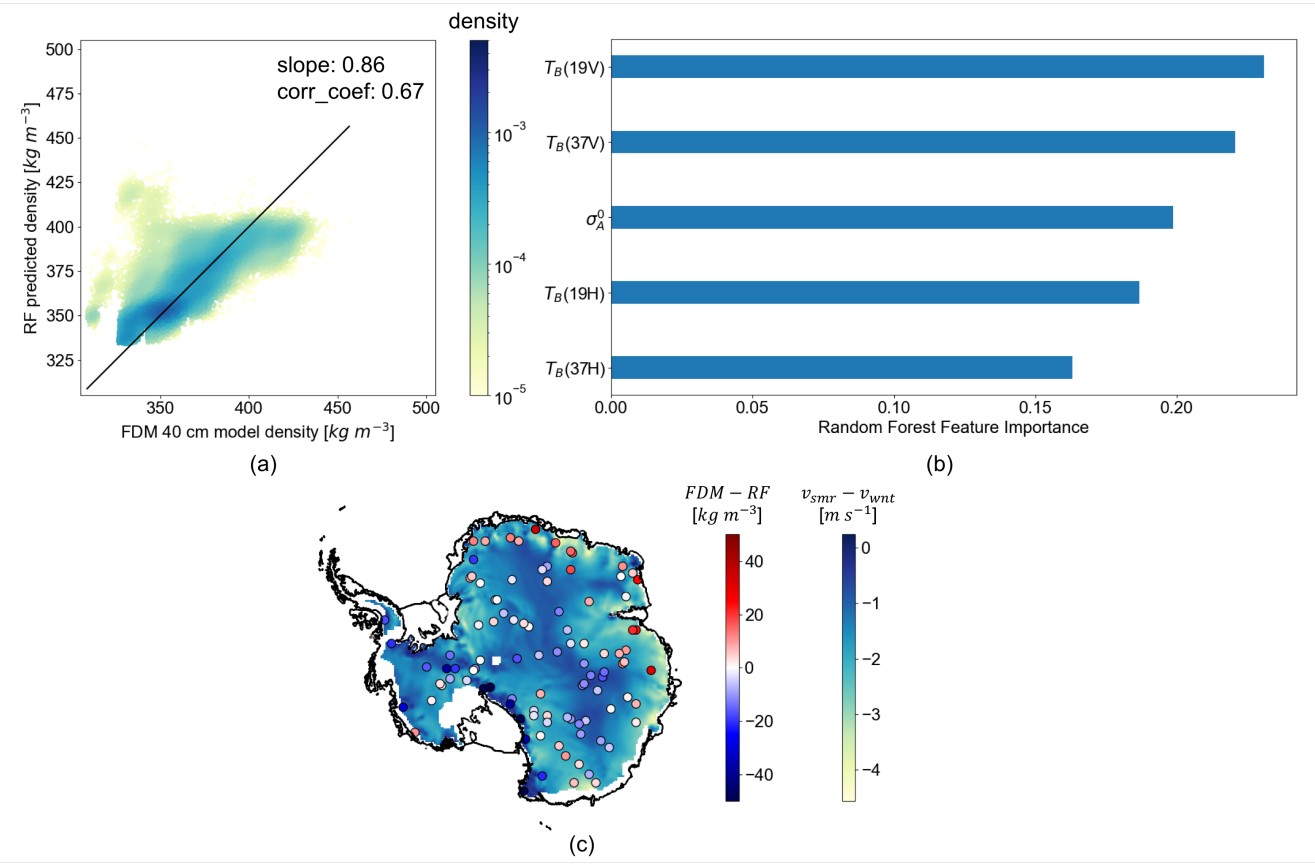

**Figure 5.** (a) Density comparison between RF densities and IMAU-FDM densities at sample pixels referred to as Subset II, the colour of the points showing the density distribution of points; the colour bar is in logarithmic scale, (b) RF feature importance of different input satellite parameters, and (c) the temporally averaged difference between IMAU-FDM and RF densities at the pixels, visualised on top of the map of the difference between the summer ($v_{smr}$) and winter wind velocity ($v_{wnt}$) from ERA5. The coastline is from Depoorter et al. (2013).

between approximately $325\ kg\ m^{-3}$ and $375\ kg\ m^{-3}$, whereas it fails to capture the large densities as no RF estimate exceeds $410\ kg\ m^{-3}$, which can partially be due to a well-known extrapolation problem intrinsic to the RF regression (Hengl et al., 2018). The RF densities also exhibit an overestimation when the IMAU-FDM density is lower than $325\ kg\ m^{-3}$. The pixels with large overall underestimation (in dark red) and overestimation (in dark blue) of RF is also visible in Fig. 5c. In general, the large underestimation of RF occurs in the coastal regions of East Antarctica, where the winter wind velocity largely exceeds the summer wind velocity (by approximately $3\ m\ s^{-1}$). The large overestimation of RF occurs along the Transantarctic Mountains, where the topography is more complex, introducing strong surface scattering instead of volume scattering. The feature importance provided by Gini impurity index (Fig. 5b) shows the ranked importance of satellite parameters in the predictive performance of the model, indicating that the vertical polarisation of $T_B$ is dominant in predicting $\rho_{40cm}$. The higher importance of 19 GHz is also clearly visible in the temporal correlation coefficients in Fig. 3. We attribute the high importance

of $\sigma^0$ to the fact that it can be influenced by other parameters that have an impact on dry-snow scattering properties, such as wind and precipitation; the mechanism may not necessarily be linear, but rather complex (Fraser et al., 2016).

## 4.4 Spatial assessment of RF densities

In Fig 6, the temporally averaged RF density estimates and their differences relative to IMAU-FDM densities at the $40$ cm depth and SUMup in situ densities are presented. The comparison in Fig. 6c shows that temporally averaged RF density estimations are in general larger than temporally averaged IMAU-FDM density in interior regions of Antarctica except for megadune regions, whereas they are lower towards coastal regions. The RMSE between the IMAU-FDM and RF averages (referred to as FDM-RF) is $17.30\ kg\ m^{-3}$ and the mean FDM-RF difference is $-0.40\ kg\ m^{-3}$. An overestimation of RF is most pronounced in West Antarctica close to Vinson Massif (location shown in Fig. 2a), which possibly corresponds to the overestimation in Fig. 5a. Meanwhile, the comparison with the SUMup densities shows that RF and IMAU-FDM densities have comparable error patterns. The RMSE of FDM-SUMup is $59.17\ kg\ m^{-3}$, and the mean of FDM-SUMup bias is $23.92\ kg\ m^{-3}$; the RMSE of RF-SUMup is $62.22\ kg\ m^{-3}$, and the mean of RF-SUMup is $26.46\ kg\ m^{-3}$. This shows a general overestimation and a large bias of both IMAU-FDM and the RF models when validated with the SUMup measurements. In Fig. 6d, it can be observed that neither IMAU-FDM nor RF manages to follow the large SUMup dynamics. This difference between models and in situ measurements can be attributed to the temporal discrepancies between the measurements and the IMAU-FDM and satellite observations, and the IMAU-FDM model errors or uncertainties that can also be learned by the RF regressor.

Aided by Fig. 7, we then analyse the temporal distribution of the offsets between the IMAU-FDM densities and the RF densities in more depth. Figure 7a generally shows low RMSE between IMAU-FDM and RF densities in high-elevation regions of East Antarctica and part of West Antarctica. The errors increase towards the coastal regions. The low correlation coefficients in Fig. 7b indicate a low temporal agreement between IMAU-FDM and RF densities. Furthermore, the correlation coefficients are generally positive; high correlation coefficients ($\geq 0.5$) can mainly be observed in high-elevation regions of East Antarctica (except for megadune regions Fahnestock et al., 2000) and a part of West Antarctic Peninsula. The regions with high correlation coefficients also mainly correspond to regions with high correlation coefficients ($\geq 0.5$) in Fig. 3a, with parts of West Antarctica as an exception, which generally matches the observation in Fig. 5 where $T_B(19V)$ has the highest importance. The temporal mismatch and low correlation between IMAU-FDM and RF may be in part due to the modelling errors of IMAU-FDM. The density changes that are not modelled by the IMAU-FDM, but affect the satellite observations, are expected to degrade the quality of the RF regressor. The satellite data might be affected by other climate parameters that are not included in the IMAU-FDM model. The comparison with LR density shows that RF largely outperforms LR in terms of RMSE, especially in the interior of the ice sheet. While the average correlation coefficient is comparable between RF and LR, RF outperforms LR in high-elevation regions of East Antarctica, and performs worse in the megadune regions. By assessing the temporal agreement (mainly correlation coefficients) with ERA5 wind velocity (Fig. 7d and e), we can learn that a high temporal correspondence is spatially correlated with a small wind velocity difference ($>-2.5\ m\ s^{-1}$) between Antarctic summer (Oct.–Mar.) and winter (Apr.–Sept.). However, despite the small wind velocity difference and a relatively high temporal correspondence, the RMSE between IMAU-FDM and RF is high, in regions close to Vinson Massif and along Transantarctic Mountains (locations shown

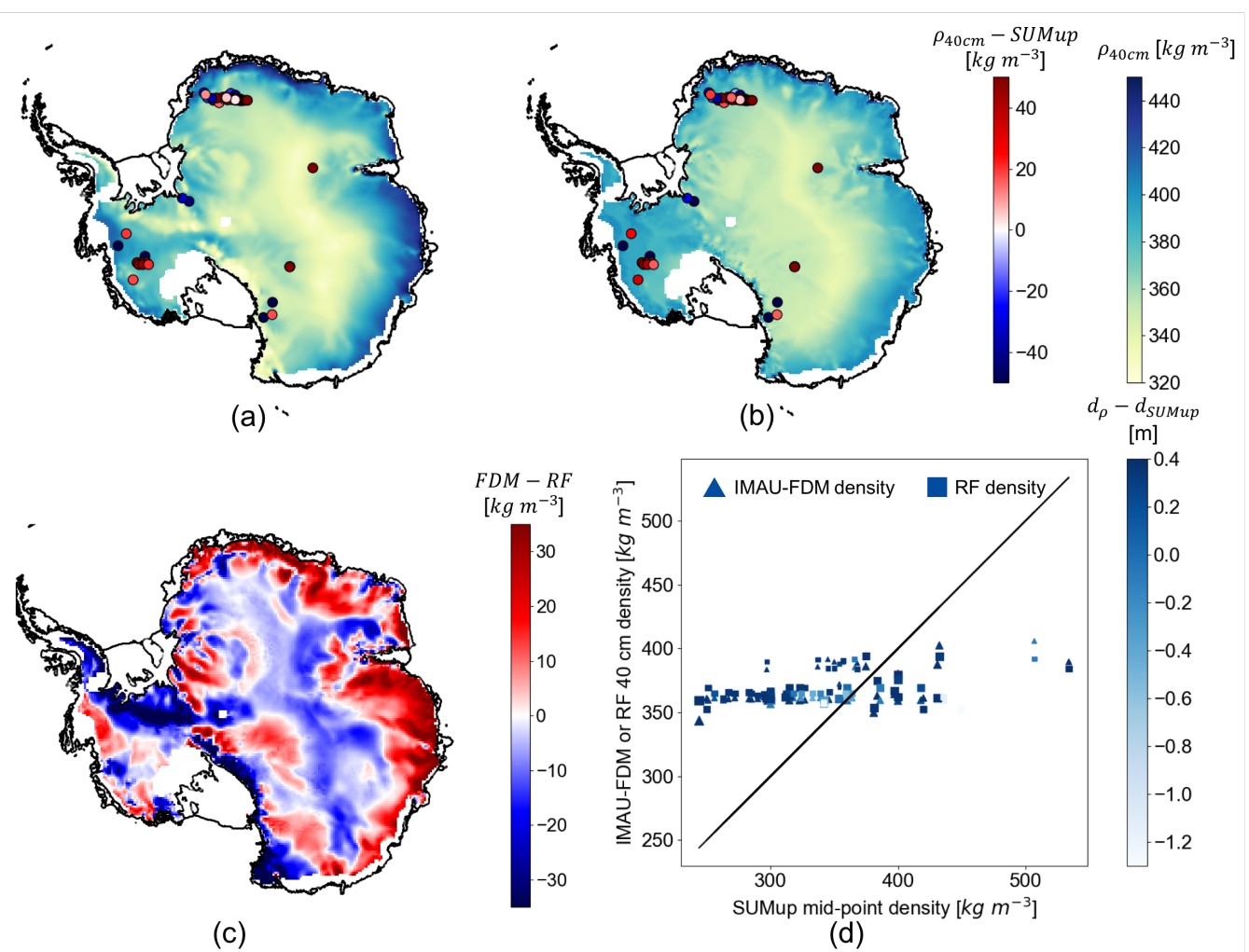

**Figure 6.** (a)–(c) Maps of (a) temporally averaged IMAU-FDM 40 cm densities, (b) temporally averaged RF densities, (c) difference between averaged IMAU-FDM densities and RF densities ($FDM - RF$). Difference between the modelled or estimated densities and the SUMup densities are shown in scattered points in (a) and (b) as FDM-SUMup or RF-SUMup. (a) and (b) share the same colour bar, in which blue–red shows the difference between the IMAU-FDM or RF densities and the SUMup densities ($\rho_{40cm} - SUMup$), and green–light blue shows the IMAU-FDM or RF densities ($\rho_{40cm}$). The coastline is from Depoorter et al. (2013). (d) shows the relationship between IMAU-FDM or RF densities and SUMup densities. The sizes of the scattered points indicate the time difference between the SUMup measurements and year 2020, and the colour shows the difference in depth between IMAU-FDM or RF measurements (both fixed at 40 cm) and SUMup measurements ($d_\rho - d_{SUMup}$).

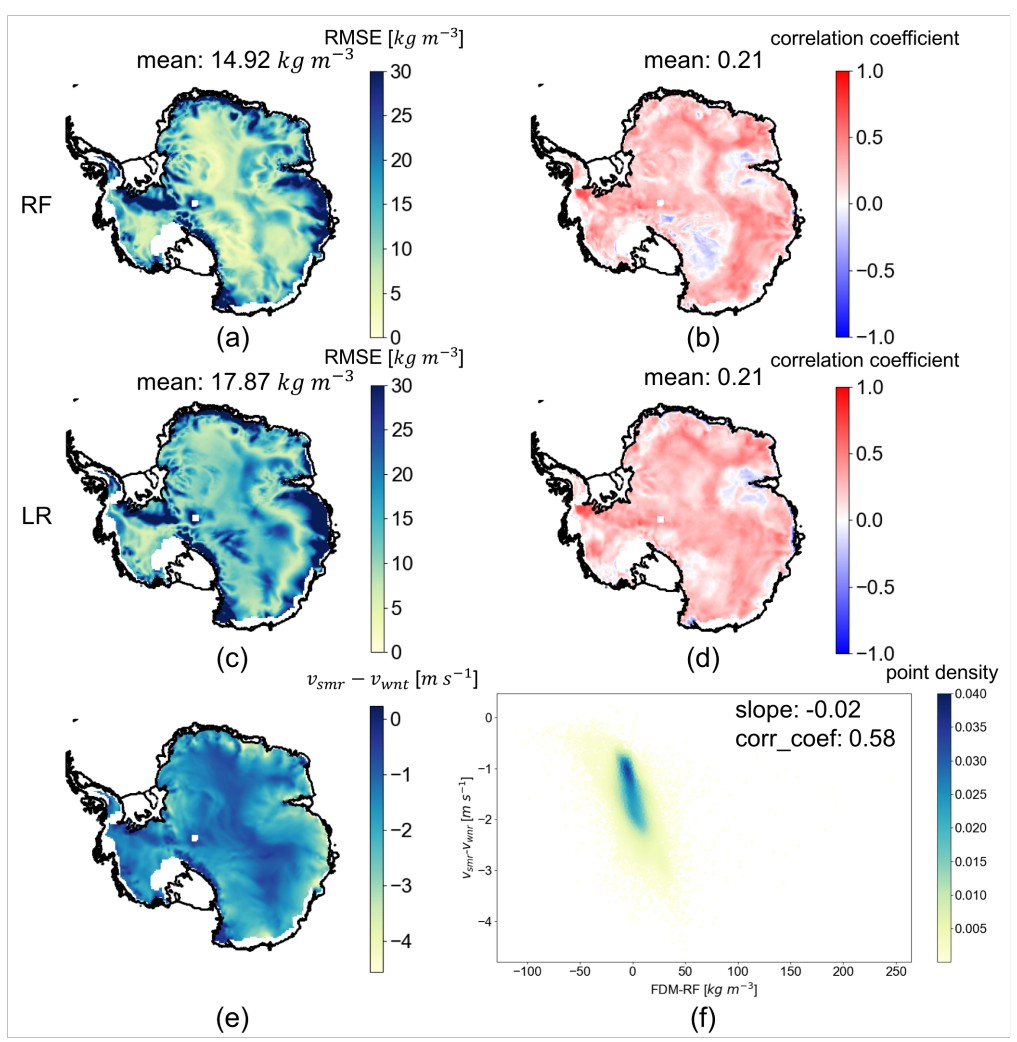

**Figure 7.** Map of (a) root mean square error (RMSE) between IMAU-FDM 40 cm densities and RF densities, (b) the correlation coefficient between IMAU-FDM 40 cm densities and RF densities, (c) root mean square error (RMSE) between IMAU-FDM 40 cm densities and LR densities, (d) the correlation coefficient between IMAU-FDM 40 cm densities and LR densities, (e) the difference between the summer ($v_{smr}$) and winter wind velocity ($v_{wnt}$) from ERA5, and (f) scatterplot of density difference between IMAU-FDM and RF versus the difference between summer and winter wind velocities, coloured by the density distribution of points. The coastline of the maps is from Depoorter et al. (2013).

in Fig. 2a), indicating uncertainties potentially introduced by topography which has an impact on coarse-resolution satellite data. Finally, a potential usability of the RF regression at other depths persists, therefore a comparison between the performance of RF at different depths is provided in Appendix D.

## 4.5   Temporal assessment of RF densities at random pixels

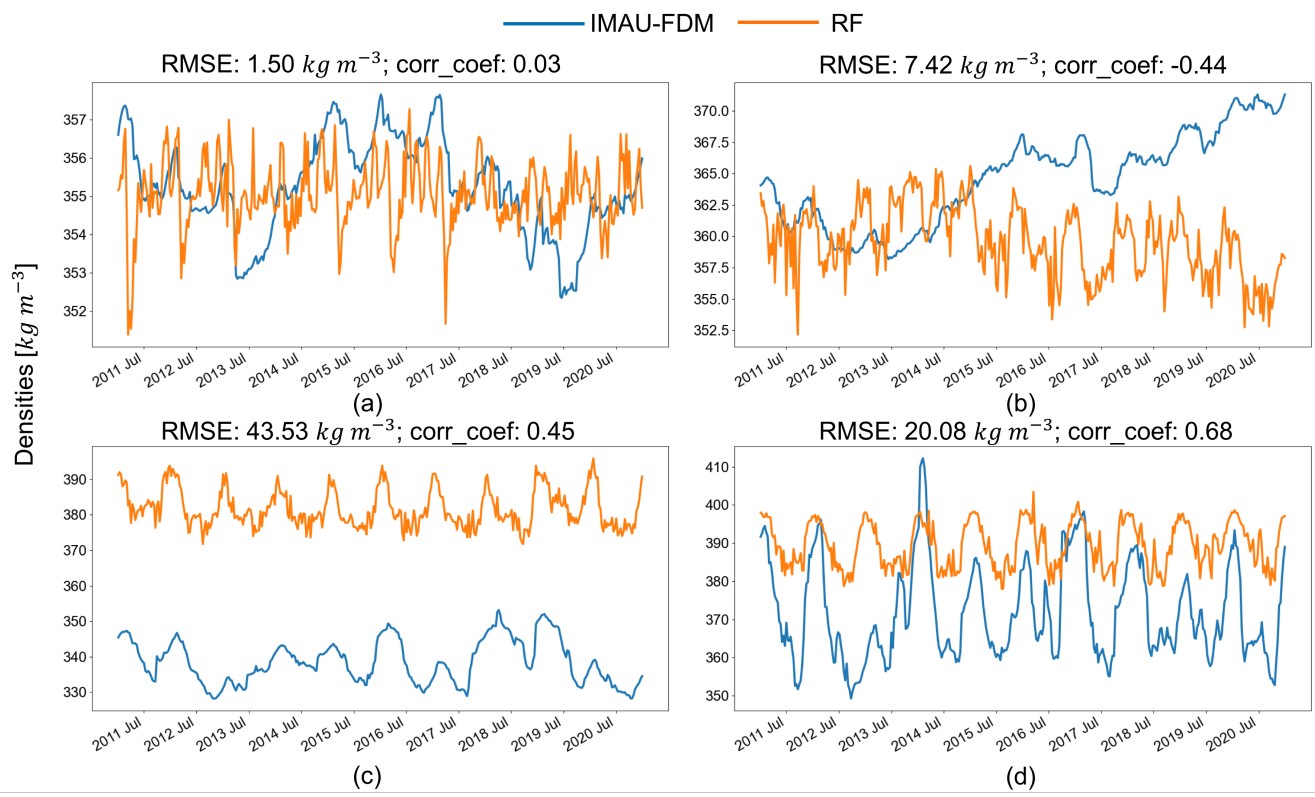

**Figure 8.** Comparison between time series of IMAU-FDM densities (in blue) and RF densities (in orange) at 4 representative sample points and Dome C. Panels (a)–(d) correspond to A–D labelled in Fig.4. The RMSE and correlation coefficient (corr_coef) between the IMAU-FDM densities and the RF densities are shown above each figure.

In Fig. 8, individual pixels are inspected to understand the temporal differences between IMAU-FDM and RF densities.
Pixels A shows a low RMSE as well as a low correlation. Pixel B shows a relatively low RMSE, but a negative correlation. Pixel C shows a reasonable correlation coefficient, but a large bias. Pixel D shows overall most ideal RF performance. From the time series, it is apparent that the RF density estimations generally exhibit a stronger and more consistent seasonal cycle compared to the IMAU-FDM densities, which display a less consistent seasonal pattern with stronger inter-annual variations. This discrepancy explains the relatively low correlation coefficients, as only the pixels with similar seasonal cycles to the
satellite observations (e.g., panels C and D) exhibit a higher correlation between the two datasets.

## 5 Discussion

In this study, we developed a novel approach to estimate Antarctic firn densities using satellite radiometer and scatterometer observations using a RF regressor and IMAU-FDM density outputs as reference data. Our study is based on the complexity of the relationship between satellite observations and firn density. Despite a theoretical impact of surface climate conditions such as temperature, wind and precipitation on both satellite parameters and firn density at a shallow depth (Fraser et al., 2016), the lack of a consistent linear relationship was evident in the examination of the individual satellite observations, as the highest mean temporal correlation between satellite observations and the 40 cm IMAU-FDM firn density is 0.24.

Our study first adopted an unsupervised machine learning method (hierarchical clustering) to distinguish dry snow zones from zones that experienced melt as a preparation step to the density estimation using the random forest (RF) regressor. In contrast to Tran et al. (2008), our study could distinguish melt occurrences, possibly based on the abrupt rise of $T_B$ during melt (Johnson et al., 2020) and the $\sigma_A^0$ rise due to ice-layer formation following melt events (Trusel et al., 2012). However, in some coastal regions in East Antarctica, our clustering method may be less sensitive to melt compared to Brucker et al. (2010) and Picard et al. (2012), resulting in more dry-snow pixels. Among dry-snow zones, Firn 1 consists of most interior regions, hence is characterised by the smallest variations in satellite parameters and is overall most stable, whilst Firn 4 is located in West Antarctica, hence is least stable with the largest variations in satellite parameters. The main difference between Firn 2 and Firn 3 is characterised by a larger $\sigma_{Aanom}^0$ variation in Firn 2; the spatial separation between the two clusters resembles Fig. 4 in Stokes et al. (2022), in which the region overlapping with Firn 2 tends to lose mass while the region overlapping with Firn 3 tends to slightly gain mass. Therefore, we infer that this result might indicate that Firn 2 has a less stable condition than Firn 3.

To address the non-linear and complex nature of the relationship between satellite parameters and firn density, we employed an RF regressor model. This model allowed us to incorporate multiple input parameters and handle non-linear relationships effectively. The implementation of the RF regressor successfully reproduced the spatial pattern of the IMAU-FDM density, achieving a low root mean square error (RMSE) of $14.92\ kg\ m^{-3}$, which outperforms the RMSE of a simple linear regression model ($17.87\ kg\ m^{-3}$). This highlights the potential of using satellite parameters to create a map of long-term mean densities, matching the conclusion of Fraser et al. (2016), who managed to reconstruct one of the satellite observations ($\sigma_A^0$) using climate and firn parameters in the long term.

However, it is important to note some limitations and discrepancies in the RF density map. We observed a slight overestimation of densities in the interior of the Antarctic ice sheet, coupled with an underestimation towards the coastal regions, when compared to the IMAU-FDM densities. This discrepancy may arise from the inability of the RF regressor to extrapolate beyond the training data, leading to the restricted density range in the RF density map (maximum density of $\leq 450\ kg\ m^{-3}$). Furthermore, when comparing the RF and IMAU-FDM densities with the in situ SUMup measurements, we found comparable errors. Similar errors were reported by Keenan et al. (2021), who attributed them to local meteorological phenomena not captured by climate models and possible measurement uncertainties. These factors, which are not explicitly accounted for in the IMAU-FDM model or the RF regressor trained on that dataset, may contribute to the discrepancies observed. Finally, our combination of satellite parameters cannot be used to assess densities at depths deeper than approximately 80 cm. This

limitation is first because of the theoretical penetration depth as shown in Appendix A: a depth exceeding 80 cm is physically not meaningful for the 37 GHz microwave. Another reason for this limitation is that our study is based on the assumption that the surface climate conditions can affect both shallow-depth firn densities and satellite parameters simultaneously (Fraser et al., 2016). Firn densities at larger depth are not largely affected by surface conditions, hence our combination of satellite parameters is not applicable, even if 19 GHz and C-band microwave have a theoretical penetration depth larger than 5 m (as shown in Appendix D). Finally, C-band microwave is more sensitive to surface roughness than to densities at larger depths (as shown in Appendix A).

While the RF regressor successfully captures the spatial variability of the long-term mean density, it falls short in accurately predicting the temporal variation in IMAU-FDM, particularly in coastal regions and megadune areas. Apart from the aforementioned potential underestimation of melt pixels of our clustering method in coastal regions, the temporal discrepancies between the RF regressor and IMAU-FDM can be attributed to the differences in seasonal patterns and the presence of complex climate conditions near the ice shelves. Coastal regions, characterised by large negative differences in wind velocity between summer and winter, exhibit larger temporal discrepancies. These findings suggest that IMAU-FDM may not capture the seasonal cycle of fresh snow density in these regions with high wind speeds during winter. The simplicity of how the density of freshly fallen snow is calculated within IMAU-FDM, assuming linear dependencies with wind speed and surface temperature (Veldhuijsen et al., 2023), fails to account for the intricate processes involving crystal size, shape, and riming, which are influenced by temperature and wind speed conditions (Judson and Doesken, 2000). The dependence of fresh snow density on wind speed may differ under various temperature conditions, which contributes to the discrepancies observed.

In summary, the RF regressor trained using IMAU-FDM and satellite parameters demonstrates promising results in capturing the spatial pattern of firn density. However, it may not fully capture the temporal fluctuations of IMAU-FDM, primarily due to the dominant influence of surface temperature (represented by $T_B$) in the RF estimation. The effects of precipitation (e.g., represented by changes in $\sigma_A^0$ Fraser et al., 2016), and wind velocity (e.g., documented by Champollion et al., 2013) are therefore potentially compromised in the RF model. Additionally, the discrepancy between the meteorological forcing in the IMAU-FDM model and the actual meteorological phenomena can also play a role. The meteorological phenomena can affect the satellite parameters, which in turn influence the RF results, but may not be reflected in the IMAU-FDM output. Our approach of training the RF regressor on IMAU-FDM, which may exhibit spatial and temporal differences compared to actual in situ densities, can therefore be considered a major shortcoming. This limitation should be taken into consideration when interpreting the RF density estimations. Future research could benefit from incorporating more in situ measurements for training the RF regressor, which would improve the accuracy of the temporal density estimates. Furthermore, care should also be taken when using the coarse resolution IMAU-FDM and satellite data to represent the local firn densities. The firn property variation may be small in pixels with relatively flat topography such as Dome C (Picard et al., 2014). However, towards the coastal or mountainous regions, the ability of such coarse resolution to represent firn densities could be compromised, as a mismatch between the local meteorological phenomena, the satellite parameters and the modelled densities can be introduced. Indirect correlations between different layers of firn should also be considered when applying data fusion of multiple microwave frequencies. Additionally, exploring alternative machine learning algorithms, neural network or ensemble approaches may further

enhance the performance of density estimation and capture the complex relationships between satellite observations and firn density, as assessed by Santi et al. (2012b); Anilkumar et al. (2023). Finally, our study only demonstrated a simple approach in understanding the long-term correlation between firn density and satellite parameters, based on climate conditions that potentially affect them (Fraser et al., 2016). However, due to the different penetration abilities of different microwave frequencies (Surdyk, 2002) at different locations (Picard et al., 2009), future research can benefit from a more quantitative assessment regarding to what extent the penetration depths and other climate parameters affect the results. Better parametrisation of satellite observations which can indicate the variation of firn depth (Santi et al., 2012a; Michel et al., 2014) as well as surface and depth hoar-crystal formation and disappearance (Champollion et al., 2013) can also be adopted.

Despite the limitations and discrepancies observed, the RF density map generated in this study can serve as an important intermediate step in translating satellite data into density estimations. It provides valuable insights into the discrepancy between firn models and satellite observations, shedding light on the complexities of the relationship between satellite parameters and firn density. The RF regressor captures the long-term mean density pattern, offering a useful tool for investigating spatial variations in firn density across Antarctica. However, it is essential to exercise caution when interpreting the temporal variations, particularly in coastal regions with complex climate conditions. Our study is also mainly limited to firn densities at shallow depths where the climate phenomena have a large impact; it cannot indicate the actual scattering of firn grains, as a more complicated mechanism persists (Picard et al., 2022).

Further improvements can be made to enhance the accuracy of the RF regressor in capturing the temporal variations of firn density. This could involve refining the training data and incorporating additional meteorological parameters that influence the satellite observations, as also suggested by Kar and Aksoy (2024). By better accounting for the effects of precipitation and wind velocity on the satellite parameters, the RF regressor could potentially capture a more accurate representation of the temporal dynamics of firn density. Furthermore, advancements in the parametrisation of fresh snow density within firn models, considering the complex processes driven by temperature and wind speed conditions, could help bridge the gap between model predictions and satellite observations. Finally, as the performance of the machine learning method varies based on different meteorological phenomena and topography, it can also be recommended for further studies to apply different parametrisations for different regions or test other machine learning methods.

# 6 Conclusions

In conclusion, this study demonstrates the potential of using multiple satellite observations to estimate Antarctic firn densities, with the IMAU-FDM densities serving as a reference. Our findings highlight several key points. Firstly, while satellite observations exhibit a certain level of spatial correlations with firn densities, a consistent linear relationship cannot be established. The correlations between $\rho_{40cm}$ and satellite parameters, particularly $T_B$, indicate the potential influence of firn density on variations of satellite observations.

Secondly, the impact of firn melt and refreeze on satellite observations is significant. Temporal anomalies in satellite parameters can be adopted to differentiate between wet and dry firn regions. Clustering of satellite observation time series helps to

**Table A1.** Firn properties adopted from Larue et al. (2021), including geographical coordinates, annual temperature, vertically averaged density, and vertically averaged specific surface area (SSA) at different locations.

| Name | Latitude [°] | Longitude [°] | Temperature [°C] | SSA [$m^{-2}\ kg^{-1}$] | Density [$kg\ m^{-3}$] |
|---|---|---|---|---|---|
| charcot[A] | -69.38 | 139.02 | -37.9 | 12.0 | 433 |
| ago5[E] | -77.24 | 123.48 | -54.4 | 7.4 | 361 |
| paleo[E] | -79.85 | 126.20 | -50.5 | 7.7 | 392 |

identify melt extents and assess the temporal correlation with densities at the cluster level. Notably, the scattering impact of refrozen melt layers is reflected in prolonged elevated $\sigma_A^0$ anomalies. However, in dry snow clusters, the correlation between densities and satellite observations is not evident.

Based on these complexities, a non-linear model, such as the random forest (RF) regressor, is necessary to capture the relationship between firn densities and satellite observations. Our implementation of the RF regressor successfully reproduces the spatial pattern of firn densities, exhibiting good agreement with IMAU-FDM and even outperforming it in certain locations when compared with SUMup density measurements. However, the temporal simulation of densities by the RF regressor is compromised. Individual pixel analyses reveal that the RF densities tend to overlook the inter-annual variations in firn densities when the variations of satellite observations are not in phase with IMAU-FDM densities. In coastal regions, where satellite signals with strong variability dominate, the RF densities are not directly comparable to IMAU-FDM densities. These temporal discrepancies can be attributed to the simplifications in the IMAU-FDM model, particularly in capturing wind and temperature dependencies that strongly influence satellite observations. Furthermore, limitations of the RF regressor, including the inability to extrapolate from the training dataset and its strong dependence on brightness temperatures, result in a limited range of density estimation and primarily reflect surface temperatures.

*Data availability.* The SSMIS data are available at https://nsidc.org/data/nsidc-0001/versions/6. The ASCAT Enhanced Resolution Image Products are available at https://www.scp.byu.edu/data/Ascat/SIR/Ascat_sir.html. The SUMup data are available at https://arcticdata.io/catalog/view/doi:10.18739/A2ZS2KD0Z. The ERA5 land hourly data are available at https://doi.org/10.24381/cds.e2161bac. The IMAU-FDM data are available upon request from s.b.m.veldhuijsen@uu.nl.

## Appendix A: Sensitivity of microwave to changes in firn properties at different depths

For setting up the experiment, it is important to understand up to which depth can different microwave frequencies indicate firn properties. This appendix presents a simple sensitivity analysis using SMRT where densities and grain sizes at different depths of the firn are varied, and the impact of changes in firn properties on $\sigma_A^0$ and on $T_B$ is presented. The initial state is a firn layer with a 20 m thickness, composed of small internal layers of 40 cm. The density and grain size are changed by $50\ kg\ m^{-3}$ and 0.5 mm, respectively. The changes are applied to one layer at a time. The sensitivity can also vary per location,

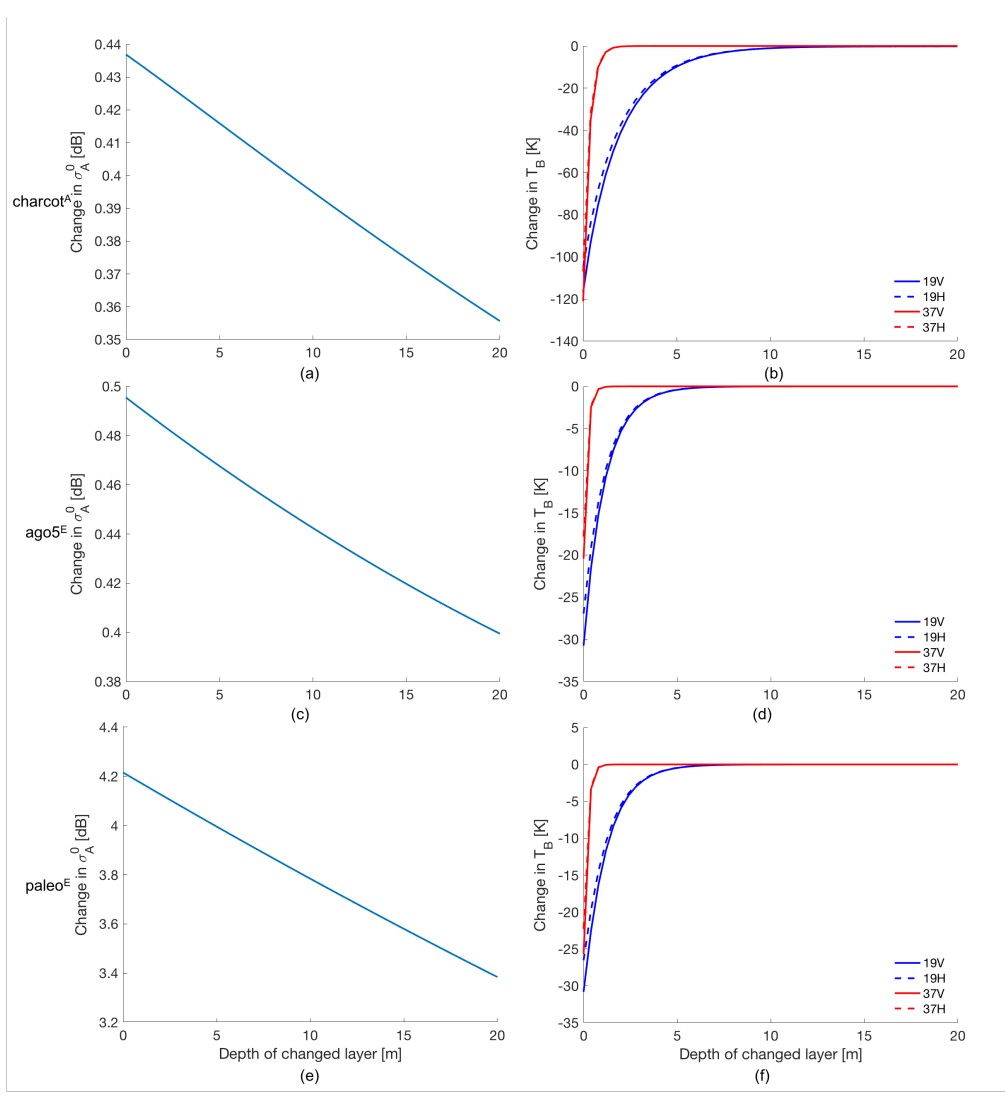

**Figure A1.** Change in $\sigma_A^0$ and $T_B$ as a function of the depth of the layer whose density and grain size are changed.

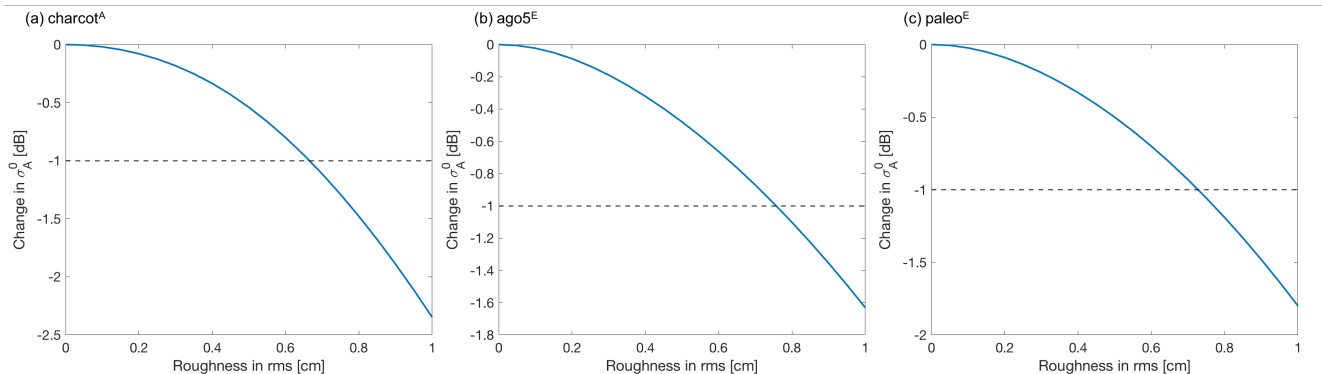

**Figure A2.** Change in C-band $\sigma_A^0$ when the surface roughness (expressed as root mean square heights; rms) is changed. The dashed line indicates when the sensitivity exceeds 1 dB.

therefore we adopt the field measurements in East Antarctica from Larue et al. (2021) to define the initial density, temperature and grain size. The locations and parameters are summarised in Table A1. For the implementation of SMRT, we use a sticky-hard-spheres microstructure model represented by the grain radius and a stickiness parameter (Picard et al., 2018). The grain radius is derived from SSA with

$$r = \frac{3 \times 2.3}{SSA\rho_{ice}} \tag{A1}$$

where $\rho_{ice} = 917\ kg\ m^{-3}$ (Larue et al., 2021). The stickiness is defined as 0.2 for all locations (Picard et al., 2018, 2022). For solving the radiative transfer equation, SMRT uses the discrete ordinate and eigenvalue (DORT) method, and the empirical electromagnetic theory we adopt is the improved Born approximation (IBA) (Mätzler, 1998). The simulated results are shown in Fig. A1, where the changes in $\sigma_A^0$ and $T_B$ with respect to the original state are presented. In general, the sensitivity of both $\sigma_A^0$ and $T_B$ decreases with an increasing depth. 19 GHz and 37 GHz are sensitive up to 6–10 m and 0.8–1 m, respectively. However, the variation of $\sigma_A^0$ is below 1 dB (the radiometric uncertainty; Schmidt et al., 2018), indicating that C-band may not be sufficiently sensitive to volume scattering. Therefore, we consider the effect of surface scattering, which can be modelled by SMRT using the Integral Equation Method (IEM) theory (Fung et al., 1992). Applying the IEM theory requires the snow surface to be defined by the surface roughness expressed as root mean square (rms) heights and correlation length (Larue et al., 2021). In this experiment, we fix the correlation length to 0.1 cm, and vary the surface roughness between 0 and 1 cm. The sensitivity of $\sigma_A^0$ to surface roughness is shown in Fig. A2, where the change in $\sigma_A^0$ indicates the difference between an increased surface roughness and a smooth surface. For all tested locations, $\sigma_A^0$ shows a reduction that exceeds 1 dB when surface roughness increases by 0.7 cm, indicating a sufficient sensitivity to surface roughness. Typically, the changes in surface roughness is related to both wind patterns and surface firn density. Therefore, for the setting of our study, an optimal range to assess firn densities should be chosen between the surface and a depth of 80 cm.

## Appendix B: Temporally averaged satellite parameters and IMAU-FDM density

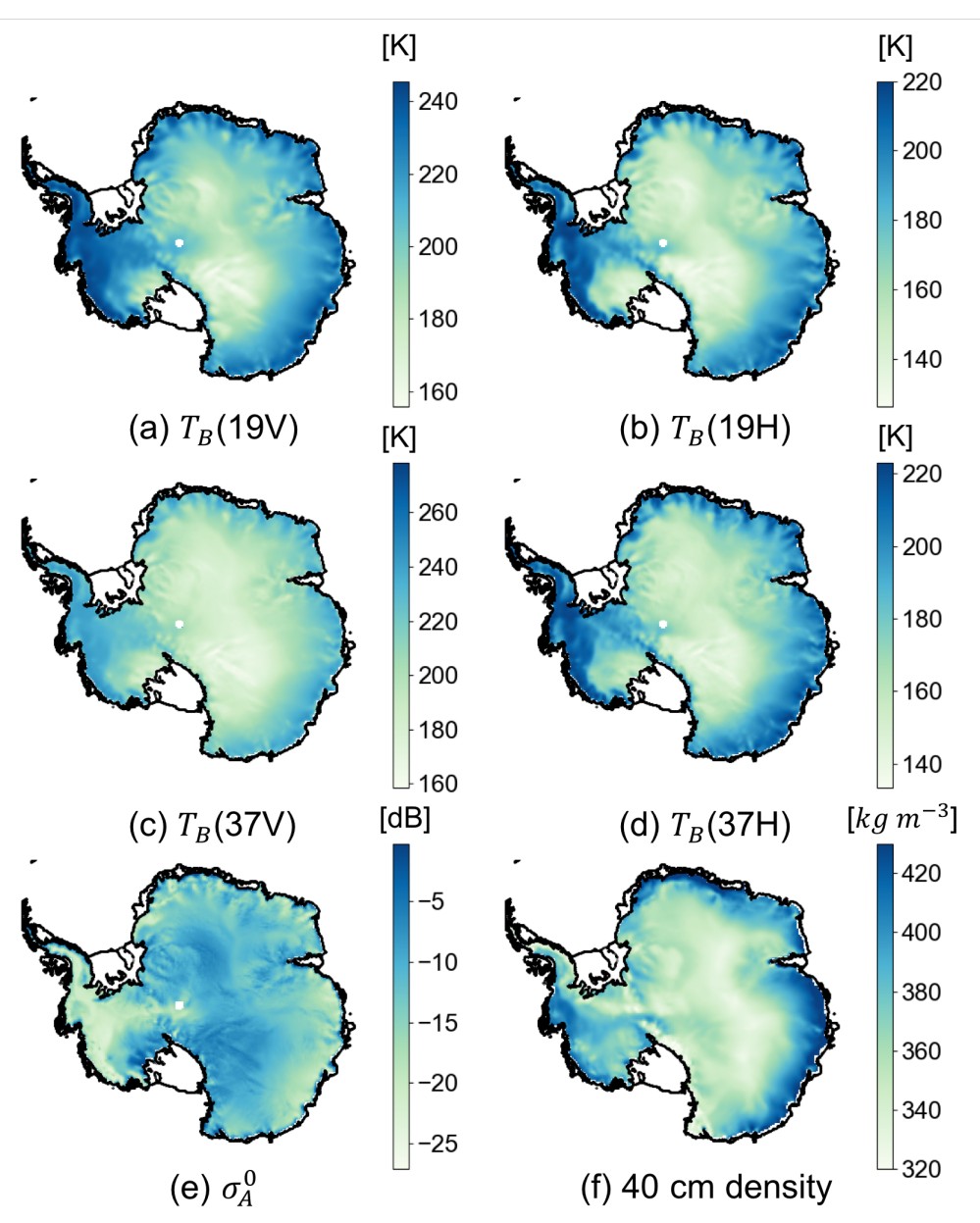

**Figure B1.** Temporally averaged map of (a) brightness temperature ($T_B$) from 19 GHz vertical polarisation, (b) $T_B$ from 19 GHz horizontal polarisation, (c) $T_B$ from 37 GHz vertical polarisation, (d) $T_B$ from 37 GHz horizontal polarisation, (e) backscatter intensity ($\sigma_A^0$), and (f) 40 cm IMAU-FDM density ($\rho_{40cm}$). (a)–(d) are acquired or derived parameters from SSMIS, and (e) is derived from ASCAT. The coastline is from Depoorter et al. (2013).

**Table D1.** Average temporal correlation coefficient between IMAU-FDM near-surface density (4 cm) and IMAU-FDM density at deeper depths.

| Depth | 12 cm | 40 cm | 1 m | 5 m |
|-------|-------|-------|-----|-----|
| 4 cm  | 1.00  | 0.73  | 0.36 | 0.10 |

Figure B1 displays the averaged maps of satellite parameters and $\rho_{40cm}$. This figure aims to demonstrate an overview of the spatial patterns of the data applied in this study. The figure shows that, although all satellite parameters reflect some of the spatial patterns of firn density, none of the parameters shows a spatially consistent relation with $\rho_{40cm}$. For example, in high-elevation regions of East Antarctica, firn densities show similar spatial patterns to $T_B$ and reversed spatial patterns of $\sigma_A^0$. However, these patterns are not consistently observed in West Antarctica, along the Transantarctic Mountains (location shown in Fig. 2a), and in cluster Firn 5 (Fig. 4), where a significant melt event in 2016 affected the satellite observations (Nicolas et al., 2017).

## Appendix C: Time series of clustering results

This appendix section presents the time series of different clusters following Sect. 3.2. Figure C1 presents the time series of the mean and 20th–80th percentiles of each parameter for each cluster, (a)–(g) corresponding to clusters 1–7, respectively. Clusters Firn 1–4 exhibit small and short-term variations in $T_{Banom}$ and $\sigma_{Aanom}^0$; the extent of variations differ between different clusters. Firn 1 has the smallest variations in $T_{Banom}$ and $\sigma_{Aanom}^0$, which are within $\pm5$ K and $\pm0.25$ dB, respectively. Firn 2 and Firn 3 have a $T_{Banom}$ between $-5$ K and 10 K, however, Firn 2 has a $\sigma_{Aanom}^0$ within $\pm1$ dB, while Firn 3 has a $\sigma_{Aanom}^0$ within $\pm0.5$ dB. Firn 4 is characterised by a $T_{Banom}$ variation within $\pm10$ K and a $\sigma_{Aanom}^0$ variation within $\pm0.5$ dB.

On the contrary, clusters Firn 5–7 all show large and abrupt variations in $T_{Banom}$ and $\sigma_{Aanom}^0$, mainly as a result of melt events (e.g., Nicolas et al., 2017) that drastically change absorption, emission and scattering of microwave radiation and thus the $T_{Banom}$ and $\sigma_{Aanom}^0$. The effects of these melt events are also evident in the time series of the IMAU-FDM densities, as the abrupt changes in firn density are associated with the occurrence of melt events (Amory et al., 2024). For example, this can be clearly seen in the time series of cluster Firn 5, where the melt event of 2016 shows a prolonged effect on the $\sigma_{Aanom}^0$ time series due to the formation of a sub-surface refrozen high-density layer in IMAU-FDM. The high-density layer is detected by the scatterometer with stronger snow penetrating capability. In IMAU-FDM, this high density layer appears also in $\rho_{40cm}$ where it increases by approximately $100 \; kg \; m^{-3}$. The comparison of all clusters highlights the dominant influence of melt events on $T_{Banom}$ and $\sigma_{Aanom}^0$ in the wet-firn pixels, whereas the dry-firn pixels exhibit a more pronounced seasonal variation in satellite parameters. It is important to note that the wet firn clusters are not used in the following RF steps due to the complex impact of the melt–refreeze cycle on satellite observations.

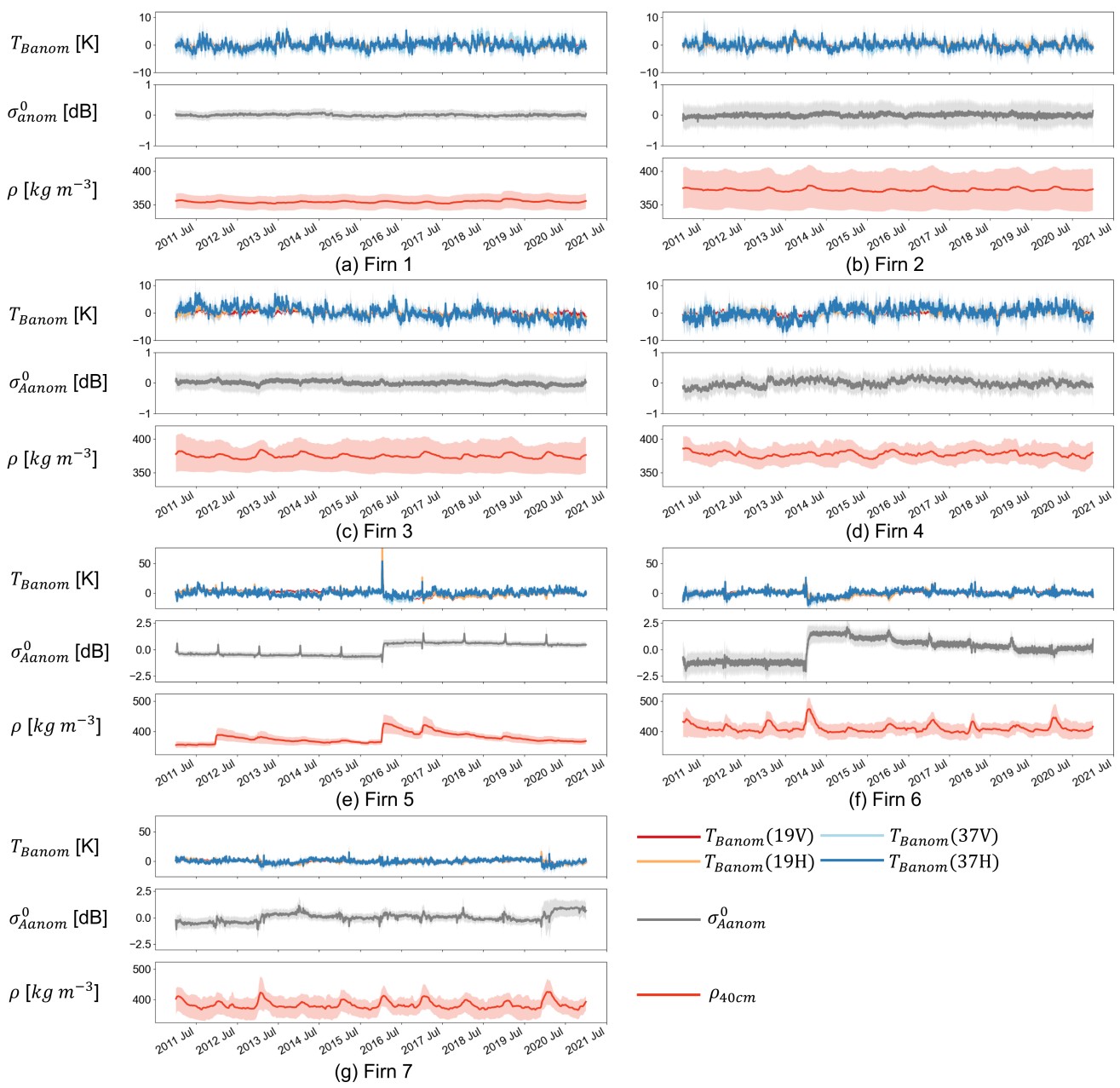

**Figure C1.** Time series of mean (curves) and 20th–80th percentiles (shaded areas) of the clustering results in Fig. 4, (a)–(g) corresponding to Snow facies 1–7. The visualised satellite observations are: time series anomalies of brightness temperature ($T_B$) from 19 GHz and 37 GHz, both horizontal and vertical polarisation ($T_{B_{anom}}(19V)$, $T_{B_{anom}}(19H)$, $T_{B_{anom}}(37V)$ and $T_{B_{anom}}(37H)$, respectively), time series anomalies of backscatter intensity ($\sigma^0_{Aanom}$), and IMAU-FDM density at 40 cm ($\rho_{40cm}$) depth. The colours of the curves correspond to the legends in (g).

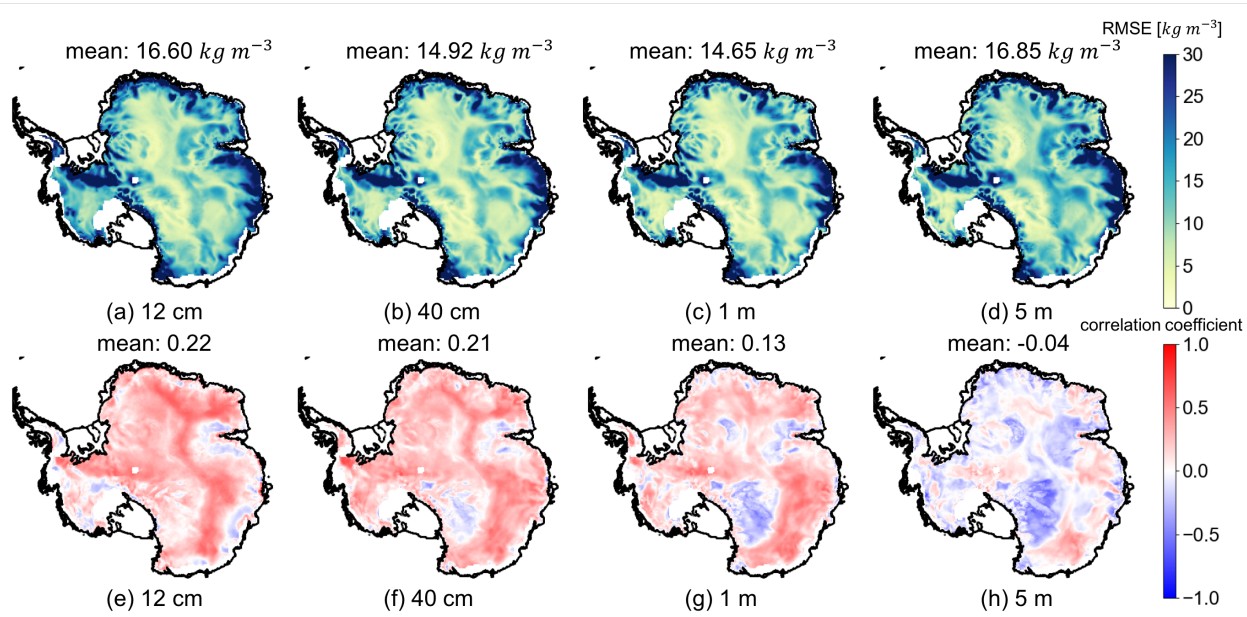

**Figure D1.** Maps of root mean square error (RMSE; upper panel) and correlation coefficients (lower panel) at different depths. The coastline is from Depoorter et al. (2013).

## Appendix D: RF performance with varying depth

In this section, we demonstrate the impact of the depth on the performance of RF. The result shows that as the depth increases, the mean correlation coefficient decreases. Moreover, the reduction of correlation coefficients first occurs in the megadune regions; this observation corresponds with e.g. Picard et al. (2009), who modelled and demonstrated that the penetration depth of 19 GHz is compromised in these regions. When we apply the RF regressor at 5 m density, the RMSE is the highest and the correlation is the lowest, showing most compromised performance. Since our study is based on the impact of surface climate conditions on firn depth, similar performances can be obtained at different depth due to similar impacts from surface climate conditions. We present in Table D1 the correlation coefficients between IMAU-FDM near-surface density (at 4 cm depth) and desity at different depths, and show that this correlation decreases with increasing depths. Therefore, the explanations for different performances in Fig. D1 can be: i) the temporal variation of deeper firn layers are not as sensitive as the upper firn layers and the satellite parameters to climate conditions on the surface; ii) the penetration ability of 37 GHz and 19 GHz largely decreases with firn depths; and iii) biases in IMAU-FDM. This experiment depicts the limitation of our approach, as our combination of satellite parameters is mostly sensitive to surface temperature, and potentially to wind patterns and precipitation on the surface; therefore, it is not indicative of properties of deeper firn layers, although they should be within the radar penetration depths (Rott et al., 1993; Surdyk, 2002). Further studies are therefore encouraged to incorporate better parametrisation of satellite data.

*Author contributions.* WL and SL designed the study. WL conducted data management, processing and analysis; produced the figures; and provided the manuscript with contributions from all co-authors. SV processed and provided the IMAU-FDM densities. SL provided support on data visualisation and analysis.

*Competing interests.* Stef Lhermitte is a member of the editorial board of The Cryosphere.

*Acknowledgements.* Weiran Li is supported by the Dutch Research Council (NWO) on the ALWGO.2017.033 project. Sanne B. M. Veldhuijsen is supported by the Netherlands Organization for Scientific Research (grantno.OCENW.GROOT.2019.091).

We acknowledge National Snow and Ice Data Center (NSIDC) for providing the SSMIS brightness temperature data, Brigham Young University (BYU) Center for Remote Sensing for providing the ASCAT incidence angle normalised backscatter intensity, ECMWF for providing the ERA5 data, and the Surface Mass Balance and Snow on Sea Ice Working Group (SUMup) for providing the firn density measurements over Antarctica.

The authors would also like to thank Ghislain Picard, Pavel Ditmar, Jan Haacker, Sophie de Roda Husman, Ann-Sofie Zinck, and Shashwat Shukla for valuable discussions. Finally, we would like to thank the referees for reviewing and providing recommendations to improve this paper.

ChatGPT is used for grammar checks in parts of the manuscript.

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
