# Peer review of "Machine learning of Antarctic firn density by combining radiometer and scatterometer remote sensing data"

_EGUsphere, 2023_

## Author Comment (AC1)

Response to Emanuele Santi (Referee 2) on egusphere-2023-1556

First, we would like to thank the Referee for reviewing and commenting on the manuscript, which will improve the quality of the manuscript. Please find the item-by-item reply below, with the original comments in *italics* and the responses in blue. All the suggested changes will be implemented in the revised text that will be uploaded.

*The subject of this manuscript is of definite interest for the scientific community. Introduction correctly frames this study in the existing literature, language is clear, and thread deploys smoothly. Innovation with respect to other studies should be however better pointed out and description should be improved in some respects, as well as the presentation of the results. Beside this, the paper suffers from some lacks in the microwave background and I'm suspecting two conceptual issues: the first deals with the attempt to retrieve the density for the 4 cm top layer, which should be quite transparent at the considered MW frequencies in dry conditions. The second concern is about merging direct satellite measurements and derived indices in the RF inputs: based on the information theory, the indices should not bring any additional information independent of the Tb from which they have been computed, so, also based on my experience, these indices should negligibly affect the results.*

We thank the referee for the constructive review and suggestions. We will polish the introduction and the presentation of the results in the revised manuscript. Regarding the penetration of the MW frequencies, while we agree that theoretically a depth of 0.1—2 m should be a more reasonable choice for both 19 GHz and 37 GHz, as we cited in Line 94 of the manuscript (Surdyk, 2002; Brucker et al., 2010), our study is also based on the assumption that the frequency ratios should reflect near-surface (0—2 cm) density, as in Champollion et al. (2013) and Leduc-Leballeur et al. (2017). Therefore, as a compromise between the theoretical penetration depth and the aforementioned applications, we will switch to a depth of 0.12 m to perform the experiments in the revised manuscript. But certainly, as the Referee also pointed out in the detailed comments, the relatively reasonable results can be obtained based on indirect correlation of the top layer density with deeper layers which indeed influence the adopted frequency more. This will also be included in our discussion.

Regarding the validity of using the derived indices, our study was motivated by Tran et al. (2007) and Champollion et al. (2013). Tran et al. (2007) combined a derived Tb ratio with Tb values to cluster snow facies in both Greenland and Antarctica, and Champollion et al. (2013) could associate frequency ratios to near-surface grain size and density at Dome C, Antarctica to a certain extent. In both studies, the validity of using such ratios exists to an extent that should be interesting to discuss, hence we included them.

Moreover, since we use Random Forest regression, we do not agree that the indices cannot bring additional information or performance. While the principle of information theory indeed suggests that indices derived from the original should not introduce additional information, it's important to consider the context in which certain techniques, such as random forest regression, operate. Random forest regression is a powerful ensemble learning method that harnesses the collective strength of multiple decision trees. In the case of random forest regression, the combination of diverse decision trees allows for the detection and extraction of intricate patterns and relationships within the data that may not be readily

apparent in the original dataset. Each tree contributes its unique perspective, and the ensemble's output is often more robust and accurate than that of an individual tree. Therefore, although the indices derived from the original data may seem, from an information theory standpoint, to contain similar information, the strength of random forest lies in its ability to uncover latent, complex patterns that might not be explicitly present in the raw data. This enables the model to provide more nuanced and accurate predictions, surpassing the limitations of a single decision tree.

However, since our study aims to assess a method and discuss the validity of the parameters, we assume that a sensitivity analysis of using different combinations of parameters could be added, where we use as input:
- All parameters as what we are using now
- Only absolute Tb and sigma0
- Only absolute Tb and sigma0, and derived ratios (as also pointed out by Referee 1)

*Detailed comments:*

*Introduction.*

- *The introduction contains a review of the state of the art more than enough to frame this paper. I would only suggest clarifying the aspects related to different spatial resolution, coverage and revisiting when mentioning active and passive MW.*
  This will be clarified in the revised manuscript.

*Section 2*

*Section 2.1.*

- *Equation 1 and 2 are properly referred to the original publications, however a short sentence about the physical principles behind would be useful for the reader.*
  This will be added in the revised manuscript.
- *Line 94-95. The dramatic change in emission mechanism due to the presence of liquid water within the ice sheet might be commented, although this point is mentioned later in section 3.2. Same applies to the scattering in section 2.2.*
  This will be added in the revised manuscript.

*Section 2.2.*

- *the linear correction for local incidence (LIA) sounds me a bit odd. LIA should be already accounted for when computing NRCS to extract the backscattering ($\sigma°$). In any case the backscattering dependence on LIA is not linear at all. Finally, as far as I understand from pag. 5 line 125, at the end you did not use data corrected with eq. 3. Could you further clarify?*
  Perhaps we did not specify the parameters properly. Equation 3 does not describe how we processed the data, but which kind of dataset we used. The same equation can be found in Lindsley and Long (2010), Eq. 3 on page 3. What we are using is the $\sigma°$ normalized to the reference angle (40°), referred to as $A$ in this equation. The $A$

products we are using are already available via Brigham Young University (BYU) Microwave Earth Remote Sensing (MERS) laboratory platform and are directly used in our study. However, since $A$ as a single letter could be misleading, we called it σ° again in the following texts of the manuscript, which is more familiar to the common knowledge. We will clarify that in the revised manuscript.

- *The spatial and temporal co-registration between ASCAT and SSMIS should be better described, this could lead to error and artifacts depending on the processing you applied. At the end, how many co-located Tb and σ° you obtained? It is an important information for better understanding the RF implementation, although something is addressed later.*

  Eventually, we obtained 19,027 valid pixels within the Antarctic ice sheet range (Table 1 of this document). We admit that with a linear interpolation, artifacts occur at the edge of the images. However, we filtered them out using the coastline from Depoorter et al. (2013). What falls within the range of the Antarctic ice sheet should be reliable.

*Section 2.3*

- *line 135 – 138. As stated in the general comments, the attempt to retrieve density at 4 cm raises a conceptual issue. The top 4 cm layer should be almost transparent not only at C-band but also at Ka band in case of dry firn. I'm wondering if you are obtaining results based on indirect correlation of the top layer density with deeper layers to which microwaves are instead sensitive. No wonders if RF achieves successful retrievals: machine learning can exploit almost any kind of input/output relation, but the risk of finding out something based on apparent relationships is always around the corner. If used as "black boxes", ML could potentially relate newborns in China and weather in USA, but which is the utility? I believe a robust physical justification is needed.*

  We agree that it is likely that we have obtained results based on indirect correlation of the top layer density with deeper layers. Please refer to the major comments.

- *Line 138 – 140. The sentence is unclear to me, could you rephrase please. Where was density at 1m depth used later?*

  It was shown in Fig. 4 of the manuscript to prove that melt events have a prolonged impact on deeper snow densities, hence our clustering step to separate dry and melted pixels was quite reliable. However, we agree that overall it does not have added values to the following analyses, hence will remove it in the revised manuscript.

*Section 3*

*Section 3.2.*

- *Is Tb Ratio the same of eq. 1? If so, no need to introduce it again with reference.*
  Yes. This will be changed in the revised manuscript.

- *In my understanding, volume decorrelation was not introduced before. The cited work by Rizzoli is using X band SAR, it is not clear if this finding is also valid for radiometric measurements (scattering and emission are complimentary each other)*
  This application was introduced in Line 59 as previous studies. We repeatedly mentioned it here to point out the difference between our clustering and the previous

studies. However, X-band SAR indeed does not have anything to do with our method, hence we will revise this paragraph.

- *Line 195 – 199. The normalization by firn temperature is embedded in both parameters you defined in eq. 1 and 2. Which is therefore the reason for removing the average seasonal Tb signal? And which the one for doing the same with backscattering that is almost insensitive to temperature? Moreover, machine learning techniques as RF can cope with redundant, noisy, and biased data, so dealing with timeseries of measurements or their anomalies should not change much the results. Finally, there is also a concern in merging Tb with their ratios that is commented below.*

    This step (everything in Section 3.2) serves to separate dry pixels from pixels that suffered from melt, therefore Tb ratios and RF are not used here. While backscattering is almost insensitive to temperature, it is very sensitive to melt events and subsequent melt layers, hence the anomalies should be a good indicator of melt pixels. We agree that this motivation has not been clarified in the manuscript, hence we will improve it.

- *Lines 201 – 211. The clustering algorithm should be better explained maybe with a supporting figure/diagram. I don't believe a reader unfamiliar with Ward algorithm can understand this section.*

    This will be improved in the revised manuscript.

*Section 3.3*

- *Lines 231 – 246. With "sample" do you refer to the set of temporally and spatially coregistered SSMIS and ASCAT measurements for the given pixel? In my understanding, for both subsets I and II you selected randomly 100 pixels from the 7 clusters over Antarctica described in section 3.2 (that is spatial, 25 km resolution each pixel) and you considered the timeseries of satellite measurements (that is temporal, approx. 1 set of SSMIS + ASCAT measurements per pixel day per 10 years). At the end you should have used 365300 sets for training and the same data amount for testing. In other words, you considered about 125000 Km2 for training and testing and applied the trained RF on the remaining ≃14000000 Km2 of Antarctic surface, which is notable. Maybe some more information could be provided…*

    We indeed refer to "the set of temporally and spatially coregistered SSMIS and ASCAT measurements for the given pixel". This will be clarified in the revised manuscript. But for subsets I and II we selected 100 pixels from the 4 dry clusters instead of the 7 clusters. We should also clarify a mistake in the original manuscript that since we use the 10-day resolution IMAU-FDM, the total time slots should be 366 instead of 3653. Therefore, we used approximately 0.6% of the total data for training. We will use 10% of the total data for training (as subset I) in the revised manuscript, as is slightly improves the result (Fig. 1 of this document) and is theoretically more reasonable than using 100 pixels (~0.6% of the data).

Table 1. Statistics of pixels per cluster and pixels used for RF estimating in the original manuscript.

| Cluster | Number of pixels | Number of training pixels |
|---------|------------------|---------------------------|
| Firn 1 | 4540 | 26 |

| | | |
|---|---|---|
| Firn 2 | 7360 | 42 |
| Firn 3 | 3465 | 20 |
| Firn 4 | 2284 | 12 |
| Firn 5 | 429 | 0 |
| Firn 6 | 325 | 0 |
| Firn 7 | 624 | 0 |
| Total | 19027 | 100 |

[Figure]

Figure 1. Comparison between using 100 pixels and 10% of the data for training.

- *Equation 4. The proposed input combination raises another concern: from the information theory, the Tb ratios do not bring to the RF additional information independent of the Tb from which they have been computed, therefore (this is also my personal experience) the results should not be affected by these inputs (or conversely by Tb if you use the ratios). Clarification is needed.*
  Please refer to the major comments, where we argue why derived indices can effectively add information when used in Random Forest regression as these are based on decision trees and derived indices can play an important role there (as they might be important in different phases of the decision tree)
- *Line 258. Gini importance should be better referred and briefly commented. Which is the difference with e.g., predictor importance proposed by Breiman?*
  This will be improved in the revised manuscript. Regarding the difference between the Gini importance and the permutation (Breiman) importance (Fig. 2), we notice that using the permutation importance, the ranking of the original horizontal channels goes down. We will use both in the revised manuscript, or switch to the permutation importance.

[Figure]

Figure 2. Comparison between upper: Gini importance, and lower: permutation (Breiman) importance.

*Section 4*

- *Section 4.1. Following the comment above, this is the core of my concerns: the scarce correlation with density at 4 cm could be depending on the microwaves' scarce sensitivity to such shallow depth. Also, the reverse correlation along the coasts should be depending on melting not entirely removed that occurs more frequently than in the central part of Antarctica. Again, the physics behind should be analysed.*
  We agree. A deeper snow density (12 cm) will be assessed, and the potential melting will be added in the discussion.
- *Figure 1. Although referred to in section 4.1, I find this figure poorly informative. My suggestion is to remove or replace with something more meaningful.*
  We would like to keep it to give the reader an overview of the parameters we used, including their spatial patterns.
- *Figure 2. Did you evaluate the correlation with density at 1 m? At the end which was the role of this parameter in your study?*
  We did not evaluate the correlation with density at 1 m. Ideally we intended to show that our analysis should serve for multiple depths, however this was not well addressed in the manuscript. Please note that in the revised manuscript, we opt for

assessing the densities at 12 cm depth instead of 4 cm, therefore the descriptions will be revised accordingly.

- *Figure 4. The plots in the figure are quite small and difficult to read. I would suggest revising.*
  This will be improved in the revised manuscript.
- *Figure 5 left: the scatterplot should refer to the test results (i.e. those obtained on subset II), not to the training results (Subset I). Usually, retrieval scatterplots show the estimated vs. target, not vice-versa. The plot or caption should also cite the statistics and total data amount. Finally, the R value seems even worse than the one of direct correlation with Tb Ku and Ka in figure 2 for most of the pixels. Isn't it? Which is the explanation?*
  It was a mistake in the captions. We indeed used Subset II for this analysis. This figure will be improved in the revised manuscript where we will also swap the axes.

  However, here the R^2 value (ranging between 0 and 1) refers to the linarity, i.e. if we fit a line to the estimated vs. target scatter plot, how the goodness of fit is. This is not the same indicator as the correlation coefficient (ranging between -1 and 1) in Fig. 2.
- *Figure 6 with doubled colorbar is difficult to interpret (especially figure 6d). I would suggest revising.*
  This will be changed into different markers, as also pointed out by Referee 1.
- *Figure 7 why do not also add the Correlation/Determination coefficient maps as those in figure 2? In my view this is more informative than e.g. the 10- years averaged maps of figure 6.*
  We have shown R^2 in Fig. 7 of the manuscript which is the coefficient of determination between FDM and RF, so we believe it has been an informative indicator already. An averaged map in our opinion shows that our method works reasonably well spatially (in contrast to the performance temporally, as shown in the sections afterwards). However, we agree that the metrics of our assessments should be clarified.

**Reference**

Brucker, L., Picard, G., and Fily, M.: Snow grain-size profiles deduced from microwave snow emissivities in Antarctica, Journal of Glaciol ogy, 56, 514–526, https://doi.org/10.3189/002214310792447806, 2010.

Champollion, N., Picard, G., Arnaud, L., Lefebvre, E., and Fily, M.: Hoar crystal development and disappearance at Dome C, Antarctica: observation by near-infrared photography and passive microwave satellite, The Cryosphere, 7, 1247–1262, https://doi.org/10.5194/tc-7-1247-2013, 2013.

Depoorter, M. A., Bamber, J. L., Griggs, J., Lenaerts, J. T. M., Ligtenberg, S. R. M., van den Broeke, M. R., and Moholdt, G.: Synthesized grounding line and ice shelf mask for Antarctica, https://doi.org/10.1594/PANGAEA.819151, supplement to: Depoorter, MA et al. (2013): Calving fluxes and basal melt rates of Antarctic ice shelves. Nature, 502, 89-92, https://doi.org/10.1038/nature12567, 2013.

Leduc-Leballeur, M., Picard, G., Macelloni, G., Arnaud, L., Brogioni, M., Mialon, A., and Kerr, Y.: Influence of snow surface properties on L-band brightness temperature at Dome C, Antarctica, Remote Sensing of Environment, 199, 427–436, https://doi.org/https://doi.org/10.1016/j.rse.2017.07.035, 2017.

Lindsley, R. D. and Long, D. G.: Standard BYU ASCAT Land/Ice Image Products, Tech. rep., Brigham Young University Microwave Earth Remote Sensing (MERS) Laboratory, https://www.scp.byu.edu/docs/pdf/MERS1002.pdf, [Access date: Oct. 12, 2023], 2010.

Surdyk, S.: Using microwave brightness temperature to detect short-term surface air temperature changes in Antarctica: An analytical approach, Remote Sensing of Environment, 80, 256–271, https://doi.org/10.1016/s0034-4257(01)00308-x, 2002.

Tran, N., Remy, F., Feng, H., and Femenias, P.: Snow Facies Over Ice Sheets Derived From Envisat Active and Passive Observations, IEEE Transactions on Geoscience and Remote Sensing, 46, 3694–3708, https://doi.org/10.1109/tgrs.2008.2000818, 2008.

---

## Author Comment (AC2)

Response to Referee 1 on egusphere-2023-1556

First, we would like to thank the Referee for reviewing and commenting on the manuscript, which will improve the quality of the manuscript. Please find the item-by-item reply below, with the original comments in *italics* and the responses in blue. All the suggested changes will be implemented in the revised text that will be uploaded.

*This paper details a study using machine learning (ML) to examine Antarctic firn density. The paper is interesting and needs some further revisions before it is suitable for publication. I have put some suggestions and questions below.*

*Major comments:*

*Introduction, I suggest you start bigger, why does Antarctica ice sheets matter to the globe? Also, I think you need to define firn for folks who are not clear on what it is.*
We appreciate the suggestion, and we will add the importance of Antarctic mass loss and sea-level rise. We will also add an explanation of firn in the revised manuscript.

*On line 142, you say that the firn model has a resolution of 27 km – is that sufficient to capture the firn variations? This is quite coarse, in my opinion. Is this 27 km by 27 km grid cells? I think this needs to be stated more clearly.*
The 27km model resolution is indeed coarse as it corresponds to the resolution of Antarctic wide state-of-the-art climate models that typically drive firn models. This coarse resolution is therefore not expected to capture the fine scale variations on the steep slopes of the Antarctic Peninsula or along grounding lines as the 27x27 km horizontal resolution is too coarse to resolve atmospheric variables. However, this study focuses on dry pixels, which are mainly located in regions of the AIS where climatic gradients, and thus firn property gradients, are not that large.

Moreover, we want to stress that our study is also based on/limited by the coarse resolution of the satellite radiometer (25 km). According to Picard et al. (2014), who compared the metre-scale ground-based brightness temperature measurements to the coarse-resolution satellite brightness temperature measurements around Dome C in Antarctica, there is indeed metre-scale density variation, but "the study also shows that, for the hectometre to kilometre scales, the variations are much smaller. The average of the ground-based brightness temperature is close to the SSM/I and WindSat satellite observations meaning that the investigated area was representative of the pixel of the satellites including Dome C. An important consequence is that spaceborne passive microwave sensors cannot spatially resolve these wind-formed features, but they are very sensitive to the areal proportion of these features." Given the gentle slopes in the interior of Antarctica, we expect this representativeness also to apply to the dry region pixels we studied.

Nevertheless, based on the previous arguments for the representativeness of coarse resolution for both models and satellite observations, we do agree that the coarse resolution may raise questions. To address these, we will adapt the discussion to clarify the impact of resolution.

*I think you need at least one study site figure that has all of the locations you refer to in the paper on one introductory map. See my comment from Line 152, for example.*

We will try to improve the indication of locations in the revised manuscript. To address the concerns of the reviewer, we refer to Fig. 1 of this document (below) for the locations. However, following both reviewers' suggestions, we will increase the training dataset and should assess how to better present the figures.

[Figure]

Figure 1. Indication of the mentioned locations.

*Overall, the study design seems confusing. You take the time to cluster the data, but then you do not use it for the analysis, really. Why would you not use that to identify the dry-snow zones, and then perhaps build multiple RF models to see what zone could be best captured? This seems like an interesting approach to take but was not used. I think that this would also eliminate the need to only model the non-wet areas if you simply remove the regions that do poorly in satellite observations.*

We admit that the description of the study design could be better elucidated. To simply answer the reviewer's question, the purpose of clustering was indeed to identify the dry-snow zones. Then, the clusters are used to ensure that different regions are represented sufficiently.

Overall, we hope the following flowchart (Fig. 2) is helpful in resolving the confusion, which we also noticed in the other comments. In this flowchart, the rectangles represent original parameters consisting of: i) satellite parameters (TB and sigma0), ii) IMAU-FDM densities, iii) external datasets used for result analysis, and iv) a set of hyperparameters to define the RF regressor. The ovals represent derived parameters. The rounded rectangles represent steps of our study. To be specific, the time series anomalies from TB and sigma0 are clustered to identify dry snow zones. Four distinct dry snow zones have been identified, but we have to admit that we could not relate the separation of dry snow zones to actual physical phenomena. Then, for the dry snow zones, estimation of firn densities using RF regressor is performed.

[Figure]

Figure 2. Flowchart of the study design.

The application of the RF regressor consists of three steps (Lines 231—243 of the manuscript). To reduce overfitting, the first step is to use a training dataset (Dataset I in Fig. 2) to perform a hyperparameter tuning through a 5-fold cross validation process (orange rounded rectangle in Fig. 2). The selection of pixels for Dataset I is proportional to the total pixels in each of the clustered dry snow zones, and the actual numbers of pixels are shown in Table 1 below. Please note that both 19027 and 100 are the numbers of pixels, but the features include 10 years of satellite parameters with a temporal resolution of 10 days, therefore the training dataset consists of 100pixels*366time_steps = 36600 samples. RF is trained with the IMAU-FDM densities.

Table 1. Statistics of pixels per cluster and pixels used for further RF estimating.

| Cluster | Number of pixels | Number of training pixels |
|---|---|---|
| Firn 1 | 4540 | 26 |
| Firn 2 | 7360 | 42 |
| Firn 3 | 3465 | 20 |
| Firn 4 | 2284 | 12 |
| Firn 5 | 429 | 0 |
| Firn 6 | 325 | 0 |
| Firn 7 | 624 | 0 |
| Total | 19027 | 100 |

The second step of the application of the RF regressor is to provide a simple visualisation of the performance of the tuned RF regressor, and the importance of each feature. In this step, another 100 pixels (Dataset II) are used. They are again proportional to the number of pixels per cluster, but the locations are different from Dataset I. The target parameter is the densities of Dataset II, again consisting of 100pixels*366time_steps.

The third step is using the tuned RF regressor to estimate the densities over the entire dry snow zones in Antarctica. Please note that after the hyperparameter tuning in the first step, we use the identical set of hyperparameters for the RF regressor in both the second and the third steps. The training dataset is also identical, which remains the samples from Dataset I. We would like to point out that the proportional selection of Dataset I is important, because we also tried using 100 random pixels not restricted by the clusters, and the result degraded in central Antarctica in terms of RMSE (see figure below).

[Figure]

Figure 3. Comparison of performance between using randomly selected pixels (upper row), and proportionally selected pixels (lower row).

Therefore, the clusters are used to ensure that the training samples are selected in a way where different regions are sufficiently represented. We did not train different RF models for different clusters although this should be feasible and interesting, but is outside of the scope of the current paper. Nevertheless, we can add the suggestion to the discussion.

*I do not understand why you didn't use the RF and importances to reduce your model variables. As you show in Figure 5, it looks like these anomalies are not adding much to the RF model. I think you might be able to remove them in the analysis.*
We appreciate the suggestion. However, the hyperparameters are already tuned based on the whole set of parameters. Changing the combination of parameters requires tuning another set of hyperparameters. Therefore, we can add another section to the manuscript regarding changing the combination of the parameters.

*Did you consider other types of ML models, or did you just decide to use RF approaches? Why not consider other approaches?*

We considered using support vector machines (SVMs), but as the previous major comment pointed out, we would like to take advantage of the importances from the RF regressor to understand which parameters are the most influential factors. Moreover, we would like to stress that the scope of this study is to assess the feasibility of "combining radiometer and scatterometer remote sensing data to assess Antarctica-wide dry firn density by using a state-of-the-art ML method" and not to compare different ML algorithms. Therefore, discussing the performances of different supervised ML algorithms is beyond the scope of our study. However, we appreciate the reviewer's suggestion, and agree that a comparison between different machine learning algorithms can be an interesting scope for future studies and we will stress this explicitly in the discussion.

*On lines 325, you say "that do not correspond to changes in densities in dry-firn regions?. This line has me wondering about the objective of your work. Are you interested in the firn estimation or are you interested in the change in firn over time? Is the RF model developed for this? Or, are the clusters? You say in the beginning of the paper (Line 71) that the objective of this paper is to "assess the feasibility of combining radiometer and scatterometer remote sensing data to assess Antarctica-wide dry firn density." But, you also say on Line 220 "As our goal is to relate the satellite time series to assess spatio-temporal variations in firn density, we adopt an alternative approach that uses the output of IMAU-FDM as training data instead of relying on in situ data.". What is the objective of this work? If it is average firn, then you can develop your model in one way, but if it is not, then you should develop it in another.*

The main objective is to propose and assess a methodology to derive firn density and its spatial and temporal variations over the Antarctic ice sheet based on on daily satellite observations (and not on changes in these observations). More specifically, assuming firn densities in several locations are known, our study tries to assess firn densities of the unknown regions in space and time using a combination of satellite observations, namely brightness temperature (Tb) from SSMIS, and backscatter intensity (sigma0) from ASCAT. The motivation is that multiple drivers (e.g. wind velocity, firn temperature) of changes in satellite observation can also drive the changes in firn densities, but the mechanism has not been explicitly quantified or modelled. The "known densities" in our study, are assumed to be the modelled firn density from IMAU-FDM, which is a firn model. Therefore, this paper focuses on both the spatial estimation of firn density, which seems to work well, but also on the temporal variations, which performs less well. Since RF method is based on the daily observations, it does not directly account for changes (e.g. by including change parameters in the RF model), but it does so indirectly by assuming that the satellite data reflect these changes as well. We will clarify that better in the revised version.

*Minor Comments:*

*Line 17, short-term (or seasonal) variations, is it both or do you just mean seasonal?*
It should be both. The parentheses will be removed in the revised manuscript.

*Line 30, This statement needs a reference.*
We assumed that the references in the previous sentence (Macelloni et al., 2007 and Champollion et al., 2013) would also be applied here. This will be clarified in the revised manuscript.

*Line 63 However, the precise mechanisms underlying the interaction between firn densities and satellite observations cannot always be fully understood (Champollion et al., 2013; Fraser et al., 2016; Rizzoli et al., 2017). What do you mean by this? Interaction implies they are interacting, which they are not…*

Indeed. We will clarify and adapt it in the manuscript that we look at the effects of density on satellite observations.

*Line 67, "to other areas or time periods therefore requires further assessment (Tran et al., 2008; Fraser et al., 2016; Nicolas et al., 2017; Rizzoli et al., 2017)". What did they find? Was it successful, i.e, did it work?*

Tran et al. (2008): this study classified snow facies over both Greenland and Antarctica in 2004 based on passive microwave data (brightness temperature) and altimeter data (backscatter intensity) using an unsupervised ML method. The study regarding Antarctica did not capture melt zones, but indicated "a strong topographic control on the class distribution". This is already different from our study, as we managed to detect melt zones in the more recent decade.

Fraser et al. (2016): this study discussed the scatterometer "backscatter response to surface forcing parameters (wind speed and persistence, precipitation, surface temperature, density and grain size)" by comparing the backscatter with modelled parameters between 2007 and 2012. The study shows that sigma0 is affected by surface temperature and wind speed, hence provides theoretical background for our study.

Nicolas et al. (2017): this study identified a melt region in West Antarctica, close to the Ross Ice Shelf, hence provides theoretical background for our study.

Rizzoli et al. (2017): this study characterised snow facies over Greenland using interferometric synthetic aperture radar (InSAR) acquisitions. The study identified melt zones using an unsupervised ML algorithm, hence provides theoretical background for our study.

We will try to add it to the introduction concisely.

*Generally, italicize In situ.*

Perhaps it is not necessary for The Cryosphere; see Orsolini et al. (2019), for example.

*On line 70, you talk about calibration. You did not mention calibration previously, and it is unclear what this is referring to. Models? The satellites? Fusion methods? I think this needs to be tied to modeling and why calibration is needed. Otherwise it seems to be coming in the text out of the blue.*

We agree and this statement will be rephrased.

*Line 72, you talk here at three experiments, did you compare /use the observations in situ ever? It seems like the SUMup is not used (or mentioned) in any one of the experiments. I think if you are going to mention SUMup, you need to say where it was applied in the experiments.*

SUMup is not used for setting up the experiments, but for the validation and analysis of where the potential errors come from. This will be clarified in the revised manuscript.

*Line 132, "outputs of the regional atmospheric climate model RACMO2.3p2" These scales seem really different… What resolution is the model run at?*
We will clarify this as follows:

Lines 131-133:
IMAU-FDM simulates the transient evolution of the Antarctic firn column, and is forced at the upper boundary by outputs of the Regional Atmospheric Climate Model (RACMO2.3p2) at a 27 km horizontal resolution (van Wessem et al., 2018).

*Line 140 – move these two sentences up to say this earlier (perhaps line 131), that will assist with my previous comment. The first sentence of this paragraph could be combined with the previous one.*
This will be implemented in the revised manuscript.

*Line 132, RACMO2.3p2 – define?*
We will give the definition with capitals. Please see the comments above for the definition.

*Generally, through the text, you refer to "the model output", or "models". As you have multiple models, I suggest calling the models by their names, or ensuring they are referenced clearly to distinguish the model.*
This will be better clarified in the revised manuscript.

*Line 135, we focus on the density of the… How many layers are there in this model in total?*
The model employs up to 300 layers in total of 3 to 15 cm thickness, which represent the firn properties in a Lagrangian way. The output is resampled to a regular grid with layers of 4 cm.

*Line 142, the firn data are reprojected – this is modeled data, correct? I think you want to make sure to differentiate the model from the observations.*
Yes, this will be clarified as "the firn density model data from IMAU-FDM".

*Line 138, "…have been acquired at approximately this depth…" Why? This seems kind of arbitrary. Also, 4 cm seems very shallow for firn. Is this because it is in Antarctica?*
It is mainly because in Fig. 9, we compare IMAU-FDM with in situ measurements acquired in 2014—2015. The in situ measurements were acquired within 0—2 cm depth. This is comparable to the highest vertical resolution of IMAU-FDM dataset we are using (4 cm).

*Line 145, Surface Mass Balance and Snow on Sea Ice Working Group (SUMup) dataset. You have already used this acronym, define it earlier.*
This will be improved in the revised manuscript (together with the issue on Line 72).

*Line 146, "at the smallest mid-point depths" More clarity please, what is 'small' and what is the mid-point of?*
Mid-point refers to the mid-point of the ice sample. SUMup provides information on start-point, end-point and mid-point. We use the mid-point here to define the depth of the reference data. Sometimes multiple samples are taken at each location. To make the visualisation clearer, we only use the shallowest depth of the samples at each location. This will be clarified in the revised version.

*Line 151- For each date of measurement at each location, talk about the locations and dates first... What locations are these dates at?*

We are sorry but we did not really understand the nature of this comment. However, we have SUMup data at specific locations (shown in Fig. 6a and 6b of the manuscript) sampled at different moments in the period between 1984 and 2017 and a time series at Dome-C (shown in Fig. 3 of the manuscript).

*Line 152, Dome C, where is this? Map?*

This is mentioned in Line 269 and shown in Fig. 3. We will improve the manuscript to refer to the figure which shows the location (Label E in Fig. 3).

*Line 159, By incorporating this information... I don't understand how the ERA5 data was used and why it was used. This needs to be better explained.*

As mentioned in Line 156 of the manuscript: To assess the difference between the measured, modelled and estimated densities, it is important to understand the effects of climate conditions. Therefore, we use the climatic data as a comparison. This should also resolve the comment below. We will explain it better in the revised manuscript.

*In Section 2.5, are you talking about comparing model and observations (at points?).. and the satellites? I think this needs to be thought through and justified in the text. Comparing satellite and model data with single point measurements is tricky. There are a lot of references out there about how to do this, particularly in the climate modeling realm. I suggest the authors read some of these papers and at least add a discussion in the text around this.*

The point of this section is to point out that potential errors with IMAU-FDM are linked to certain climate conditions, which can be propagated through the training process to further bias the results. ERA5 serves to help understand in which conditions IMAU-FDM leads to more ideal results. This analysis is done Antarctica-wide, and has nothing to do with comparing model and observations at points.

*Sometimes you say "firn data" and other times you talk dry firn. Should this be defined? Can you make sure you are being consistent through the text?*

Yes, this will be made clear in the revised manuscript.

*Line 168m dry-snow zones, what are these?*

Section 3 (until 3.1) is a high level description of the next section to provide an overview of the approach. The dry-snow zones are therefore explained in Section 3.2, as is also indicated between brackets. It is a preview that will be explained in Section 3.2.

*Line 184, model training procedure. Which model are you talking about?*

Random forest model. This will be clarified in the revised manuscript.

*Line 190-195, this is not very clear. Can you rephrase?*

We will rephrase it into:

We expect that clustering the time series of satellite observations will effectively differentiate pixels experiencing melting from those unaffected. By identifying and excluding melt-affected pixels, we can ensure the accuracy of density estimations using the RF regressor. Additionally,

to enhance the RF regressor's ability to capture the characteristics of various dry snow types, we choose training samples based on the identified dry snow types. This approach enables the representation of diverse snow types in the training dataset, improving the RF regressor's accuracy in estimating density across different snow types.

*Line 196, variations of other properties. What other properties?*
This statement intended to tell that by removing the surface temperatures, the non-annual variations such as melt—refreezing cycles, potential precipitations and density or snow grain size variations could be kept, which in turn helps us distinguish different snow regions especially distinguish melt from non-melt regions and facilitates the following steps (please refer to Fig. 2 of this document as well). This will be better clarified in the revised manuscript.

*Line 196, In addition, although may not have such large dependence on firn temperature as TB, we use its time series anomalies to maintain consistency with TB. This is unclear, can you rephrase?*
There was a mistake. sigma0 is also affected by firn temperature. We will rewrite the whole concept in the revised manuscript.

*Line 204, 'distance' between pixels. Make sure to clarify that this is not spatial distance.*
Should be distance between features of the pixels. We will clarify that in the revised manuscript.

*Line 205, between the parameters of different pixels. What parameters?*
All parameters mentioned in Lines 195—200. Please refer to the following answer as well.

*Line 210, different satellite parameters, together with the IMAU-FDM density for each cluster. I do not understand what this means. What parameters? I think you need to make a table of parameters.*
The satellite parameters include brightness temperatures and derived ratios, as well as scatterometer backscatter intensity. We will clarify it better in the revised manuscript.

*Line 217, RF regressor. Add more references since this approach is now widely used in climate science, for snow distribution mapping and other work.*
This will be improved in the revised manuscript. For example, we will add Vafakhah (2022) and Viallon-Galinier (2023).

*Line 225, for pattern recognition in noisy datasets. Add a reference here.*
This will be added in the revised manuscript.

*Line 226, reduce the variance of the model and prevent overfitting. Add a reference.*
This will be added in the revised manuscript.

*Did you consider other ML approaches?*
Please refer to the major comments.

*Line 230-onward. Are you building a RF for each time step to estimate the timeseries at each grid cell? How many samples in total went into the model? Line 244 talked about pixels, and*

*the resulting sample size. Is this the total number of samples? Do you think this is enough to train a RF, especially considering the results?*

No, we do not rebuild an RF for each time step. We build one RF model that can be used for multiple time steps. The RF is tuned for multiple pixels (100 and please refer to Fig. 1 of this document) and multiple time steps (3,66 daily samples between January1, 2011, and December31, 2020) using the training dataset through a 5-fold cross validation process, and the tuned RF is then used throughout the other experiments. For the details please refer to the major comments.

However, we do appreciate the suggestion to increase the sample size, and also considered using 10% of all pixels instead of 100 pixels for training. Please note again that when we use 10% of the pixels, the total number of samples consists of 1739pixels*366time_steps. The performances (RMSE and R^2) are shown below. From this comparison, we see that increasing the training samples slightly improves the RMSE overall, and enhances R^2 in some regions. We will include it in the revised manuscript.

[Figure]

Figure 4. Comparison between using 0.5% of the data and 10% of the data for training.

*Table 1. These are all hyperparameters, do you call them parameters (which you use other times in the paper for the actual input parameters to the model, which in itself is confusing).*
They are not the same. Hyperparameters are hyperparameters used by the RF regressor, as stated in the caption of this table. It is also a typical element of supervised machine learning (see Anilkumar et al. (2023)). Input parameters are referred to as parameters, including radiometer and scatterometer measurements and the derived products. Please refer to the major comments (Fig. 2 of this document).

*Line 258. Why did you only use Gini importance vs other importance metrics? Did this choices affect any of your importance rankings?*

We considered the permutation importance, and the ranked importance (using 10% of the pixels) is shown below:

[Figure]

Figure 5. Comparison between upper: Gini importance, and lower: permutation (Breiman) importance.

The rankings indeed changed. We will include both importances in the revised manuscript.

*Line 264, by means of the RMSE. Do you mean averages or do you mean via using RMSEs?*
Sorry for the confusion. We mean by using the RMSEs. This will be clarified in the manuscript

*Line 269, this is the first time you refer to Figures... why do it here and not in the rest of the methods?*
This figure will be updated (see major comments) to reflect all sample locations and will be used throughout the methods section to better show the locations of the different sample locations.

*Line 277, 'cluster Firn 5', you have not introduced the cluster results yet so we do not know what these are.*
We will better refer to the clusters in the revised manuscript.

*Figure 1, can you tell us which ones are from the satellites parameters and which ones are from the model? Again, a table might help with this.*

We will improve the manuscript. To address the reviewer's concern, the datasets and the differences of data sources are: 1) the observable Tb from SSMIS satellite mission, 2) sigma0 from ASCAT satellite, and 3) densities from IMAU-FDM, which is a modelled parameter.

*Line 283, especially at the location of Dome C, again, need to show on the maps or include a figure.*

Please refer to the major comments.

*Line 287, There are a lot of other good reasons why the RF should be used, I do not think that this is the strongest one.*

We refer to Anilkumar et al. (2023) for the performance of the RF. Moreover, the advantages of RF include the capability of capturing non-linear relationships between input features and the target variable; less sensitivity to outliers compared to simple linear regression; handling the correlated predictor variables; providing a unique feature importance; no assumption about data distribution; as well as less overfitting. This will be clarified in the revised manuscript.

*Section 4.2 Do you think that you could produce different RF models for each cluster, perhaps? I think this would be very interesting to understand the difference between the performance of each of these models. For instance, if the dry firn can be modeled with lower RMSE /error than some of the other clusters. Honestly, I am still unclear if you did it this way or not.*

We are sorry that the approach was not 100% clear. The clusters are used to ensure that the training samples are selected in a way where different regions are sufficiently represented. We did not train different RF models for different clusters although this should be feasible, but is outside of the scope of the current paper. (Please also see the major comments).

*Figure 4 and Figure 5a*
*These figures are making me wonder what it is that you are trying to do. For instance, are you trying to estimate the time series of the seasonality and variability you see in Figure 4 with the RF? What is the RF estimating, exactly…? You say "firn densities based on satellite parameters" and you talk about a time series, but I am wondering how you are doing this. What is X in Equation 4, actually? I do not know if this is ever said.*

Figures 4 and 5 are from separate experiments. First, we would like to refer to the answer to the major comments. So, Fig. 4 shows that after clustering, dry firn zones and firn zones that experience melt can be distinctively recognised. However, between different dry firn zones, we cannot intuitively relate the time series to actual physical firn properties (mainly due to lack of field measurements). Nevertheless, by proportionally choosing training points within each cluster, we do observe an optimal performance of the experiment (see Fig. 3 of this document). We attribute this to the reason that all types of regions are represented sufficiently in this way.

X refers to the set of features.

*Figure 5b, are all these parameters standardized? Is the importance based on the standardized inputs? I just wonder because the anomalies appear to be the least important, which makes*

*me wonder if perhaps the other parameters are not. Again, if these are not contributing much to the model, did you play around with them being removed? Does the model improve with fewer parameters? Are there any strong correlations between these parameters at all? How are they related or not related to each other?*

The parameters are not standardised. We assumed that random forest does not require standardising, as the tree partitioning depends on the scales of the independent variables. Moreover, Fig. 1 of the manuscript shows that sigma0 varies between -25dB and 0dB, yet ranks as an important feature.

Typically, all TB values are highly correlated to each other, as they are mainly affected by firn temperature. We understand that one may be concerned to have multiple correlating features, hence performed another experiment using only vertical channel of 19 GHz brightness temperature, sigma0, polarisation ratios and frequency ratios. The comparison is shown below. It is interesting to see that the original setting still outperforms in terms of RMSE.

[Figure]

Figure 6. Comparison between using selected parameters (upper row) and the original setting (lower row).

*Line 301, The differences between these clusters mainly arise from deviations in TBanom and, to a lesser extent, sigma0anom. What is different about them?*

We notice that for cluster 1, TBanom varies between -5K and 5K, for cluster 2 and 3, TBanom varies betwee -5K and 10K, and for cluster 4, TBanom varies between -10K and 10K. Moreover, compared to cluster 2, TBanom of cluster 3 experienced a decreasing trend over time. What we could assume is that cluster 1 consists of most interior regions, hence is overall most stable, whilst cluster 4 is located in West Antarctica, hence is least stable (with the largest variations). The separation between cluster 2 and cluster 3 resembles Fig. 4 in Stokes et al. (2022), in which cluster 2 tends to lose mass while cluster 3 tends to slightly gain mass. However, we

can only infer that this result might indicate that cluster 2 has a less stable condition than cluster 3, but the conclusion is not solid.

*Line 298, If 1-4 are the basically the same, why are they not being treated as a single cluster?*
We would not conclude that clusters 1—4 are "basically the same". Rather, at the moment we cannot relate the differences to actual physical phenomena.

*Line 303, Are the melt events shown /evidenced in time in the region? Can you talk about this a little bit? You refer to a paper, but don't go into detail otherwise.*
We refer to de Roda Husman et al. (2022), where it shows that satellite-based melt events are commonly well recognised.

*Line 305, Can you describe why how density would change under these melt events, and why? You do not give much background on that.*
After melt events, the density increases by refreezing. This is a typical phenomenon, which is also documented in Fig. 4 of Nilsson et al. (2015) that showed the high-density melt layers during the famous melt over Greenland in 2012.

*Line 306, Firn 5, where the melt event of 2016 shows a prolonged effect on the anom time series due to the formation of a sub-surface refrozen high-density layer in IMAU-FDM. Again, what I the implications for this, and what does it mean for firn?*
We refer to Nilsson et al. (2015), where it shows that a sub-surface refrozen layer drastically changes the volume scattering mechanism hence changes the backscattering signals.

*Figure 6. Again, this figure only shows results as temporal averages. How did the time series of the RF do?*
We cannot show all the time series as there are 17649 pixels all together. That is the reason why we took 9 sample pixels to visualise the time series in Fig. 8, and conclude that the precise temporal performance of RF is compromised.

*Line 383, It is important to note that the wet firn clusters are not used in the following RF steps due to the complex impact of the melt–refreeze cycle on satellite observations. Again, I am thinking that the RF and this cluster analysis is not related.*
Please also refer to the major comments and Fig. 2 of this document. To briefly address this question, the clustering separated the wet firn from the dry firn, so it helps the following analysis.

*Line 317, Exhibiting a linear relationship between predictors and the predicted variable – predictand? Saying it this way is confusing.*
We will check the consistency of descriptions here.

*Figure 5, add units.*
This will be corrected in the revised manuscript.

*Figure 5a, Why do no values exceed this amount? I wonder if perhaps your training data set somehow selects lower firn values… are you randomizing between your training and test sets?*

As y-axis (IMAU-FDM densities) of this figure shows, we have selected density values up to 500kg/m^3. Please note that within 4cm depth of the snow in Antarctica, it is normal to have most of the density below 400kg/m^3, so it is possible that the values that exceed 400kg/m^3 are less represented in the RF training process, which could indeed indicate a sampling issue. Therefore, by using 10% of the pixels as training samples, we hope to better resolve this and add the analysis to the revised manuscript accordingly.

*Figure 6 shows that the model is basically as good as the RF (if not better at anything other than the mean). So, why do you need an RF model in this case? How difficult is the model to set up and apply? Again, is there a good reason for the RF here if it doesn't perform that well, is not finer in scale, or it doesn't really do that well except on average?*

Please refer to the major comments. The objective of our study is to assess the ability of using a combination of ML algorithms and satellite parameters to estimate firn densities, not to reproduce the modelled density. To do this, we require sufficient training data. However, due to the limitation of the in situ measurements, we use IMAU-FDM as an assumption of "real densities". Actually, as Fig. 6 and Fig. 9 indicate, IMAU-FDM does not capture many variations in the in situ data, resulting in temporal gaps in the RF estimations. This has been pointed out and analysed in Lines 390 onwards.

*Figure 6d, can you show which is which? Use different symbols instead of colors? They are difficult to differentiate. This intercomparison with observations is likely very challenging to achieve (which I think is what you are attempting to do). I might suggest some sort of spatial upscaling for single point /insitu observations.*

This will be improved in the revised manuscript.

*Figure 8 illustrates how poor the RF is for a time series. But, I am unclear if you are doing this in the right way. Clarity of methods is required.*

Please refer to the previous comments.

**Reference**

Anilkumar, R., Bharti, R., Chutia, D., and Aggarwal, S. P.: Modelling point mass balance for the glaciers of the Central European Alps using machine learning techniques, The Cryosphere, 17, 2811–2828, https://doi.org/10.5194/tc-17-2811-2023, 2023.

de Roda Husman, S., Hu, Z., Wouters, B., Munneke, P. K., Veldhuijsen, S., and Lhermitte, S.: Remote Sensing of Surface Melt on Antarctica: Opportunities and Challenges, IEEE Journal of Selected Topics in Applied Earth Observations and Remote Sensing, pp. 1–20, https://doi.org/10.1109/jstars.2022.3216953, 2022.

Nilsson, J., Vallelonga, P., Simonsen, S. B., Sørensen, L. S., Forsberg, R., Dahl-Jensen, D., Hirabayashi, M., Goto-Azuma, K., Hvidberg, C. S., Kjaer, H. A., and Satow, K.: Greenland 2012 melt event effects on CryoSat-2 radar altimetry, Geophysical Research Letters, 42, 3919–3926, https://doi.org/10.1002/2015gl063296, 2015.

Orsolini, Y., Wegmann, M., Dutra, E., Liu, B., Balsamo, G., Yang, K., de Rosnay, P., Zhu, C., Wang, W., Senan, R., and Arduini, G.: Evaluation of snow depth and snow cover over the

Tibetan Plateau in global reanalyses using in situ and satellite remote sensing observations, The Cryosphere, 13, 2221–2239, https://doi.org/10.5194/tc-13-2221-2019, 2019.

Picard, G., Royer, A., Arnaud, L., and Fily, M.: Influence of meter-scale wind-formed features on the variability of the microwave brightness temperature around Dome C in Antarctica, The Cryosphere, 8, 1105–1119, https://doi.org/10.5194/tc-8-1105-2014, 2014.

Stokes, C.R., Abram, N.J., Bentley, M.J., Edwards, T.L., England, M.H., Foppert, A., Jamieson, S.S.R., Jones, R.S., King, M.A., Lenaerts, J.T.M., Medley, B., Miles, B.W.J., Paxman, G.J.G., Ritz, C., van de Flierdt, T., Whitehouse, P.L. Response of the East Antarctic Ice Sheet to past and future climate change, Nature, 608, 275–286. https://doi.org/10.1038/s41586-022-04946-0, 2022.

Vafakhah, M., Nasiri Khiavi, A., Janizadeh, S., Ganjkhanlo, H. Evaluating different machine learning algorithms for snow water equivalent prediction, Earth Sci Inform 15, 2431–2445. https://doi.org/10.1007/s12145-022-00846-z, 2022.

Viallon-Galinier, L., Hagenmuller, P., and Eckert, N.: Combining modelled snowpack stability with machine learning to predict avalanche activity, The Cryosphere, 17, 2245–2260, https://doi.org/10.5194/tc-17-2245-2023, 2023.

---

## Author Response (AR1)

**Response to Referees on egusphere-2023-1556**
We appreciate the reviews and comments from both Referees. Please find the response to Referee 1 on pages 1-15, and the response to Referee 2 on pages 16-23.

**Response to Referee 1 on egusphere-2023-1556**

First, we would like to thank the Referee for reviewing and commenting on the manuscript, which will improve the quality of the manuscript. Please find the item-by-item reply below, with the original comments in *italics* and the responses in blue. All the suggested changes are implemented in the revised manuscript.

*This paper details a study using machine learning (ML) to examine Antarctic firn density. The paper is interesting and needs some further revisions before it is suitable for publication. I have put some suggestions and questions below.*

*Major comments:*

*Introduction, I suggest you start bigger, why does Antarctica ice sheets matter to the globe? Also, I think you need to define firn for folks who are not clear on what it is.*
We appreciate the suggestion. This has been implemented in the revised manuscript Lines 21—26.

*On line 142, you say that the firn model has a resolution of 27 km – is that sufficient to capture the firn variations? This is quite coarse, in my opinion. Is this 27 km by 27 km grid cells? I think this needs to be stated more clearly.*
The 27km model resolution is indeed coarse as it corresponds to the resolution of Antarctic wide state-of-the-art climate models that typically drive firn models. This coarse resolution is therefore not expected to capture the fine scale variations on the steep slopes of the Antarctic Peninsula or along grounding lines as the 27x27 km horizontal resolution is too coarse to resolve atmospheric variables. However, this study focuses on dry pixels, which are mainly located in regions of the AIS where climatic gradients, and thus firn property gradients, are not that large.

Moreover, we want to stress that our study is also based on or limited by the coarse resolution of the satellite radiometer (25 km). According to Picard et al. (2014), who compared the metre-scale ground-based brightness temperature measurements to the coarse-resolution satellite brightness temperature measurements around Dome C in Antarctica, there is indeed metre-scale density variation, but "the study also shows that, for the hectometre to kilometre scales, the variations are much smaller. The average of the ground-based brightness temperature is close to the SSM/I and WindSat satellite observations meaning that the investigated area was representative of the pixel of the satellites including Dome C. An important consequence is that spaceborne passive microwave sensors cannot spatially resolve these wind-formed features, but they are very sensitive to the areal proportion of these features." Given the gentle slopes in the interior of Antarctica, we expect this representativeness also to apply to the dry region pixels we studied.

Nevertheless, based on the previous arguments for the representativeness of coarse resolution for both models and satellite observations, we do agree that the coarse resolution may raise questions. To address these, we added the impact of topography on the coarse resolution satellite data on Lines 391—394 and Lines 464—468.

*I think you need at least one study site figure that has all of the locations you refer to in the paper on one introductory map. See my comment from Line 152, for example.*
We improved the indication of locations in the revised manuscript. Following both reviewers' suggestions, we added an example figure in the revised manuscript (Fig. 2) and added labels to indicate specific locations we mentioned in the manuscript.

*Overall, the study design seems confusing. You take the time to cluster the data, but then you do not use it for the analysis, really. Why would you not use that to identify the dry-snow zones, and then perhaps build multiple RF models to see what zone could be best captured? This seems like an interesting approach to take but was not used. I think that this would also eliminate the need to only model the non-wet areas if you simply remove the regions that do poorly in satellite observations.*
We admit that the description of the study design could be better elucidated. To simply answer the reviewer's question, the purpose of clustering was indeed to identify the dry-snow zones. Then, the clusters are used to ensure that different regions are represented sufficiently.

Overall, we hope the following flowchart (Fig. 1) is helpful in resolving the confusion, which we also noticed in the other comments. This flowchart has also been added in the revised manuscript (Fig. 1). In this flowchart, the rectangles represent original parameters consisting of: (i) satellite parameters (TB and sigma0), (ii) IMAU-FDM densities, (iii) external datasets used for result analysis, and (iv) a set of hyperparameters to define the RF regressor. The ovals represent derived parameters. The rounded rectangles represent steps of our study. To be specific, the time series anomalies from TB and sigma0 are clustered to identify dry snow zones. Four distinct dry snow zones have been identified, but we have to admit that we could not relate the separation of dry snow zones to actual physical phenomena. Then, for the dry snow zones, estimation of firn densities using RF regressor is performed.

[Figure]

Figure 1. Flowchart of the study design.

The application of the RF regressor consists of three steps (Lines 253—263 of the manuscript). To reduce overfitting, the first step is to use a training dataset (Dataset I in Fig. 1) to perform a hyperparameter tuning through a 5-fold cross validation process (orange rounded rectangle in Fig. 1). The amount of pixels for Dataset I is 10% of the total pixels in each of the clustered dry snow zones, and the actual numbers of pixels are shown in Table 1 below. Please note that "Total" indicates the numbers of pixels, but the features include 10 years of satellite parameters with a temporal resolution of 10 days, therefore the training dataset consists of 1764pixels*366time_steps = 645,624 samples. RF is trained with the IMAU-FDM densities.

Table 1. Statistics of pixels per cluster and pixels used for further RF estimating.

| Cluster | Number of pixels | Number of Dataset I pixels | Number of Dataset II pixels |
|---|---|---|---|
| Firn 1 | 4540 | 454 | 26 |
| Firn 2 | 7360 | 736 | 42 |
| Firn 3 | 3465 | 346 | 20 |
| Firn 4 | 2284 | 228 | 12 |
| Firn 5 | 429 | 0 | 0 |
| Firn 6 | 325 | 0 | 0 |
| Firn 7 | 624 | 0 | 0 |
| Total | 19027 | 1764 | 100 |

The second step of the application of the RF regressor is to provide a simple visualisation of the performance of the tuned RF regressor, and the importance of each feature. In this step, another 100 pixels (Dataset II) are used. The selection of pixels for Dataset II is proportional to the total pixels in each cluster (Table 1). The target parameter is the densities of Dataset II, consisting of 100pixels*366time_steps = 36,600 samples.

The third step is using the tuned RF regressor to estimate the densities over the entire dry snow zones in Antarctica. Please note that after the hyperparameter tuning in the first step, we use the identical set of hyperparameters for the RF regressor in both the second and the third steps. The training dataset is also identical, which remains the samples from Dataset I. We would like to point out that the proportional selection of Dataset I is important, because we also tried using 100 random pixels not restricted by the clusters, and the result degraded in central Antarctica in terms of RMSE (see figure below).

[Figure]

Figure 2. Comparison of performance between using randomly selected pixels (upper row), and proportionally selected pixels (lower row).

Therefore, the clusters are used to ensure that the training samples are selected in a way where different regions are sufficiently represented. We did not train different RF models for different clusters although this should be feasible and interesting, but is outside of the scope of the current paper.

*I do not understand why you didn't use the RF and importances to reduce your model variables. As you show in Figure 5, it looks like these anomalies are not adding much to the RF model. I think you might be able to remove them in the analysis.*
We appreciate the suggestion. However, the hyperparameters are already tuned based on the whole set of parameters. Changing the combination of parameters requires tuning another set of hyperparameters. Therefore, we added a sensitivity analysis to the manuscript regarding changing the combination of the parameters (Appendix B).

*Did you consider other types of ML models, or did you just decide to use RF approaches? Why not consider other approaches?*

We considered using support vector machines (SVMs), but as the previous major comment pointed out, we would like to take advantage of the importances from the RF regressor to understand which parameters are the most influential factors. Moreover, we would like to stress that the scope of this study is to assess the feasibility of "combining radiometer and scatterometer remote sensing data to assess Antarctica-wide dry firn density by using a state-of-the-art ML method" and not to compare different ML algorithms. Therefore, discussing the performances of different supervised ML algorithms is beyond the scope of our study. However, we appreciate the reviewer's suggestion, and agree that a comparison between different machine learning algorithms can be an interesting scope for future studies and we added it to the discussion (Lines 470—472).

*On lines 325, you say "that do not correspond to changes in densities in dry-firn regions?. This line has me wondering about the objective of your work. Are you interested in the firn estimation or are you interested in the change in firn over time? Is the RF model developed for this? Or, are the clusters? You say in the beginning of the paper (Line 71) that the objective of this paper is to "assess the feasibility of combining radiometer and scatterometer remote sensing data to assess Antarctica-wide dry firn density." But, you also say on Line 220 "As our goal is to relate the satellite time series to assess spatio-temporal variations in firn density, we adopt an alternative approach that uses the output of IMAU-FDM as training data instead of relying on in situ data.". What is the objective of this work? If it is average firn, then you can develop your model in one way, but if it is not, then you should develop it in another.*

The main objective is to propose and assess a methodology to derive firn density and its spatial and temporal variations over the Antarctic ice sheet based on on daily satellite observations (and not on changes in these observations). More specifically, assuming firn densities in several locations are known, our study tries to estimate firn densities of the unknown regions in space and time using a combination of satellite observations, namely brightness temperature (Tb) from SSMIS, and backscatter intensity (sigma0) from ASCAT. The motivation is that multiple drivers (e.g. wind velocity, firn temperature) of changes in satellite observation can also drive the changes in firn densities, but the mechanism has not been explicitly quantified or modelled. The "known densities" in our study, are assumed to be the modelled firn density from IMAU-FDM, which is a firn model. Therefore, this paper focuses on both the spatial estimation of firn density, which seems to work well, but also on the temporal variations, which performs less well. Since RF method is based on the daily observations, it does not directly account for changes (e.g. by including change parameters in the RF model), but it does so indirectly by assuming that the satellite data reflect these changes as well. We have rephrased the objectives in the revised manuscript (Lines 73—86).

*Minor Comments:*

*Line 17, short-term (or seasonal) variations, is it both or do you just mean seasonal?*
It should be both. The parentheses are removed in the revised manuscript.

*Line 30, This statement needs a reference.*
(Now Line 32) The sentences have been rephrased in the revised manuscript to include references.

*Line 63 However, the precise mechanisms underlying the interaction between firn densities and satellite observations cannot always be fully understood (Champollion et al., 2013; Fraser et al., 2016; Rizzoli et al., 2017). What do you mean by this? Interaction implies they are interacting, which they are not…*

(Now Line 67) We have rephrased it into "the impact of firn density on satellite observations".

*Line 67, "to other areas or time periods therefore requires further assessment (Tran et al., 2008; Fraser et al., 2016; Nicolas et al., 2017; Rizzoli et al., 2017)". What did they find? Was it successful, i.e, did it work?*

Tran et al. (2008): this study classified snow facies over both Greenland and Antarctica in 2004 based on passive microwave data (brightness temperature) and altimeter data (backscatter intensity) using an unsupervised ML method. The study regarding Antarctica did not capture melt zones, but indicated "a strong topographic control on the class distribution". This is already different from our study, as we managed to detect melt zones in the more recent decade.

Fraser et al. (2016): this study discussed the scatterometer "backscatter response to surface forcing parameters (wind speed and persistence, precipitation, surface temperature, density and grain size)" by comparing the backscatter with modelled parameters between 2007 and 2012. The study shows that sigma0 is affected by surface temperature and wind speed, hence provides theoretical background for our study.

Nicolas et al. (2017): this study identified a melt region in West Antarctica, close to the Ross Ice Shelf, hence provides theoretical background for our study.

Rizzoli et al. (2017): this study characterised snow facies over Greenland using interferometric synthetic aperture radar (InSAR) acquisitions. The study identified melt zones using an unsupervised ML algorithm, hence provides theoretical background for our study.

We adapted Lines 42—64 to describe the previous works better and rephrased the sentence so it better indicates that these studies were already discussed.

*Generally, italicize In situ.*
We leave this decision of italicising to the copy-editor's decision but have seen other The Cryosphere publications (e.g. Orsolini et al. (2019)) where it was not done.

*On line 70, you talk about calibration. You did not mention calibration previously, and it is unclear what this is referring to. Models? The satellites? Fusion methods? I think this needs to be tied to modeling and why calibration is needed. Otherwise it seems to be coming in the text out of the blue.*
We agree and this statement has been removed.

*Line 72, you talk here at three experiments, did you compare /use the observations in situ ever? It seems like the SUMup is not used (or mentioned) in any one of the experiments. I think if you are going to mention SUMup, you need to say where it was applied in the experiments.*
SUMup is not used for setting up the experiments, but for the validation and analysis of where the potential errors come from. This has been rephrased in the revised manuscript (Line 85).

*Line 132, "outputs of the regional atmospheric climate model RACMO2.3p2" These scales seem really different... What resolution is the model run at?*
(Now Line 144) We have revised it as follows:

*"IMAU-FDM simulates the transient evolution of the Antarctic firn column, and is forced at the upper boundary by outputs of the Regional Atmospheric Climate Model (RACMO2.3p2) at a 27 km horizontal resolution (van Wessem et al., 2018)."*

*Line 140 – move these two sentences up to say this earlier (perhaps line 131), that will assist with my previous comment. The first sentence of this paragraph could be combined with the previous one.*
We have moved the description of the spatial and temporal resolution above to Line 145.

*Line 132, RACMO2.3p2 – define?*
We have given the definition with capitals (Line 145). Please see the comments above for the revised sentence as well.

*Generally, through the text, you refer to "the model output", or "models". As you have multiple models, I suggest calling the models by their names, or ensuring they are referenced clearly to distinguish the model.*
This has been referred to as IMAU-FDM in the revised manuscript.

*Line 135, we focus on the density of the... How many layers are there in this model in total?*
The model employs up to 300 layers in total of 3 to 15 cm thickness, which represent the firn properties in a Lagrangian way. The output is resampled to a regular grid with layers of 4 cm. This has been clarified in Lines 147.

*Line 142, the firn data are reprojected – this is modeled data, correct? I think you want to make sure to differentiate the model from the observations.*
(Now Line 157) Yes, this has been rephrased as "the firn density model data from IMAU-FDM".

*Line 138, "...have been acquired at approximately this depth..." Why? This seems kind of arbitrary. Also, 4 cm seems very shallow for firn. Is this because it is in Antarctica?*
In the revised manuscript we switched from firn depth of 4 cm to 12 cm (also in major comments of Referee 2) and we have clarified our selection for this depth on Lines 151—155:

"For the comparison with satellite observations, we focus on the density of the top 12 cm ($\rho_{12cm}$) from the IMAU-FDM output. We also use $\rho_{12cm}$ for density estimation using the random forest (RF) regressor. The choice of the 12 cm depth is based on (i) the fact that many in situ measurements used for evaluating the density estimations have been acquired at depths that are several centimetres below the surface, e.g. Picard et al. (2012) and Leduc-Leballeur et al. (2017), and (ii) a compromise between the expected penetration depths at 19 GHz and 37 GHz (0.1–2 m; Surdyk, 2002)."

*Line 145, Surface Mass Balance and Snow on Sea Ice Working Group (SUMup) dataset. You have already used this acronym, define it earlier.*

This has been defined in the revised manuscript (Line 86).

*Line 146, "at the smallest mid-point depths" More clarity please, what is 'small' and what is the mid-point of?*
Mid-point refers to the mid-point of the ice sample. SUMup provides information on start-point, end-point and mid-point. We use the mid-point here to define the depth of the reference data. Since sometimes multiple samples are taken at each location, we use the shallowest depth of the mid-point at each location. This has been clarified in the revised version (Lines 161—165) by using the term "shallow" instead of "small".

*Line 151- For each date of measurement at each location, talk about the locations and dates first… What locations are these dates at?*
We are sorry but we did not really understand the nature of this comment. However, we have SUMup data at specific locations (shown in Fig. 7a and 7b of the revised manuscript) sampled at different moments in the period between 1984 and 2017 and a time series at Dome C (location shown in Fig. 2 of the revised manuscript). We have tried to clarify this in the manuscript (Lines 160—168).

*Line 152, Dome C, where is this? Map?*
This is indicated in Fig. 2a of the revised manuscript and has been added in the text (Line 170).

*Line 159, By incorporating this information… I don't understand how the ERA5 data was used and why it was used. This needs to be better explained.*
To assess the difference between the measured, modelled and estimated densities, it is important to understand the effects of climate conditions. Therefore, we use the climatic data as a comparison. This should also resolve the comment below. We have adapted the description in the revised manuscript (Lines 174—181).

*In Section 2.5, are you talking about comparing model and observations (at points?).. and the satellites? I think this needs to be thought through and justified in the text. Comparing satellite and model data with single point measurements is tricky. There are a lot of references out there about how to do this, particularly in the climate modeling realm. I suggest the authors read some of these papers and at least add a discussion in the text around this.*
The idea of this section is to point out that potential errors with IMAU-FDM are linked to certain climate conditions, which can be propagated through the training process to further bias the results. ERA5 serves to help understand in which conditions IMAU-FDM leads to more ideal results. This analysis is done Antarctica-wide, and has nothing to do with comparing model and observations at points. We have clarified this in the revised manuscript (Lines 174—181)

*Sometimes you say "firn data" and other times you talk dry firn. Should this be defined? Can you make sure you are being consistent through the text?*
The definition of firn has been added to the introduction (Line 25) and "firn data" has been removed in the revised manuscript.

*Line 168m dry-snow zones, what are these?*

Section 3 (until 3.1) is a high level description of the next section to provide an overview of the approach. The dry-snow zones are therefore explained in Section 3.2, as is also indicated between brackets. It is a preview that will be explained in Section 3.2.

*Line 184, model training procedure. Which model are you talking about?*
(Now Line 203) Random forest model. This has been changed in the revised manuscript.

*Line 190-195, this is not very clear. Can you rephrase?*
We have rephrased it (Lines 206—213) in the revised manuscript.

*Line 196, variations of other properties. What other properties?*
This statement intended to tell that by removing the surface temperatures, the non-annual variations such as melt—refreezing cycles, potential precipitations and density or snow grain size variations could be kept, which in turn helps us distinguish different snow regions especially distinguish melt from non-melt regions and facilitates the following steps (please refer to Fig. 1 of this document as well). This has been rephrased in the revised manuscript (Line 216).

*Line 196, In addition, although may not have such large dependence on firn temperature as TB, we use its time series anomalies to maintain consistency with TB. This is unclear, can you rephrase?*
There was a mistake. sigma0 is also affected by firn temperature. We have rewritten the concept in the revised manuscript (Line 217).

*Line 204, 'distance' between pixels. Make sure to clarify that this is not spatial distance.*
Should be distance between features of the pixels. We rewritten it in the revised manuscript (Line 227).

*Line 205, between the parameters of different pixels. What parameters?*
TB anomalies and sigma0 anomalies. We introduced Fig. 1 in the revised manuscript to clarify which parameters are used in each step.

*Line 210, different satellite parameters, together with the IMAU-FDM density for each cluster. I do not understand what this means. What parameters? I think you need to make a table of parameters.*
The satellite parameters include brightness temperatures and derived ratios, as well as scatterometer backscatter intensity. We introduced Fig. 1 in the revised manuscript to clarify which parameters are used in each step.

*Line 217, RF regressor. Add more references since this approach is now widely used in climate science, for snow distribution mapping and other work.*
We have added Vafakhah (2022) and Viallon-Galinier (2023) in Line 241.

*Line 225, for pattern recognition in noisy datasets. Add a reference here.*
We refer the advantages of RF to Hastie et al. (2008), which is added in Line 249.

*Line 226, reduce the variance of the model and prevent overfitting. Add a reference.*

Please see the comment above.

*Did you consider other ML approaches?*
Please refer to the major comments and Line 470 of the revised manuscript.

*Line 230-onward. Are you building a RF for each time step to estimate the timeseries at each grid cell? How many samples in total went into the model? Line 244 talked about pixels, and the resulting sample size. Is this the total number of samples? Do you think this is enough to train a RF, especially considering the results?*
No, we do not rebuild an RF for each time step. We build one RF model that can be used for multiple time steps. The RF is tuned for multiple pixels (please refer to Fig. 1 of this document) and multiple time steps (366 daily samples between January 1 2011 and December 31 2020) using the training dataset through a 5-fold cross validation process, and the tuned RF is then used throughout the other experiments. For the details please refer to the major comments.

However, we do appreciate the suggestion to increase the sample size, and also considered using 10% of all pixels instead of 100 pixels for training. Please note again that when we use 10% of the pixels, the total number of samples consists of 1739pixels*366time_steps. The performances (RMSE and R^2) are shown below. From this comparison, we see that increasing the training samples slightly improves the RMSE overall, and enhances R^2 in some regions. Therefore, we have switched using 10% of pixels as training dataset in the revised manuscript.

[Figure]

Figure 3. Comparison between using 100 pixels and 10% of the data for training.

*Table 1. These are all hyperparameters, do you call them parameters (which you use other times in the paper for the actual input parameters to the model, which in itself is confusing).*
They are not the same. Hyperparameters are hyperparameters used by the RF regressor, as stated in the caption of this table. It is also a typical element of supervised machine learning

(see Anilkumar et al. (2023)). Input parameters are referred to as parameters, including radiometer and scatterometer measurements and the derived products. Please refer to the major comments. However, we did make a mistake in writing the column name of this table, which has been corrected in the revised manuscript.

*Line 258. Why did you only use Gini importance vs other importance metrics? Did this choices affect any of your importance rankings?*
We considered the permutation importance, and the comparison has been shown in Fig 6 of the revised manuscript. The descriptions of the importance metrics have been added in Lines 284—293. Using the permutation importance indeed changed the rankings of the absolute values.

*Line 264, by means of the RMSE. Do you mean averages or do you mean via using RMSEs?*
Sorry for the confusion. We mean by using the RMSEs. This has been clarified in the revised manuscript Line 296.

*Line 269, this is the first time you refer to Figures... why do it here and not in the rest of the methods?*
This figure has been updated (see major comments) and specific locations have been shown in Fig. 2a of the revised manuscript.

*Line 277, 'cluster Firn 5', you have not introduced the cluster results yet so we do not know what these are.*
We have removed the reference to cluster 5 to avoid preliminary reference to the clusters.

*Figure 1, can you tell us which ones are from the satellites parameters and which ones are from the model? Again, a table might help with this.*
The datasets and the differences of data sources are: 1) the observable Tb from SSMIS satellite mission, 2) sigma0 from ASCAT satellite, and 3) densities from IMAU-FDM, which is a modelled parameter. We have also added a brief introduction in the revised Fig. A1.

*Line 283, especially at the location of Dome C, again, need to show on the maps or include a figure.*
We have added the locations in Fig. 2 of the revised manuscript (please also refer to the major comments).

*Line 287, There are a lot of other good reasons why the RF should be used, I do not think that this is the strongest one.*
We refer to Anilkumar et al. (2023) for the performance of the RF. We have added the advantages and references in the revised manuscript (Lines 243—247).

*Section 4.2 Do you think that you could produce different RF models for each cluster, perhaps? I think this would be very interesting to understand the difference between the performance of each of these models. For instance, if the dry firn can be modeled with lower RMSE /error than some of the other clusters. Honestly, I am still unclear if you did it this way or not.*
We are sorry that the approach was not 100% clear. The clusters are used to ensure that the training samples are selected in a way where different regions are sufficiently represented.

We did not train different RF models for different clusters although this should be feasible, but is outside of the scope of the current paper (Please also see the major comments). We have clarified this in the manuscript (Lines 206—213).

*Figure 4 and Figure 5a*
*These figures are making me wonder what it is that you are trying to do. For instance, are you trying to estimate the time series of the seasonality and variability you see in Figure 4 with the RF? What is the RF estimating, exactly…? You say "firn densities based on satellite parameters" and you talk about a time series, but I am wondering how you are doing this. What is X in Equation 4, actually? I do not know if this is ever said.*
Figures 4 and 5 (Figs. 5 and 6 in the revised manuscript) are from separate experiments. First, we would like to refer to the answer to the major comments. So, Fig. 5 shows that after clustering, dry firn zones and firn zones that experience melt can be distinctively recognised. However, between different dry firn zones, we cannot intuitively relate the time series to actual physical firn properties (mainly due to lack of field measurements). Nevertheless, by proportionally choosing training points within each cluster, we do observe an optimal performance of the experiment (see Fig. 2 of this document). We attribute this to the reason that all types of regions are represented sufficiently in this way.

X refers to the set of features.

*Figure 5b, are all these parameters standardized? Is the importance based on the standardized inputs? I just wonder because the anomalies appear to be the least important, which makes me wonder if perhaps the other parameters are not. Again, if these are not contributing much to the model, did you play around with them being removed? Does the model improve with fewer parameters? Are there any strong correlations between these parameters at all? How are they related or not related to each other?*
(Figure 6 in the revised manuscript) The parameters are not standardised. We assumed that random forest does not require standardising, as the tree partitioning depends on the scales of the independent variables. Moreover, Fig. A1 of the revised manuscript shows that sigma0 varies between -25dB and 0dB, yet ranks as an important feature.

Typically, all TB values are highly correlated to each other, as they are mainly affected by firn temperature. We performed a sensitivity analysis in the revised manuscript, shown in Appendix B.

*Line 301, The differences between these clusters mainly arise from deviations in TBanom and, to a lesser extent, sigma0anom. What is different about them?*
We notice that for cluster 1, TBanom varies between -5K and 5K, for cluster 2 and 3, TBanom varies between -5K and 10K, and for cluster 4, TBanom varies between -10K and 10K. Moreover, compared to cluster 2, TBanom of cluster 3 experienced a decreasing trend over time. What we could assume is that cluster 1 consists of most interior regions, hence is overall most stable, whilst cluster 4 is located in West Antarctica, hence is least stable (with the largest variations). The separation between cluster 2 and cluster 3 resembles Fig. 4 in Stokes et al. (2022), in which cluster 2 tends to lose mass while cluster 3 tends to slightly gain mass. However, we can only infer that this result might indicate that cluster 2 has a less stable

condition than cluster 3, but the conclusion is not solid. We have clarified these differences in the manuscript as well (Lines 327—332 and Lines 422—427).

*Line 298, If 1-4 are the basically the same, why are they not being treated as a single cluster?*
We would not conclude that clusters 1—4 are "basically the same". Rather, for instance, Firn 1 shows the smallest variations in TB and sigma0 anomalies compared to the other clusters, which corresponds to the most stable firn conditions (as it also locates in the interior of the ice sheet). Firn 2—4 also exhibit such difference accordingly. Please also refer to the previous comment.

*Line 303, Are the melt events shown /evidenced in time in the region? Can you talk about this a little bit? You refer to a paper, but don't go into detail otherwise.*
We refer to de Roda Husman et al. (2022), where it shows that satellite-based melt events are commonly well recognised.

*Line 305, Can you describe why how density would change under these melt events, and why? You do not give much background on that.*
After melt events, the density increases by refreezing. This is a typical phenomenon, which is also documented in Fig. 4 of Nilsson et al. (2015) that showed the high-density melt layers during the famous melt over Greenland in 2012. We added a reference to clarify that in this context (Line 420).

*Line 306, Firn 5, where the melt event of 2016 shows a prolonged effect on the anom time series due to the formation of a sub-surface refrozen high-density layer in IMAU-FDM. Again, what I the implications for this, and what does it mean for firn?*
We refer to Nilsson et al. (2015), where it shows that a sub-surface refrozen layer drastically changes the volume scattering mechanism hence changes the backscattering signals. This process is also clearly described in a recent review on firn, which is now referenced in the revised manuscript to clarify that (Line 337).

*Figure 6. Again, this figure only shows results as temporal averages. How did the time series of the RF do?*
We cannot show all the time series as there are 17649 pixels all together. That is the reason why we took 9 sample pixels to visualise the time series in Fig. 9, and conclude that the precise temporal performance of RF is compromised. Additionally, we want to stress that Fig. 8a and b show the RMSE and the correlation coefficient, therefore reflecting the time series behaviour of the RF model and not only the temporal average.

*Line 383, It is important to note that the wet firn clusters are not used in the following RF steps due to the complex impact of the melt–refreeze cycle on satellite observations. Again, I am thinking that the RF and this cluster analysis is not related.*
Please also refer to the major comments and Fig. 2 of this document. To briefly address this question, the clustering separated the wet firn from the dry firn, so it helps with the following analysis. The selection of training data based on the clusters also facilitates the RF process.

*Line 317, Exhibiting a linear relationship between predictors and the predicted variable – predictand? Saying it this way is confusing.*

(Now Line 348) This has been changed into the IMAU-FDM and RF densities in the revised manuscript.

*Figure 5, add units.*
(Now Fig. 6) This has been corrected in the revised manuscript.

*Figure 5a, Why do no values exceed this amount? I wonder if perhaps your training data set somehow selects lower firn values… are you randomizing between your training and test sets?*
As y-axis (IMAU-FDM densities) of the original figure shows, we have selected density values up to 500kg/m^3. Please note that within 4cm depth of the snow in Antarctica, it is normal to have most of the density below 400kg/m^3, so it is possible that the values that exceed 400kg/m^3 are less represented in the RF training process, which could indeed indicate a sampling issue.

In the revised manuscript, we use the 12cm densities and 10% of the pixels as training samples. New results show that although we manage to select more training data with densities up to 450kg/m^3, RF still shows an underestimation (up to 425kg/m^3). RF also shows an overestimation when the IMAU-FDM densities are lower than 325kg/m^3. We attributed this phenomenon to the limitation of using satellite data to represent firn processes in coastal regions and in regions with more varying topography, and added the discussion in the revised manuscript (Lines 391—394, Lines 465—468, Lines 485—487).

*Figure 6 shows that the model is basically as good as the RF (if not better at anything other than the mean). So, why do you need an RF model in this case? How difficult is the model to set up and apply? Again, is there a good reason for the RF here if it doesn't perform that well, is not finer in scale, or it doesn't really do that well except on average?*
Please refer to the major comments. The objective of our study is to assess the ability of using a combination of ML algorithms and satellite parameters to estimate firn densities, not to reproduce the modelled density. To do this, we require sufficient training data. However, due to the limitation of the in situ measurements, we use IMAU-FDM as an assumption of "real densities". Actually, as Fig. 7 and Fig. 10 indicate, IMAU-FDM does not capture many variations in the in situ data, resulting in temporal gaps in the RF estimations. This has been pointed out and analysed in Lines 442 onwards in the revised manuscript.

*Figure 6d, can you show which is which? Use different symbols instead of colors? They are difficult to differentiate. This intercomparison with observations is likely very challenging to achieve (which I think is what you are attempting to do). I might suggest some sort of spatial upscaling for single point /insitu observations.*
This has been changed in the revised manuscript (Fig. 7d).

*Figure 8 illustrates how poor the RF is for a time series. But, I am unclear if you are doing this in the right way. Clarity of methods is required.*
Please refer to the previous comments. IMAU-FDM can also introduce biases, however, we only use IMAU-FDM as a "known" data to assess our method, but not as an absolute ground truth. The recommendation for further studies have been proposed in Lines 454 onwards.

**Reference**

Anilkumar, R., Bharti, R., Chutia, D., and Aggarwal, S. P.: Modelling point mass balance for the glaciers of the Central European Alps using machine learning techniques, The Cryosphere, 17, 2811–2828, https://doi.org/10.5194/tc-17-2811-2023, 2023.

de Roda Husman, S., Hu, Z., Wouters, B., Munneke, P. K., Veldhuijsen, S., and Lhermitte, S.: Remote Sensing of Surface Melt on Antarctica: Opportunities and Challenges, IEEE Journal of Selected Topics in Applied Earth Observations and Remote Sensing, pp. 1–20, https://doi.org/10.1109/jstars.2022.3216953, 2022.

Hastie, T., Tibshirani, R., and Friedman, J.: Random Forests, pp. 587–604, Springer New York, https://doi.org/10.1007/978-0-387-84858-7_15, 2008.

Nilsson, J., Vallelonga, P., Simonsen, S. B., Sørensen, L. S., Forsberg, R., Dahl-Jensen, D., Hirabayashi, M., Goto-Azuma, K., Hvidberg, C. S., Kjaer, H. A., and Satow, K.: Greenland 2012 melt event effects on CryoSat-2 radar altimetry, Geophysical Research Letters, 42, 3919–3926, https://doi.org/10.1002/2015gl063296, 2015.

Orsolini, Y., Wegmann, M., Dutra, E., Liu, B., Balsamo, G., Yang, K., de Rosnay, P., Zhu, C., Wang, W., Senan, R., and Arduini, G.: Evaluation of snow depth and snow cover over the Tibetan Plateau in global reanalyses using in situ and satellite remote sensing observations, The Cryosphere, 13, 2221–2239, https://doi.org/10.5194/tc-13-2221-2019, 2019.

Picard, G., Royer, A., Arnaud, L., and Fily, M.: Influence of meter-scale wind-formed features on the variability of the microwave brightness temperature around Dome C in Antarctica, The Cryosphere, 8, 1105–1119, https://doi.org/10.5194/tc-8-1105-2014, 2014.

Stokes, C.R., Abram, N.J., Bentley, M.J., Edwards, T.L., England, M.H., Foppert, A., Jamieson, S.S.R., Jones, R.S., King, M.A., Lenaerts, J.T.M., Medley, B., Miles, B.W.J., Paxman, G.J.G., Ritz, C., van de Flierdt, T., Whitehouse, P.L. Response of the East Antarctic Ice Sheet to past and future climate change, Nature, 608, 275–286. https://doi.org/10.1038/s41586-022-04946-0, 2022.

Vafakhah, M., Nasiri Khiavi, A., Janizadeh, S., Ganjkhanlo, H. Evaluating different machine learning algorithms for snow water equivalent prediction, Earth Sci Inform 15, 2431–2445. https://doi.org/10.1007/s12145-022-00846-z, 2022.

Viallon-Galinier, L., Hagenmuller, P., and Eckert, N.: Combining modelled snowpack stability with machine learning to predict avalanche activity, The Cryosphere, 17, 2245–2260, https://doi.org/10.5194/tc-17-2245-2023, 2023.

**Response to Emanuele Santi (Referee 2) on egusphere-2023-1556**

First, we would like to thank the Referee for reviewing and commenting on the manuscript, which will improve the quality of the manuscript. Please find the item-by-item reply below, with the original comments in *italics* and the responses in blue. All the suggested changes are implemented in the revised manuscript.

*The subject of this manuscript is of definite interest for the scientific community. Introduction correctly frames this study in the existing literature, language is clear, and thread deploys smoothly. Innovation with respect to other studies should be however better pointed out and description should be improved in some respects, as well as the presentation of the results. Beside this, the paper suffers from some lacks in the microwave background and I'm suspecting two conceptual issues: the first deals with the attempt to retrieve the density for the 4 cm top layer, which should be quite transparent at the considered MW frequencies in dry conditions. The second concern is about merging direct satellite measurements and derived indices in the RF inputs: based on the information theory, the indices should not bring any additional information independent of the Tb from which they have been computed, so, also based on my experience, these indices should negligibly affect the results.*

We thank the referee for the constructive review and suggestions. We have changed the introduction and the presentation of the results in the revised manuscript. Regarding the penetration of the MW frequencies, while we agree that theoretically a depth of 0.1—2 m should be a more reasonable choice for both 19 GHz and 37 GHz, as we cited in Line 99 of the revised manuscript (Surdyk, 2002; Brucker et al., 2010), our study is also based on the assumption that the frequency ratios should reflect near-surface (0—2 cm) density, as in Champollion et al. (2013) and Leduc-Leballeur et al. (2017). Therefore, as a compromise between the theoretical penetration depth and the aforementioned applications, we switched our FDM data to a depth of 0.12 m (instead of 0.04 m) to perform the experiments in the revised manuscript. But certainly, as the Referee also pointed out in the detailed comments, the relatively reasonable results can be obtained based on indirect correlation of the top layer density with deeper layers which indeed influence the adopted frequency more. This has also been included in our discussion in Line 461 of the revised manuscript.

Regarding the validity of using the derived indices, our study was motivated by Tran et al. (2007) and Champollion et al. (2013). Tran et al. (2007) combined a derived Tb ratio with Tb values to cluster snow facies in both Greenland and Antarctica, and Champollion et al. (2013) could associate frequency ratios to near-surface grain size and density at Dome C, Antarctica to a certain extent. In both studies, the validity of using such ratios exists to an extent that should be interesting to discuss, hence we included them.

Moreover, since we use Random Forest regression, we do not agree that the indices cannot bring additional information or performance. While the principle of information theory indeed suggests that indices derived from the original should not introduce additional information, it's important to consider the context in which certain techniques, such as random forest regression, operate. Random forest regression is a powerful ensemble learning method that harnesses the collective strength of multiple decision trees. In the case of random forest regression, the combination of diverse decision trees allows for the detection

and extraction of intricate patterns and relationships within the data that may not be readily apparent in the original dataset. Each tree contributes its unique perspective, and the ensemble's output is often more robust and accurate than that of an individual tree. Therefore, although the indices derived from the original data may seem, from an information theory standpoint, to contain similar information, the strength of random forest lies in its ability to uncover latent, complex patterns that might not be explicitly present in the raw data. This enables the model to provide more nuanced and accurate predictions, surpassing the limitations of a single decision tree.

However, since our study aims to assess a method and discuss the validity of the parameters, we assume that a sensitivity analysis of using different combinations of parameters could be added, where we use as input:
- All parameters as what we are using now
- Only absolute Tb and sigma0
- Only absolute Tb and sigma0, and derived ratios (as also pointed out by Referee 1)

The comparison has been shown in Appendix B of the revised manuscript. The comparison shows that using all parameters still slightly outperforms the other combinations, so we still adopted its results in Figs. 6—9 in the revised manuscript.

*Detailed comments:*

*Introduction.*

- *The introduction contains a review of the state of the art more than enough to frame this paper. I would only suggest clarifying the aspects related to different spatial resolution, coverage and revisiting when mentioning active and passive MW.*
  We appreciate the recommendation and tried to add the information. It has been added in Lines 46 and 56 in the revised manuscript.

*Section 2*

*Section 2.1.*

- *Equation 1 and 2 are properly referred to the original publications, however a short sentence about the physical principles behind would be useful for the reader.*
  This has been added in the revised manuscript (Lines 114—117).
- *Line 94-95. The dramatic change in emission mechanism due to the presence of liquid water within the ice sheet might be commented, although this point is mentioned later in section 3.2. Same applies to the scattering in section 2.2.*
  This has been added in the revised manuscript (Line 101 and Line 137).

*Section 2.2.*

- *the linear correction for local incidence (LIA) sounds me a bit odd. LIA should be already accounted for when computing NRCS to extract the backscattering (σ°). In any case the backscattering dependence on LIA is not linear at all. Finally, as far as I understand*

*from pag. 5 line 125, at the end you did not use data corrected with eq. 3. Could you further clarify?*

We apologize for not specifying the parameters properly. Equation 3 does not describe how we processed the data, but which kind of dataset we used. The same equation can be found in Lindsley and Long (2010), Eq. 3 on page 3. What we are using is the σ° normalized to the reference angle (40°), referred to as $A$ in this equation. The $A$ products we are using are already available via Brigham Young University (BYU) Microwave Earth Remote Sensing (MERS) laboratory platform and are directly used in our study. However, since $A$ as a single letter could be misleading, we called it σ° again in the following texts of the manuscript, which is more familiar to the common knowledge. We have changed that in the revised manuscript (Lines 131—136).

- *The spatial and temporal co-registration between ASCAT and SSMIS should be better described, this could lead to error and artifacts depending on the processing you applied. At the end, how many co-located Tb and σ° you obtained? It is an important information for better understanding the RF implementation, although something is addressed later.*

  We have tried to clarify the spatial and temporal co-registration between ASCAT and SSMIS better by adding some information on the interpolation methods we used (Lines 139, 157 and 182).

  Eventually, we obtained 19,027 valid pixels within the Antarctic ice sheet range (Table 1 of this document). We admit that with a linear interpolation, artifacts occur at the edge of the images. However, we filtered them out using the coastline from Depoorter et al. (2013). What falls within the range of the Antarctic ice sheet should be reliable.

*Section 2.3*

- *line 135 – 138. As stated in the general comments, the attempt to retrieve density at 4 cm raises a conceptual issue. The top 4 cm layer should be almost transparent not only at C-band but also at Ka band in case of dry firn. I'm wondering if you are obtaining results based on indirect correlation of the top layer density with deeper layers to which microwaves are instead sensitive. No wonders if RF achieves successful retrievals: machine learning can exploit almost any kind of input/output relation, but the risk of finding out something based on apparent relationships is always around the corner. If used as "black boxes", ML could potentially relate newborns in China and weather in USA, but which is the utility? I believe a robust physical justification is needed.*

  We agree that it is likely that we have obtained results based on indirect correlation of the top layer density with deeper layers. We have corrected this throughout the manuscript by using FDM data from 12 cm depth (Please refer to the major comments).

- *Line 138 – 140. The sentence is unclear to me, could you rephrase please. Where was density at 1m depth used later?*

  It was shown in Fig. 4 of the original manuscript to prove that melt events have a prolonged impact on deeper snow densities, hence our clustering step to separate dry and melted pixels was quite reliable. However, we agree that overall it does not have added values to the following analyses, hence removed all comments about the 1 m density in the revised manuscript.

*Section 3*

*Section 3.2.*

- *Is Tb Ratio the same of eq. 1? If so, no need to introduce it again with reference.*
  The Tb ratio from Tran et al. (2008) is not the same as our Eq. 1. However, we agree that this reference is repetitive to the introduction, hence removed it in the revised manuscript.
- *In my understanding, volume decorrelation was not introduced before. The cited work by Rizzoli is using X band SAR, it is not clear if this finding is also valid for radiometric measurements (scattering and emission are complimentary each other)*
  This application was introduced in Line 61 as previous studies as a motivation for applying the unsupervised classification method. We also removed it in Sect. 3.2 of the revised manuscript.
- *Line 195 – 199. The normalization by firn temperature is embedded in both parameters you defined in eq. 1 and 2. Which is therefore the reason for removing the average seasonal Tb signal? And which the one for doing the same with backscattering that is almost insensitive to temperature? Moreover, machine learning techniques as RF can cope with redundant, noisy, and biased data, so dealing with timeseries of measurements or their anomalies should not change much the results. Finally, there is also a concern in merging Tb with their ratios that is commented below.*
  This step (everything in Sect. 3.2) serves to separate dry pixels from pixels that suffered from melt, therefore Tb ratios and RF are not used here. While backscattering is almost insensitive to temperature (although Fraser et al. (2016) showed some correlations), it is very sensitive to melt events and subsequent melt layers, hence the anomalies should be a good indicator of melt pixels. We agree that this motivation has not been clarified in the manuscript, hence we rephrased it as Lines 201—213 of the revised manuscript.
- *Lines 201 – 211. The clustering algorithm should be better explained maybe with a supporting figure/diagram. I don't believe a reader unfamiliar with Ward algorithm can understand this section.*
  This has been implemented in the revised manuscript (Fig. 2 and Lines 222—231).

*Section 3.3*

- *Lines 231 – 246. With "sample" do you refer to the set of temporally and spatially coregistered SSMIS and ASCAT measurements for the given pixel? In my understanding, for both subsets I and II you selected randomly 100 pixels from the 7 clusters over Antarctica described in section 3.2 (that is spatial, 25 km resolution each pixel) and you considered the timeseries of satellite measurements (that is temporal, approx. 1 set of SSMIS + ASCAT measurements per pixel day per 10 years). At the end you should have used 365300 sets for training and the same data amount for testing. In other words, you considered about 125000 Km2 for training and testing and applied the trained RF on the remaining ≈14000000 Km2 of Antarctic surface, which is notable. Maybe some more information could be provided…*

We indeed refer to "the set of temporally and spatially coregistered SSMIS and ASCAT measurements for the given pixel". This has been changed in the revised manuscript (Line 257). But for subsets I and II we selected random pixels from the 4 dry clusters instead of the 7 clusters (Table 1 below). We should also clarify a mistake in the original manuscript that since we use the 10-day resolution IMAU-FDM, the total time slots should be 366 instead of 3653. Therefore, we used approximately 0.6% of the total data for training. We now use 10% of the total data for training (as Dataset I) in the revised manuscript, as is slightly improves the result (Fig. 1 of this document) and is theoretically more reasonable than using 100 pixels (~0.6% of the data).

Table 1. Statistics of pixels per cluster and pixels used for RF estimating in the original manuscript.

| Cluster | Number of pixels | Number of Dataset I pixels | Number of Dataset II pixels |
|---------|------------------|----------------------------|-----------------------------|
| Firn 1 | 4540 | 454 | 26 |
| Firn 2 | 7360 | 736 | 42 |
| Firn 3 | 3465 | 346 | 20 |
| Firn 4 | 2284 | 228 | 12 |
| Firn 5 | 429 | 0 | 0 |
| Firn 6 | 325 | 0 | 0 |
| Firn 7 | 624 | 0 | 0 |
| Total | 19027 | 1764 | 100 |

[Figure]

Figure 1. Comparison between using 100 pixels and 10% of the data for training.

- *Equation 4. The proposed input combination raises another concern: from the information theory, the Tb ratios do not bring to the RF additional information independent of the Tb from which they have been computed, therefore (this is also my*

*personal experience) the results should not be affected by these inputs (or conversely by Tb if you use the ratios). Clarification is needed.*

Please refer to the major comments, where we argue why derived indices can effectively add information when used in Random Forest regression as these are based on decision trees and derived indices can play an important role there (as they might be important in different phases of the decision tree)

- *Line 258. Gini importance should be better referred and briefly commented. Which is the difference with e.g., predictor importance proposed by Breiman?*

This is improved in the revised manuscript by including also predictor importance. Regarding the difference between the Gini importance and the permutation (Breiman) importance (Fig. 6 of the revised manuscript), we notice that using the permutation importance, the ranking of the original horizontal channels goes down. Therefore, we have used both in the revised manuscript.

*Section 4*

- *Section 4.1. Following the comment above, this is the core of my concerns: the scarce correlation with density at 4 cm could be depending on the microwaves' scarce sensitivity to such shallow depth. Also, the reverse correlation along the coasts should be depending on melting not entirely removed that occurs more frequently than in the central part of Antarctica. Again, the physics behind should be analysed.*

We agree. A deeper snow density (12 cm) has been assessed, and the potential melting has been added in the discussion (Lines 417—422).

- *Figure 1. Although referred to in section 4.1, I find this figure poorly informative. My suggestion is to remove or replace with something more meaningful.*

We would like to keep it to give the reader an overview of the parameters we used, including their spatial patterns. However, we moved it to Appendix A instead of using it as in the original manuscript.

- *Figure 2. Did you evaluate the correlation with density at 1 m? At the end which was the role of this parameter in your study?*

We did not evaluate the correlation with density at 1 m. Originally we intended to show that our analysis should serve for multiple depths, however this was not well addressed in the original manuscript. Please note that in the revised manuscript, we opt for assessing the densities at 12 cm depth instead of 4 cm, therefore the descriptions have been revised accordingly.

- *Figure 4. The plots in the figure are quite small and difficult to read. I would suggest revising.*

This has now changed into Fig. 5 in the revised manuscript.

- *Figure 5 left: the scatterplot should refer to the test results (i.e. those obtained on subset II), not to the training results (Subset I). Usually, retrieval scatterplots show the estimated vs. target, not vice-versa. The plot or caption should also cite the statistics and total data amount. Finally, the R value seems even worse than the one of direct correlation with Tb Ku and Ka in figure 2 for most of the pixels. Isn't it? Which is the explanation?*

It was a mistake in the captions. We indeed used Dataset II for this analysis. This figure is changed in the revised manuscript where we also swapped the axes.

However, here the R^2 value (ranging between 0 and 1) refers to the linarity, i.e. if we fit a line to the estimated vs. target scatter plot, how the goodness of fit is. This is not the same indicator as the correlation coefficient (ranging between -1 and 1) in Fig. 3 of the revised manuscript. However, using a linear regression, R^2 is also equivalent to the correlation coefficient squared. For consistency, we switched to the correlation coefficient as the indicator of temporal performance in the revised manuscript. It can be observed that the correlation between IMAU-FDM and RF densities most resembles that between Tb(37V) and IMAU-FDM densities, which matches the importance shown in Fig. 6.

- *Figure 6 with doubled colorbar is difficult to interpret (especially figure 6d). I would suggest revising.*
  This has been changed into different markers (Fig. 7d of the revised manuscript), as also pointed out by Referee 1.

- *Figure 7 why do not also add the Correlation/Determination coefficient maps as those in figure 2? In my view this is more informative than e.g. the 10- years averaged maps of figure 6.*
  We have switched to the correlation coefficient in Fig. 8 of the revised manuscript (please also see comment above). An averaged map in our opinion shows that our method works reasonably well spatially (in contrast to the performance temporally, as shown in the sections afterwards).

**Reference**

Brucker, L., Picard, G., and Fily, M.: Snow grain-size profiles deduced from microwave snow emissivities in Antarctica, Journal of Glaciol ogy, 56, 514–526, https://doi.org/10.3189/002214310792447806, 2010.

Champollion, N., Picard, G., Arnaud, L., Lefebvre, E., and Fily, M.: Hoar crystal development and disappearance at Dome C, Antarctica: observation by near-infrared photography and passive microwave satellite, The Cryosphere, 7, 1247–1262, https://doi.org/10.5194/tc-7-1247-2013, 2013.

Depoorter, M. A., Bamber, J. L., Griggs, J., Lenaerts, J. T. M., Ligtenberg, S. R. M., van den Broeke, M. R., and Moholdt, G.: Synthesized grounding line and ice shelf mask for Antarctica, https://doi.org/10.1594/PANGAEA.819151, supplement to: Depoorter, MA et al. (2013): Calving fluxes and basal melt rates of Antarctic ice shelves. Nature, 502, 89-92, https://doi.org/10.1038/nature12567, 2013.

Fraser, A. D., Nigro, M. A., Ligtenberg, S. R. M., Legrésy, B., Inoue, M., Cassano, J. J., Kuipers Munneke, P., Lenaerts, J. T. M., Young, N. W., Treverrow, A., van den Broeke, M., and Enomoto, H.: Drivers of ASCAT C band backscatter variability in the dry snow zone of Antarctica, Journal of Glaciology, 62, 170–184, https://doi.org/10.1017/jog.2016.29, 2016.

Leduc-Leballeur, M., Picard, G., Macelloni, G., Arnaud, L., Brogioni, M., Mialon, A., and Kerr, Y.: Influence of snow surface properties on L-band brightness temperature at Dome C, Antarctica, Remote Sensing of Environment, 199, 427–436, https://doi.org/https://doi.org/10.1016/j.rse.2017.07.035, 2017.

Lindsley, R. D. and Long, D. G.: Standard BYU ASCAT Land/Ice Image Products, Tech. rep., Brigham Young University Microwave Earth Remote Sensing (MERS) Laboratory, https://www.scp.byu.edu/docs/pdf/MERS1002.pdf, [Access date: Oct. 12, 2023], 2010.

Surdyk, S.: Using microwave brightness temperature to detect short-term surface air temperature changes in Antarctica: An analytical approach, Remote Sensing of Environment, 80, 256–271, https://doi.org/10.1016/s0034-4257(01)00308-x, 2002.

Tran, N., Remy, F., Feng, H., and Femenias, P.: Snow Facies Over Ice Sheets Derived From Envisat Active and Passive Observations, IEEE Transactions on Geoscience and Remote Sensing, 46, 3694–3708, https://doi.org/10.1109/tgrs.2008.2000818, 2008.

---

## Referee Report (RR1)

This paper used machine learning (ML) and satellite microwave data to examine Antarctic firn density. The authors did a good job of responding to all comments in the first two round of review. I only have one comments.

The addition of Appendix A with SMRT simulation is good except the claims that C-band data is sensitive to changes in snow layers up to 20 m. The change in dB reported in the graph are negligeable. A change in 0.1 dB is not. The usual radiometric uncertainty is around 1 dB (Schmidt et al. 2018), everything under that can be noise. I would remove all the statements that C-band is sensitive to snow layers up to 20 m.

The C-band signal (useful in the density RF model) is probably not sensitive to volume scattering like the passive data but is sensitive to change in snow surface roughness condition (wind) and snow permittivity which are both link to snow surface density. This could be shown by using a surface scattering model like IEM in SMRT.

Schmidt, K., Tous Ramon, N., and Schwerdt, M.: Radiometric Accuracy and Stability of Sentinel-1A Determined Using Point Targets, International Journal of Microwave and Wireless Technologies, 10, 538–546, https://doi.org/10.1017/S1759078718000016, 2018

---

## Author Response (AR2)

**Response to Referees on egusphere-2023-1556**
We appreciate the reviews and comments from both Referees. Please find the response to Referee 2 on pages 1-8, and the response to Referee 3 on pages 9-10.

**Response to Referee 1 on egusphere-2023-1556**

First, we would like to thank the Referee for reviewing and commenting on the manuscript, which will improve the quality of the manuscript. Please find the item-by-item reply below, with the original comments in *italics* and the responses in blue. All the suggested changes are implemented in the revised manuscript.

*The authors provided detailed replies to my comments, thank you. The manuscript has been significantly modified, and most contents improved. However, some of the issues pointed out in the previous round still hold after the modifications.*

*The main concern is still related to the investigated depth. Although it has been increased from the former 4 cm, 12 cm is still a too tiny layer especially at C- band - see e.g. Surdyk 2002 (10.1016/S0034-4257(01)00308-X), Picard et al. 2009 (10.3189/002214309788816678), Champollion et al. 2019 (10.5194/tc-13-1215-2019), Brucker et al., 2009 (10.3189/002214310792447806). I'm still convinced you should provide a physical explanation supporting this choice.*

We understand the concern of the reviewer, and agree that indeed the penetration depths can exceed 1 m for 19 GHz and for C-band, as mentioned by e.g. Surdyk (2002) and Fraser et al. (2016). Meanwhile, such ranges of penetration depths are not always certain, e.g. in Picard et al. (2009), *"the deepest penetrations at the 19V channel are located in Marie Byrd Land (4–7 m) and on the East Antarctic divide (4–6 m). The shallowest penetrations are found in the wind-glazed surface regions and megadunes, with values as low as 0.3 m. Intermediate values are found in Wilkes Land between the coast and the divide (2.5–5 m)."* Furthermore, according to Arndt and Haas (2019), although the penetration depth of C-band exceeds 1 m, *"however, increased backscatter along the propagation path through the snow at any depth will result in the observed overall backscatter increases,"* and according to Cartwright et al. (2022), *"azimuthal anisotropy arises primarily due to the interaction between the incident microwave radiation and regularly aligned roughness (on the Rayleigh roughness scale, or larger) of the surface and subsurfaces within the penetration depth (Ulaby et al., 1996; Bingham and Drinkwater, 2000; Partington and Flach, 2003; Yurchak, 2009; Fraser et al., 2016),"* therefore, the shallower depth firn properties should not be completely negligible to long-wavelength microwave.

With the aforementioned reference, we understand it is true that ASCAT in principle is sensitive to the scattering properties up to several metres' depths, as that is indeed the estimated penetration of the C-band wavelength. However, there is no evidence that the relationship between the density at those depths and the C-band radar backscatter is a strong one; although a few modelling efforts have been carried out in that direction, there is no literature showing that the physical modelling is mature for active microwave sensing. Therefore, the contribution of C-band radars to the retrieval of snow properties at a wide range of depths remains an interesting aspect to investigate, especially through machine learning techniques which are more data-driven, including assimilation with other sensors.

We applied the approach conducted at 12 cm also to the snow density at other depths, and added the figure to this document (Fig. R1). The figure shows that both the RMSE and correlation coefficient reduce with an increasing depth. We believe that this is due to the fact that the firn density at larger depths is not largely influenced by surface temperature and precipitation, which have a larger impact on temporal variations of microwave signals.

[Figure]

Figure 1. RMSE (left) and correlation coefficients (right) at different depths (12 cm, 40 cm and 1 m, respectively).

Furthermore, we also understand the concern that the microwave signals can be affected by layers that are at a larger depth (i.e. even larger than 1 m). Therefore, we computed the mean temporal correlation coefficients between densities at different depths and satellite parameters, summarised in Table R1. The table shows that within our available dataset, the correlation between all satellite parameters and densities on average reaches the maximum at 40 cm depth. It is then worth noting that the correlation first drastically decreases (by 40 %) between 40 cm and 1 m depths for 37 GHz, and then largely decreases (by 21 %) between 1 m and 2 m for 19 GHz. Therefore, we have adopted the density from 40 cm depth in the

revised manuscript, and added Table R1 to the revised manuscript. It is important to note that since all depths within the penetration depth affect the scattering properties, the "density at depth…" should be the mean density of the upper x depths (x = 12 cm, 40 cm, 1m, etc.).

*Table R1. Correlation coefficients between IMAU-FDM densities at different depths and satellite parameters.*

|        | TB(19V) | TB(19H) | TB(37V) | TB(37H) | sigma^0 |
|--------|---------|---------|---------|---------|---------|
| 12 cm  | 0.19    | 0.18    | 0.20    | 0.20    | -0.05   |
| 40 cm  | 0.24    | 0.23    | 0.20    | 0.19    | -0.06   |
| 1 m    | 0.23    | 0.20    | 0.12    | 0.12    | -0.06   |
| 2 m    | 0.18    | 0.12    | 0.03    | 0.02    | -0.06   |
| 5 m    | 0.08    | 0.02    | -0.07   | -0.08   | -0.04   |
| 10 m   | 0.05    | 0.01    | -0.07   | -0.07   | -0.03   |

*Also, the concern about redundant information in coupling microwave observations with their combinations in the RF retrieval, according to information theory, has not been solved: my point is confirmed by the predictor importance analysis in fig.6 that clearly shows the dominant role of direct observations and the minor role of derived indices.*

The original purpose of using the brightness temperature ratios was to reproduce the Champollion et al. (2013) study, where the ratios could be related to near-surface hoar-crystal formation. But it is also true that we could not reproduce the method in our study, especially because it was not applicable to the entire Antarctic ice sheet. Hence, we agree to remove the derived parameters. The revised manuscript is now based on the new setting.

*Another concern is about the SSM/I derived indices FR and PR: beside the reason for using the formulation in eq. 1 and 2 instead of the other ratio generally adopted (e.g. Kelly et al., 2003; Tedesco et al., 2004, Chang et al., 1987; Chang et al., 1990; Santi et al. 2012), the PR correlation with the target parameter does not seem exceptional (especially at Ku band) and the reasons because it is sometimes positive and sometimes negative quite is difficult to explain because it does not seem related to environmental factors as for the other observables shown in the figure.*
We have removed the ratios in the revised manuscript.

*I believe further revision should be done to clarify the compensation for observation angle: σ° is universally adopted to refer to backscattering, which is derived from the NRCS that already accounts for incident angle. Please also mention backscattering when introducing σ° notation.*
It is true that $\sigma^0$ already includes the normalisation by the area of the cell resolution. However, this extra normalisation done in the BYU product also accounts for the differences in the physical response of the distributed target per metre square which is a function of the incidence angle. This is also documented in the methodology paper of Long and Drinkwater (2000):"Because scatterometers make measurements over a range of incidence angles, the incidence angle dependence of $\sigma^0$ must be accounted for." To avoid confusion while being coherent with the $A$ parameter as defined in Long and Drinkwater (2000) and used by Fraser et al. (2016), we have changed the $\sigma^0$ into $\sigma_A^0$ in the revised manuscript.

*Total data, training and test datasets still need to be clearly quantified: I did not find the numbers as declared in the authors replies (possibly my fault?). At lines 264-269 it is stated*

*that Subset I contains 10% of the non-melting pixels (how many in total?) and Subset II is composed of 100 pixels (only?). Nothing is declared about independence of the two datasets. These numbers seem significantly smaller than my deductions for the previous round and this could lead to some over-dimensioning of RF parametrization shown in Table 1 (and consequent overfitting) Moreover they do not seem consistent with the amount of data arguable from fig. 6 left and even insufficient for generating the maps in figure 3 and 8. I suspect some misunderstanding.*

Regarding this problem, we agree that the numbers declared in the previous replies have not been added to the manuscript, and now we have added the numbers in the revised version. We would like to clarify that each pixel we use consists of time series of all parameters, i.e. 366 density estimations. Therefore, Subset I used for hyperparameter tuning and training consists of 1748*366 samples (instead of 1748 points), and Subset II used for testing and importance computation, i.e. Fig. 6 consists of 100*366 samples (instead of 100 points). We have double-checked that Subset I and Subset II do not have overlapping pixels. Regarding the amount of data in Fig. 6, we presented the testing result using Subset II, which consists of 100*366 samples. This is because the entire dry-snow zones in Antarctica according to our clustering method consists of 17478 pixels, and subsequently 17478*366 samples, which is too large and redundant for visualising in a scatter plot such as Fig. 6. However, for Fig. 7 and Fig. 8, we used the densities of the 17478*366 samples, and calculated the resulting mean densities and errors. The amount of data has been added to the revised manuscript lines 253—258.

*Table I. In my personal experience, increasing the number of trees above 50 does heavily affect the computational cost without providing accuracy improvements. But this is just my experience, not absolute truth. In any case the most important thing to assess is the RF dimensioning in terms of training data amount (see comment above).*

Please refer to the comment above. We believe that by using 1764*366 samples as training data, the over dimensioning problem should be resolved.

*Figure 1 is useful addition, in my view however, it is difficult to understand in the current implementation: I would suggest revising and simplify.*

This has been changed in the revised manuscript (also with the ratios removed).

*Evaluating quantitatively the results in figure2 with respect to the corresponding figure in the former manuscript is not straightforward, however it seems that, by comparing against 12 cm rather than 4 cm, some small improvements are obtained at Ku and Ka band while the C- band does not show appreciable improvements. This could go in the right direction by supporting the concern about the insufficient depth. As requested in the previous round, it would be important to provide overall correlation or determination coefficients (at least for each cluster - choice is up to you, but you should be consistent through the manuscript) also to understand which is the contribution of RF with respect to the direct correlation between single observables and target parameters. The physical reasons supporting changes from positive to negative correlations should be better discussed in any case.*

The correlation coefficients have been added to the revised manuscript. We have also attached a comparison between RF and a simple linear regression (Fig. 7 and lines 353—355).

*Figure 5 seems a bit redundant and its informative content not exceptional since the behaviours are difficult to interpret; moreover, figure 6 points out the minor contribution of these parameters in the retrieval.*

Figure 5 was originally provided with the purpose of demonstrating how complicated the relationship between satellite parameters and densities can be for dry snow, and that our clustering method could distinguish melt moments and the spatial coverage of melts. However, we agree that the informative content is not exceptional, hence will move it to appendix in case some readers may be curious about how the distinction of melt regions looks like. We also agree that these parameters are not contributive to the RF approach, hence they will be removed from the RF sections.

*Figure 6 left. The scatterplot seems slightly improved wrt the former result at 4 cm, however I still see some saturation in the retrieved density: in my experience this could depend on not proper dimensioning/training of RF, any explanation? Again, how many data in the scatterplot? The requested correlation coefficient is not provided in the figure/caption.*

The number of training data is 1748*366=639,768, and the number of testing data visualised in the scatterplot is 100*366 = 36,600. This information has been added to the revised manuscript.

Regarding the saturation in the retrieved density, we noticed that not only does RF largely underestimate the density higher than 410 $kg\ m^3$, it also overestimates the density lower than 325 $kg\ m^3$. Therefore, we presume that both the highest and the lowest IMAU-FDM densities are not properly accounted for, both due to uncertainties from the IMAU-FDM modelling process and the limitation of the combination of satellite parameters. An example is shown in Fig. R2 of this document. The scatterplot in the upper panel is calculated as the temporally averaged IMAU-FDM density subtracted by the temporally averaged RF density at the 100 sample pixels. On average, the largest underestimation for RF (exceeding 20 $kg\ m^3$) occurs in regions where the summer wind velocity is more than 2.5 $m\ s^{-1}$ lower than the winter wind velocity. This corresponds to part of our conclusion that IMAU-FDM shows pronounced different behaviours from the satellite time series when the seasonal wind velocity difference is high, hence IMAU-FDM may not adequately capture the actual physical meteorological phenomena that affect microwave scattering properties. On the other hand, RF on average largely overestimates the density in Transantarctic Mountains, potentially due to the complex terrain that affect the surface scattering of the microwave (instead of volume scattering). Finally, an overall limitation of using purely satellite data time series is that they are largely dependent on surface and near-surface temperature. Our study is therefore coherent with the Fraser et al. (2016) study, who could establish a relationship between long-term mean ASCAT backscatter and snow or climate properties for Antarctic dry snow, although our work focuses on establishing a relationship the other way around, i.e. re-constructing firn density using a combination of satellite data. However, the seasonal correlation is compromised both in the Fraser et al. (2016) study and in our study.

[Figure]

*Figure R2. Temporal mean difference between the IMAU-FDM density and RF density at the 100 sample pixels, overlayed on the seasonal wind velocity difference map (upper); scatterplot of the temporal mean difference between the IMAU-FDM density and RF density at the 100 sample pixels versus the seasonal wind velocity at the sample pixels (middle); and temporal mean difference between the IMAU-FDM density and RF density at all pixels in dry snow zones versus seasonal wind velocity difference, coloured by the density distribution of points (lower).*

*Figure 6 right. thanks for the explanation about the Breiman vs. Gini predictor importance analysis, however showing one or another histogram is enough, also because they bring some contradictory results that are difficult to justify, up to you....*

We have removed the ratios and anomalies and kept only the Gini predictor.

*The comparison with the Dome-C data from Leduc is relevant as validation against independent data. However, the data refer to the first 2/3 cm depth and the RF has been trained for 12 cm depth…. Please further address.*

This comparison was performed between the first 2 cm depth of field measurements and the 4 cm depth of IMAU-FDM. Then, both 4 cm and 12 cm IMAU-FDM densities are provided to show the large discrepancies between the model and the field measurement. Furthermore, the Dome C data from Leduc-Leballeur et al. was used also to analyse the potential correlation between near-surface hoar-crystal formation and disappearance and polarisation ratios. This is not applicable in our study anymore hence has been removed.

*Figure 8 and discussion. What I would see addressed (see my comment in the previous round), is the comparison between the R coefficients shown here and those of single observables in figure 3, with the aim of pointing out the improvement brought by RF with respect of attempting the direct retrieval from single observables, that in some cases already reach very high correlation.*

This has been added to the revised manuscript (Fig 7). However, we also show that RF outperforms the simple linear regression in terms of RMSE, which is an important indicator apart from the correlation coefficient.

*Figure 9 and discussion. The rationale of showing the PR at Dome C after showing the overall RF performances as R and RMSE maps of the entire Antarctica is unclear to me. PR is just one of the inputs of the RF algorithm and not even one of the most important.*

Originally, we referred to the Champollion et al. study where they attributed the variation in polarisation ratios to the hoar-crystal disappearance, which is characterised by an increase in near-surface density and a reduction in grain size. We understand this inclusion of the Champollion et al. study causes confusion, hence removed this part in the revised manuscript.

**Reference**

Arndt, S. and Haas, C.: Spatiotemporal variability and decadal trends of snowmelt processes on Antarctic sea ice observed by satellite scatterometers, The Cryosphere, 13, 1943–1958, https://doi.org/10.5194/tc-13-1943-2019, 2019.

Bingham, A. W. and Drinkwater, M. R.: Recent changes in the microwave scattering properties of the Antarctic ice sheet, IEEE T. Geosci. Remote, 38, 1810–1820, https://doi.org/10.1109/36.851765, 2000.

Cartwright, J., Fraser, A. D., and Porter-Smith, R.: Polar maps of C-band backscatter parameters from the Advanced Scatterometer, Earth Syst. Sci. Data, 14, 479–490, https://doi.org/10.5194/essd-14-479-2022, 2022.

Champollion, N., Picard, G., Arnaud, L., Lefebvre, E., and Fily, M.: Hoar crystal development and disappearance at Dome C, Antarctica: observation by near-infrared photography and passive microwave satellite, The Cryosphere, 7, 1247–1262, https://doi.org/10.5194/tc-7-1247-2013, 2013.

Fraser, A. D., Nigro, M. A., Ligtenberg, S. R. M., Legrésy, B., Inoue, M., Cassano, J. J., Kuipers Munneke, P., Lenaerts, J. T. M., Young, N. W., Treverrow, A., van den Broeke, M., and Enomoto, H.: Drivers of ASCAT C band backscatter variability in the dry snow zone of Antarctica, Journal of Glaciology, 62, 170–184, https://doi.org/10.1017/jog.2016.29, 2016.

Long, D. and Drinkwater, M.: Azimuth variation in microwave scatterometer and radiometer data over Antarctica, IEEE Transactions on Geoscience and Remote Sensing, 38, 1857–1870, https://doi.org/10.1109/36.851769, 2000.

Partington, K. and Flach, D.: Synergetic Use of Remote Sensing Data in Ice Sheet Snow Accumulation and Topographic Change Estimates: Comparison of model output with available data, Tech. Rep. NOV-3137-NT-1537, Noveltis, Vexcel UK and Legos, Ramonville-Saint-Agne, France, 2003.

Picard, G., Brucker, L., Fily, M., Gallée, H., and Krinner, G.: Modeling time series of microwave brightness temperature in Antarctica, Journal of Glaciology, 55, 537–551, https://doi.org/10.3189/002214309788816678, 2009.

Surdyk, S.: Using microwave brightness temperature to detect short-term surface air temperature changes in Antarctica: An analytical approach, Remote Sensing of Environment, 80, 256–271, https://doi.org/10.1016/s0034-4257(01)00308-x, 2002.

Ulaby, F. T., Siquera, P., Nashashibi, A., and Sarabandi, K.: Semi-empirical model for radar backscatter from snow at 35 and 95 GHz, IEEE T. Geosci. Remote, 34, 1059–1065, https://doi.org/10.1109/36.536521, 1996.

Yurchak, B.: Some Features of the Volume Component of Radar Backscatter from Thick and Dry Snow Cover, in: Advances in Geoscience and Remote Sensing, edited by: Jedlovec, G., Intech, Rijeka, Croatia, https://doi.org/10.5772/8339, 2009.

**Response to Referee 3 on egusphere-2023-1556**

First, we would like to thank the Referee for reviewing and commenting on the manuscript, which will improve the quality of the manuscript. Please find the item-by-item reply below, with the original comments in *italics* and the responses in blue. All the suggested changes are implemented in the revised manuscript.

*This paper used machine learning (ML) and satellite microwave data to examine Antarctic firn density. The authors did a good job of responding to all comments in the first round of review. I only have minor comments.*

*Across the document, the notation for figure should be normalized (Figure, fig, or fig.). Should be Fig. everywhere.*
We have corrected the manuscript for consistency. However, according to the guidelines of The Cryosphere (https://www.the-cryosphere.net/submission.html, Figures & tables section), we did not change everything:
"The abbreviation 'Fig.' should be used when it appears in running text and should be followed by a number unless it comes at the beginning of a sentence, e.g.: 'The results are depicted in Fig. 5. Figure 9 reveals that...'."

*Line 74: Remove on. "Antarctic ice sheet based on on daily"*
This has been corrected in the revised manuscript.

*Line 155: What about the penetration depth of C-band? The Sigma0 at C-band is sensitive to deeper than this.*
We conducted an experiment computing the temporal correlation between satellite observations and IMAU-FDM densities (Table 2 of the revised manuscript), and determined that the optimal depth where the density can be correlated with all frequencies should be 40 cm. We also refer to the Fraser et al. (2016) study where the top 1 m density was seen as the layer most sensitive to atmospheric drivers which can be correlated with C-band backscatter, despite a much larger penetration depth of C-band radar.

*Line 174: replace "parameterization" to "parametrization"*
This has been corrected (and other identical errors) in the revised manuscript.

*Line 208: Is it spatial or time clustering? Or both? You talked about spatial and then…*
It is a clustering of time series. In doing this, the regions which experienced intensive melt and subsequent ice-layer formation can be distinguished. To avoid confusion, the previous "spatial" has been removed.

*Line 294: I suggest talking about biased with correlated features since all your feature are highly correlated. In your case, all measurements at 19 (Tb, Pr, Fr, and anomalies) are correlated with each other, same for 37. This will affect the feature importance. I would just mention how it would affect it.*
It is true, and we have removed the radiometer-derived ratios and anomalies from the RF approach, also as they do not contribute to the results.

*Figure 5: Y axis are never the same. It's hard to distinguish differences between each cluster.*
It is rather difficult to completely normalise the y-axes, because doing this would make all dry-snow regions appear as a straight line. We have therefore used a same range of y-axis for dry regions, and another range of y-axis for melt regions. Due to the suggestion of Referee 2, we have moved this figure to Appendix B.

*Line 358-359: Again correlation between derived sat (PR, FR) and the Tb at 19,37 is probably high. So the importance of Pr and Fr is reduced. Most of the information contain in Pr is probably also in the Tb.*
They have been removed from the RF approach.

*Figure 7: the caption is missing a parenthesis at the end.*
This has been corrected in the revised manuscript.

*Line 393: Missing a come between high and close? "the RMSE between IMAU-FDM and RF is high close to Vinson Massi".*
This has been corrected in the revised manuscript.

*Figure 10: Can you change the bright yellow color? The label is unreadable.*
Since the comparison of density variations with polarisation ratios is not valid anymore, this figure has been removed in the revised manuscript.

*Line 413: I think that is already well established when you look at snow density, microwave data and radiative transfer... (Brucker et al 2010, 2011 . Picard et al 2009, 2014). I would reword…*
Agreed. "Our findings reveal" has been changed to "out study is based on" in the revised manuscript.

*Line 454: Grain size also influence Tb. Perhaps some words on how it affect the final result?*
It is true and is the main reason why we assumed that a non-linear machine learning model could account for the effect from grain size and other drivers. However, due to the lack of information on grain size, we cannot arrive at a better (or more solid) conclusion regarding how it affects the final result.

---

## Author Response (AR3)

**Response to Referees on egusphere-2023-1556**

First of all, we would like to thank Referee 3 for the approval of the manuscript. Since there is no comment from Referee 3, this document only includes the response to Referee 2. Please find the item-by-item reply below, with the original comments in *italics* and the responses in blue. All the suggested changes are implemented in the revised manuscript and we will refer to the different versions of our manuscript as V0 (original submission, 24 Jul 2023), V1 (first revision, 26 Jan 2024) V2 (second revision, 4 May 2024) and V3 (current submission, Aug 2024)

*I'm slightly concerned about the way the authors replied to my comments. Several contents on which I have asked for clarification have been removed, and the depth involved in the analysis has been increased for the third time, based on a questionable search of the maximum correlation between satellite observation and target parameter that neglects the depth dependence on frequency. The rationale behind this criterion is also unclear because the authors themselves stated "the relationship between satellite parameters and firn density is complex, and simple linear relationships may not adequately describe the IMAU-FDM density based on different satellite parameters…".*

*Despite the pivotal changes from one revision to another, the authors obtain almost the same good results; however, without physical support and interpretation, their findings could also depend on issues in the approach, in the data organization or in the ML implementation (overfitting?). In other words, although I am favourable to the "data driven" approaches, method and results must be placed in a robust physical framework, which seems lacking here. Considering the interesting topic, I'm giving another major, in the hope the authors will provide better support for their findings.*

First of all, we appreciate that the referee still finds the topic interesting. We would like to clarify our ideas in the following points:

1. Regarding the choice of the assess depth.

To avoid misunderstandings, we believe that the "physical support and interpretation" consists of two different aspects. One is whether C-band radar can observe near-surface (e.g. < 1 m depth) firn properties at all, and the other is why we changed the depths for the experiment.

Regarding the first aspect, whether C-band can observe near surface properties, we rely on a set of arguments that should support the physical interpretation in this response. The first argument is based on a new analysis in the revised V3 of the paper (added to Appendix A), where we performed a simple sensitivity analysis using the Snow Microwave Radiative Transfer (SMRT) model to understand the penetration depth of different sensors. For this experiment, we use a multi-layer snowpack of 20 m depth where the thickness of each layer is set to 40 cm. In this analysis, we perform two experiments. In the first experiment, we run SMRT for 3 locations with in situ measurements of temperature, density and grain size following Larue et al. (2021). All layers have the same firn properties in this experiment. Subsequently, in a second experiment, we disturb for each layer at a time the observed firn properties, in order to observe the impact of altering firn properties (i.e. density $+ 50 \ kg \ m^{-3}$, grain size + 0.5mm) on different depths. The results of the sensitivity analysis (Fig. R1 below and Figure A1 in appendix of the revised manuscript V3) show that the sensitivity of both C-band backscatter and brightness temperature decreases with an increasing depth. C-band can

be sensitive to density and grain size changes at more than 20m depths, whilst 19 GHz and 37 GHz are sensitive up to 6-10 m and 0.8-1 m, respectively.

This sensitivity analysis confirms the argument raised by the reviewer (i.e. deep penetration of C-band), but also confirms that, given the high sensitivity to top layers, C-band can be used to assess surface firn properties as it is definitely not transparent. It is as such in line with arguments made in other studies and that we used in previous revisions to argue for using C-band to assess surface firn properties (e.g. the study of Fraser et al. (2016) who only used the mean firn density of upper 1 m in dry zones of Antarctica to derive the relationship with C-band backscatter or Tran et al. (2008) who related backscatter intensities from Ku-band (13.575 GHz) and S-band (3.2 GHz) radar altimeter (with a theoretical penetration depth of over 10 m; Remy et al., 2015) and brightness temperatures from 23.8 GHz and 36.5 GHz to surface conditions over Antarctic firn).

[Figure]

*Figure R1. Sensitivity analysis using SMRT*

Regarding the second aspect, namely the change of depths throughout the different versions of the manuscript, we would like to again use SMRT sensitivity experiment in combination with the depth correlation of density to explain the good results for different depths.

First, the SMRT sensitivity experiment namely shows that both radar and radiometer are sensitive to a range of depths (e.g. C-band can be sensitive to density and grain size changes at more than 20m depths, whilst 19 GHz and 37 GHz are sensitive up to 6-10 m and 0.8-1 m, respectively). Therefore, we believe that adopting the depths within 0.8 m should physically make sense.

Second, we want to stress that at near-surface depths, variations in density are typically highly correlated (e.g. if the density at 4 or 12 cm depth is higher/lower it is probably also higher/lower at 40cm or 1 m). To illustrate that, we computed the overall correlation coefficients between 4 cm density and densities at 12 cm, 40 cm, 1 m and 5 m depths from IMAU-FDM. The obtained mean correlation coefficients are 1.00, 0.73, 0.36 and 0.10, respectively. This assessment is also added to Appendix D of the revised manuscript V3.

Both these arguments (i.e. large satellite sensitivity up 80 cm and correlated density in the first meter) explain why we obtained similarly good results at depths of 4 (V0), 12 (V1) and 40 (V2) cm. Moreover, these arguments also hint at the fact that the methodology would not work at depths larger than 80 cm. Therefore, we set up another experiment in the revised V3 of the paper (added to Appendix D), to observe whether increasing the depth largely changes the results. Interestingly, as Fig. R2 of this document (or Fig. D1 in appendix of V3) shows, the temporal correspondence between RF-derived densities and IMAU-FDM densities in terms of correlation coefficient indeed becomes much lower when we increase the assessed depth from 12 cm to 1 m, especially for megadune regions where the theoretical penetration depth of 19GHz is 0.3 m according to Picard et al. (2009). Finally, at 5 m depth, the method completely loses the ability to match the temporal pattern of IMAU-FDM density, which proves that our approach is not affected by overfitting.

[Figure]

*Figure R2. RMSE (upper) and correlation coefficients (lower) at different depths (12 cm, 40 cm 1 m, and 5 m, respectively).*

Finally, we think that the sensitivity analysis and density depth correlation analysis provides physical support of why the method works (i.e. sensitive to firn variations in the upper meter),

why it works well for different depths in the different manuscript versions (i.e. sensitive to firn variations over a range of depths and highly correlated densities at different depths in the first meter) and why it does not work for larger depths (i.e. density at deeper depths not related to near-surface density which dominates the signal).

This combination of arguments has led us to an approach in V2+V3 where we search for the optimal depth with the maximum correlation between satellite observation and target parameter that neglects. We acknowledge that this optimal depth might be different for different locations (see also Fig R2/D1), but we decided to choose one common final depth throughout the manuscript.

2. *"the depth involved in the analysis has been increased for the third time, based on a questionable search of the maximum correlation between satellite observation and target parameter that neglects the depth dependence on frequency"*

We would like to refer to the Referee's previous comment: *"it seems that, by comparing against 12 cm rather than 4 cm, some small improvements are obtained at Ku and Ka band while the C-band does not show appreciable improvements. This could go in the right direction by supporting the concern about the insufficient depth. As requested in the previous round, it would be important to provide overall correlation or determination coefficients,"* Note that here "Figure 2" refers to the revised manuscript V1. Following this discussion, we assessed the correlation between IMAU-FDM at different depths and every satellite parameter, as shown in Table 2 of the revised manuscript V2 and V3. This assessment indicated that increasing the assessed depth does not necessarily result in improvements, as suggested by the previous comments. Rather, once we apply this comparison, 40 cm returns the overall highest correlation, hence seems to be the optimal choice. Figure R1 also shows that, C-band is most sensitive to the upper firn rather than e.g. at 20 m depth.

We agree that our method does not explicitly account for different penetration depths which vary per location and sensor (Picard et al. 2009). However, the newly added sensitivity analysis (Appendix A) and correlation analysis shows that the analysis for different depths below one meter are all physically explainable (i.e. strong sensitivity) and could potentially be interchanged (i.e. highly correlated density at different depths). Similar to the previous response, we acknowledge that this optimal depth might be different for different locations (Picard et al. 2009; see also Fig. R2/D1), but we decided to choose one common final depth throughout the manuscript for clarity.

However, we appreciate the Referee for pointing out that the reasoning may not be clear. Therefore, we performed the SMRT experiment and added it to Appendix A of the revised manuscript V3. We have also added the following motivation on Line 189 of the revised manuscript:
*"The reason for this comparison is that, although the theoretical penetration depth can be larger than 20 m for C-band in Antarctic dry firn (Rott et al., 1993), the surface conditions such as temperature, wind and precipitation have more impact on shallow depth of the firn layer, as well as on the satellite parameters (Tran et al., 2008; Picard et al., 2012; Champollion et al., 2013; Fraser et al., 2016). By calculating the correlation coefficients between IMAU-FDM densities and satellite parameters, we need to understand at which depth the densities cannot*

*be affected by the surface conditions. We also need to estimate a depth threshold from which 37 GHz cannot penetrate the firn layers hence cannot provide information on spatial and temporal variation of firn from this experiment, as the penetration ability reduces with an increasing frequency (Rott et al., 1993; Surdyk, 2002). Finally, the density at the depth where the best overall correlation between satellite observations and density time series is adopted for the RF experiment."*

We also added the following discussion on Line 384 of the revised manuscript, with the hope of showing the reasoning and conclusion of this comparison more clearly:
*"Despite a theoretical impact of surface climate conditions such as temperature, wind and precipitation on both satellite parameters and firn density at a shallow depth (Fraser et al., 2016), the lack of a consistent linear relationship was evident in the examination of the individual satellite observations, as the highest mean temporal correlation between satellite observations and the 40 cm IMAU-FDM firn density is 0.24."*

We then added the discussion regarding the limitation of the depth to be studied on Line 413:
*"Finally, our combination of satellite parameters cannot be used to assess densities at depths deeper than approximately 80 cm. This limitation is first because of the theoretical penetration depth as shown in Appendix A: a depth exceeding 80 cm is physically not meaningful for the 37 GHz microwave. Another reason for this limitation is that our study is based on the assumption that the surface climate conditions can affect both shallow-depth firn densities and satellite parameters simultaneously (Fraser et al., 2016). Firn densities at larger depth are not largely affected by surface conditions, hence our combination of satellite parameters is not applicable, even if 19 GHz and C-band microwave have a theoretical penetration depth larger than 5 m (as shown in Appendix D)."*

Finally, we added the following discussion on Line 462, to clarify that our study only indicates the shallow firn densities that can be driven by climate properties, instead of showing the actual scattering mechanism:
*"Our study is also mainly limited to firn densities at shallow depths where the climate phenomena have a large impact; it cannot indicate the actual scattering of firn grains, as a more complicated mechanism persists (Picard et al., 2022)."*

3. *"Despite the pivotal changes from one revision to another, the authors obtain almost the same good results."*

See earlier response where we show that sensitivity is similar for a range of depths and that density variations over depth are highly correlated in the upper layers explaining "the same good results" for different depths in different revisions.

4. Removal of contents.
We agree that the manuscript has changed a lot based on the comments of the previous revision rounds. However, we want to stress that this is the result of demanded changes by the reviewers. Below we explain these changes again shortly as the motivation for these changes might have not been very clear for the reviewer.

a. The polarisation and frequency ratios

Hereby we would like to quote the first version of the Referee's comment: *"based on the information theory, the indices should not bring any additional information independent of the Tb from which they have been computed, so, also based on my experience, these indices should negligibly affect the results."* Referee 3 mentioned the same point: *"Again correlation between derived sat (PR, FR) and the Tb at 19,37 is probably high. So the importance of Pr and Fr is reduced. Most of the information contain in Pr is probably also in the Tb."* Furthermore, we adopted the ratios originally with the hope of including the findings of Champollion et al. (2013), where a relationship between hoar-crystal disappearance (characterised by an increase in 2 cm firn density) and polarisation ratios could be established at Dome C in some moments; this cannot be established in our study using any depth of modelled firn density, hence the physical base of using the Tb ratios is lacking and we removed the parameters from the newest experiment.

b. The Tb and sigma-0 anomalies
We quote the previous version of the Referee's comment: *"Figure 5 seems a bit redundant and its informative content not exceptional since the behaviours are difficult to interpret; moreover, figure 6 points out the minor contribution of these parameters in the retrieval."* We agreed that the anomalies do not contribute to the RF process, therefore removed them in the newest experiment as well.

c. The in situ densities at Dome C
According to the previous version of the Referee's comment: *"The comparison with the Dome-C data from Leduc is relevant as validation against independent data. However, the data refer to the first 2/3 cm depth and the RF has been trained for 12 cm depth…. Please further address."* Originally, we compared the 2 cm in situ density measurements with the modelled 4 cm measurements (as this is the finest resolution available from IMAU-FDM outputs), and showed the potential limitation in the IMAU-FDM densities in modelling local and temporary surface variations due to the simplification of wind patterns in the model. Such bias can potentially propagate towards 12 cm. However, we noticed that the inclusion of 2 cm in situ data (which was not provided by us), the 4 cm modelled density and the 12 cm modelled density (both provided by IMAU-FDM) can cause confusion. In addition, this figure once more intended to reproduce the Champollion et al. (2013) study by observing the trends of polarisation ratios versus surface density. Since we do not consider either surface density or polarisation ratio anymore, we decided to remove this comparison.

With all the reasoning above, we cannot agree that the contents that have been removed need further clarification. However, we appreciate the suggestion of the Referee in the previous comments to understand better the usage of Tb-derived indices for future studies. They should be helpful both in understanding the depth of firn and in assessing firn property variations. Therefore, we have added the following discussion on Line 450 of the revised manuscript:
*"Finally, our study only demonstrated a simple approach in understanding the long-term correlation between firn density and satellite parameters, based on climate conditions that potentially affect them (Fraser et al., 2016). However, due to the different penetration abilities of different microwave frequencies (Surdyk, 2002), future research can benefit from a more quantitative assessment regarding to what extent the penetration depths and other climate parameters affect the results. Better parametrisation of satellite observations which can*

*indicate the variation of firn depth (Santi et al., 2012a; Michel et al., 2014) as well as surface and depth hoar-crystal formation and disappearance (Champollion et al., 2013) can also be adopted."*

**Reference**

Arndt, S. and Haas, C.: Spatiotemporal variability and decadal trends of snowmelt processes on Antarctic sea ice observed by satellite scatterometers, The Cryosphere, 13, 1943–1958, https://doi.org/10.5194/tc-13-1943-2019, 2019.

Champollion, N., Picard, G., Arnaud, L., Lefebvre, E., and Fily, M.: Hoar crystal development and disappearance at Dome C, Antarctica: observation by near-infrared photography and passive microwave satellite, The Cryosphere, 7, 1247–1262, https://doi.org/10.5194/tc-7-1247-2013, 2013.

Fraser, A. D., Nigro, M. A., Ligtenberg, S. R. M., Legrésy, B., Inoue, M., Cassano, J. J., Kuipers Munneke, P., Lenaerts, J. T. M., Young, N. W., Treverrow, A., van den Broeke, M., and Enomoto, H.: Drivers of ASCAT C band backscatter variability in the dry snow zone of Antarctica, Journal of Glaciology, 62, 170–184, https://doi.org/10.1017/jog.2016.29, 2016.

Larue, F., Picard, G., Aublanc, J., Arnaud, L., Robledano-Perez, A., Meur, E. L., Favier, V., Jourdain, B., Savarino, J., and Thibaut, P.: Radar altimeter waveform simulations in Antarctica with the Snow Microwave Radiative Transfer Model (SMRT), Remote Sensing of Environment, 263, 112534, https://doi.org/10.1016/j.rse.2021.112534, 2021.

Picard, G., Brucker, L., Fily, M., Gallée, H., and Krinner, G.: Modeling time series of microwave brightness temperature in Antarctica, Journal of Glaciology, 55, 537–551, https://doi.org/10.3189/002214309788816678, 2009.

Picard, G., Löwe, H., and Mätzler, C.: Brief communication: A continuous formulation of microwave scattering from fresh snow to bubbly ice from first principles, The Cryosphere, 16, 3861–3866, https://doi.org/10.5194/tc-16-3861-2022, 2022.

Remy, F., Flament, T., Michel, A., and Blumstein, D.: Envisat and SARAL/AltiKa Observations of the Antarctic Ice Sheet: A Comparison Between the Ku-band and Ka-band, Mar. Geod., 38, 510–521, https://doi.org/10.1080/01490419.2014.985347, 2015.

Surdyk, S.: Using microwave brightness temperature to detect short-term surface air temperature changes in Antarctica: An analytical approach, Remote Sensing of Environment, 80, 256–271, https://doi.org/10.1016/s0034-4257(01)00308-x, 2002.

Tran, N., Remy, F., Feng, H., and Femenias, P.: Snow Facies Over Ice Sheets Derived From Envisat Active and Passive Observations, IEEE Transactions on Geoscience and Remote Sensing, 46, 3694–3708, https://doi.org/10.1109/tgrs.2008.2000818, 2008.

van den Broeke, M.: Depth and Density of the Antarctic Firn Layer, Arctic, Antarctic, and Alpine Research, Vol. 40, No. 2, 432–438, https://doi.org/10.1657/1523-0430(07-021)[BROEKE]2.0.CO;2, 2008.

---

## Author Response (AR4)

**Response to Referees on egusphere-2023-1556**

First of all, we would like to thank Referee 2 for the approval of the manuscript. Since there is no comment from Referee 2, this document only includes the response to Referee 3. Please find the reply below, with the original comments in *italics* and the responses in blue. In addition, we would also like to appreciate the Editor for considering and supporting our manuscript.

*This paper used machine learning (ML) and satellite microwave data to examine Antarctic firn density. The authors did a good job of responding to all comments in the first two round of review. I only have one comment.*

*The addition of Appendix A with SMRT simulation is good except the claims that C-band data is sensitive to changes in snow layers up to 20 m. The change in dB reported in the graph are negligible. A change in 0.1 dB is not. The usual radiometric uncertainty is around 1 dB (Schmidt et al. 2018), everything under that can be noise. I would remove all the statements that C-band is sensitive to snow layers up to 20 m.*

*The C-band signal (useful in the density RF model) is probably not sensitive to volume scattering like the passive data but is sensitive to change in snow surface roughness condition (wind) and snow permittivity which are both link to snow surface density. This could be shown by using a surface scattering model like IEM in SMRT.*

We appreciate the suggestion of the referee, therefore we have adopted the IEM theory in SMRT to model the surface scattering and the sensitivity of C-band. This experiment varies the surface roughness, and obtains the sensitivity by comparing the $\sigma_A^0$ of a rough surface with the $\sigma_A^0$ of a smooth surface. The outcome has been added in the revised manuscript as Fig. A2 (or Fig. R1 as attached below). It is true that applying the surface scattering model can result in a sufficient sensitivity of C-band backscatter (exceeding 1 dB). We have also removed the statement related to C-band data being sensitive to changes in snow layers up to 20 m in the revised manuscript.

[Figure]

*Figure R1. Change in $\sigma_A^0$ when the surface roughness (expressed as root mean square heights; rms) is changed.*

---

## Author Response (AR5)

**Response to the Editor on egusphere-2023-1556**

We would like to thank the Editor for the final check of and the suggestions to our manuscript. Please find the response below in blue.

1. Caption for Figure A2 should specify this is C-band

The note "C-band" has been added in the caption.

2. Line 420. Add in a sentence to explain that C-band is more sensitive to surface roughness and reference the appendix.

We have added "Finally, C-band microwave is more sensitive to surface roughness than to densities at larger depths (as shown in Appendix A)."